# GOTENNET: RETHINKING EFFICIENT 3D EQUIVARIANT GRAPH NEURAL NETWORKS

**Sarp Aykent**[1*] **and Tian Xia**[2*]
[1]Comcast AI Technologies, [2]Microsoft
https://github.com/sarpaykent/GotenNet

## ABSTRACT

Understanding complex three-dimensional (3D) structures of graphs is essential for accurately modeling various properties, yet many existing approaches struggle with fully capturing the intricate spatial relationships and symmetries inherent in such systems, especially in large-scale, dynamic molecular datasets. These methods often must balance trade-offs between expressiveness and computational efficiency, limiting their scalability. To address this gap, we propose a novel Geometric Tensor Network (GotenNet) that effectively models the geometric intricacies of 3D graphs while ensuring strict equivariance under the Euclidean group E(3). Our approach directly tackles the expressiveness-efficiency trade-off by leveraging effective geometric tensor representations without relying on irreducible representations or Clebsch-Gordan transforms, thereby reducing computational overhead. We introduce a unified structural embedding, incorporating geometry-aware tensor attention and hierarchical tensor refinement that iteratively updates edge representations through inner product operations on high-degree steerable features, allowing for flexible and efficient representations for various tasks. We evaluated models on QM9, rMD17, MD22, and Molecule3D datasets, where the proposed model consistently outperforms state-of-the-art methods in both scalar and high-degree property predictions, demonstrating exceptional robustness across diverse datasets, and establishes GotenNet as a versatile and scalable framework for 3D equivariant Graph Neural Networks.

## 1 INTRODUCTION

Accurately modeling 3D molecular systems is increasingly crucial in areas such as drug discovery (Chen et al., 2020; Jing et al., 2021; Nguyen et al., 2021; Huang et al., 2020; Aykent & Xia, 2022; Yang et al., 2022), materials science (Reiser et al., 2022; Pablo-García et al., 2023; Polat et al., 2025), and structural biology (Zhang et al., 2021; Xia & Ku, 2021; Zhang et al., 2022). These tasks require a precise understanding of the spatial configurations and symmetries inherent in molecular structures, as these factors are fundamental to predicting molecular properties. While predicting scalar molecular properties, such as energy and stability, is challenging, predicting molecular forces is particularly difficult due to the vector nature of forces and their dependence on local geometric environments (Klicpera et al., 2021; Liao & Smidt, 2023; Wang et al., 2024; 2023b; Du et al., 2023). Traditional graph neural networks (GNNs), while effective for general graph-structured data, face difficulties in handling the geometric and topological complexities of 3D molecular systems, where achieving equivariance remains a significant challenge (Thomas et al., 2018; Satorras et al., 2021; Jing et al., 2021; Aykent & Xia, 2023).

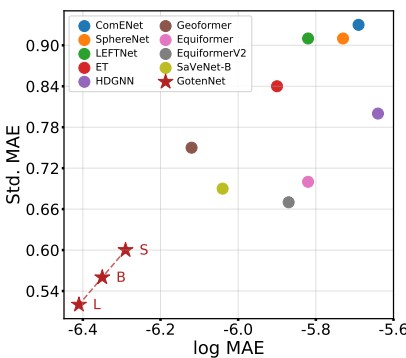

Figure 1: Comparison of GotenNet and baseline models on the QM9 dataset. The $x$-axis shows the logarithmic MAE across all targets, while the $y$-axis shows the standardized MAE. Lower values on both axes indicate better performance. Points marked as S, B, and L represent small, base, and large variations of the GotenNet, respectively.

---

*Sarp Aykent and Tian Xia are co-first authors. Correspond to aykentsarp@gmail.com

Recent advances in equivariant neural networks have led to two distinct approaches (Han et al., 2024): scalarization-based models and high-degree steerable models. Scalarization-based models (Satorras et al., 2021; Zhang & Zhao, 2021; Schütt et al., 2021; Thölke & De Fabritiis, 2022; Aykent & Xia, 2023; Du et al., 2023) operate by projecting 3D geometric information into scalar features before reconstruction, offering computational efficiency and scalability for large-scale applications. However, this projection process may limit expressiveness in capturing complex geometric patterns, particularly in scenarios requiring precise understanding of spatial relationships and symmetries. On the other hand, high-degree steerable models (Batzner et al., 2022; Batatia et al., 2022b; Musaelian et al., 2023; Batatia et al., 2022a; Qiao et al., 2022; Liao & Smidt, 2023; Liao et al., 2024) achieve impressive performance through irreducible representations (irreps) and Clebsch-Gordan (CG) transforms, enabling direct manipulation of geometric features in higher-resolution representation spaces. Despite their theoretical rigor and strong performance, these models incur significant computational overhead due to their reliance on complex tensor operations (Cen et al., 2024; Liao & Smidt, 2023; Liao et al., 2024). This fundamental dichotomy between computational efficiency and geometric expressiveness presents a critical challenge in the field: how to achieve both qualities while maintaining strong performance across diverse molecular properties.

Fundamental breakthroughs in theoretical understanding have revealed promising directions for addressing this challenge. Rather than relying on explicit CG coefficients, recent advances have illuminated how inner product operations can effectively capture similar geometric relationships (Cen et al., 2024) while being computationally more tractable. This insight suggests the potential for more efficient architectures that maintain the expressiveness of high-degree representations without the computational burden of explicit CG transforms. However, translating this theoretical understanding into practical architectures remains challenging – it requires not only a novel formulation of geometric operations but also careful consideration of how to maintain numerical stability and computational efficiency at scale. The challenge is evident in existing models' inability to bridge the gap between scalarization-based and high-degree steerable approaches while maintaining practical applicability. Most current architectures (Han et al., 2024; Wang et al., 2024; Liao & Smidt, 2023; Liao et al., 2024) either compromise on expressiveness for efficiency or forfeit computational tractability for geometric accuracy, leaving a clear divide between models optimized for scalar property prediction and those designed for force field calculations, with few achieving strong performance in both domains.

To address these challenges, we propose a novel framework, the Geometric Tensor Network (Goten-Net). Our approach focuses on addressing the trade-off between expressiveness and efficiency. First, we introduce a spherical-scalarization model with an efficient representation and embedding strategy designed specifically with geometric tensors, eliminating the need for irreps and CG transforms, thereby reducing computational complexity without sacrificing the expressiveness required for modeling intricate 3D structures. Second, we present geometry-aware tensor attention and hierarchical tensor refinement mechanisms. These mechanisms enhance transformer-based architectures by refining edge representations through high-degree steerable features, enabling the self-attention mechanism to leverage refined geometric relationships in determining node interactions. This refinement process enriches the attention weights with granular geometric information, allowing more precise modeling of spatial relationships in molecular structures. These innovations allow the model to represent molecular properties across multiple scales, adapting to both broad patterns and fine-grained molecular details. As shown in Figure 1, our model consistently outperforms the baselines on the QM9 dataset, excelling in both standard MAE and log MAE metrics. This highlights GotenNet' ability to maintain accuracy for large-scale properties while ensuring precision for smaller-scale ones, resulting in strong overall performance across diverse molecular properties.

Through rigorous evaluations on benchmark datasets—QM9, Molecule3D, rMD17, and MD22—our approach consistently outperforms state-of-the-art methods, even in its smallest configuration, establishing GotenNet as a versatile and scalable framework for future developments in 3D equivariant graph neural networks. The demonstrated robustness in predicting both scalar and higher-degree tensor properties highlights its broad potential for applications in fields such as drug discovery, materials science, and molecular dynamics simulations.

## 2 RELATED WORK

The field of machine learning for molecular representation learning has seen significant advancements in recent years (Gasteiger et al., 2020b; Liu et al., 2022; Gasteiger et al., 2020a; Wang et al., 2022; Liao & Smidt, 2023; Liao et al., 2024), especially for the development of graph neural networks (GNNs) to predict quantum mechanical properties and simulate molecular dynamics. These approaches can be broadly categorized into two main groups: invariant GNNs and equivariant GNNs.

### 2.1 INVARIANT GNNS

Invariant GNNs focus on extracting rotation and translation invariant features from molecular graphs (Schütt et al., 2017; Xie & Grossman, 2018; Unke & Meuwly, 2019; Gasteiger et al., 2020b;a; Klicpera et al., 2021; Liu et al., 2022; Wang et al., 2022). DimeNet (Gasteiger et al., 2020b) introduced the concept of directional message passing, embedding messages between atoms instead of atoms themselves. This approach allowed the incorporation of angular information while maintaining rotational equivariance. GemNet (Klicpera et al., 2021) extended this idea by incorporating dihedral angles, and SphereNet (Liu et al., 2022) efficiently integrated torsion information in the message passing scheme. ComENet (Wang et al., 2022) built upon these approaches, introducing a novel message passing scheme that operates within 1-hop neighborhoods and achieves global and local completeness in incorporating 3D information.

### 2.2 EQUIVARIANT GNNS

Equivariant GNNs, on the other hand, directly model rotational equivariance and translational invariance in their architectures (Thomas et al., 2018; Kondor et al., 2018; Schütt et al., 2021; Jing et al., 2021; Thölke & De Fabritiis, 2022; Aykent & Xia, 2022; Le et al., 2022; Du et al., 2022; 2023; Aykent & Xia, 2023; Wang et al., 2023b; 2024). Current equivariant GNNs can be divided into two primary approaches based on their feature processing strategies (Han et al., 2024): scalarization-based models and high-degree steerable models. Scalarization-based models focus on deriving invariant scalar features from 3D coordinates and then reconstruct directional information for equivariant updates (Han et al., 2024; Satorras et al., 2021). Several models have successfully implemented this strategy: PaiNN (Schütt et al., 2021) incorporated both scalar and vectorial features in its message passing framework, LEFTNet (Du et al., 2023) developed local frame-based representations with structural encodings, and TorchMD-NET (Thölke & De Fabritiis, 2022) introduced an equivariant transformer architecture that processes scalar and vector features separately.

High-degree steerable models leverage high-degree representations and CG tensor products for molecular modeling (Batzner et al., 2022; Batatia et al., 2022b; Qiao et al., 2022; Batatia et al., 2022a; Musaelian et al., 2023; Liao & Smidt, 2023; Liao et al., 2024). SE(3)-Transformer (Fuchs et al., 2020) pioneered this direction by introducing attention mechanisms with high-degree steerable features, through computational limitations arose from tensor product operations. Subsequent works like NequIP (Batzner et al., 2022) and MACE (Batatia et al., 2022b) introduced approaches using CG coefficients for equivariance, while Allegro (Musaelian et al., 2023) introduced a local equivariant architecture using iterated tensor products. Equiformer (Liao & Smidt, 2023) introduced $SE(3)$ equivariance into Transformers through depth-wise tensor products and MLP-based attention mechanisms. Building on this work, EquiformerV2 (Liao et al., 2024) enhanced computational efficiency by incorporating eSCN (Passaro & Zitnick, 2023), although its Fibonacci grid sampling approach incurs an $\mathcal{O}(L^3)$ computational overhead in achieving quasi-equivariance. While these methods show impressive performance, their computational overhead remains significant.

Recent approaches have explored alternative methods to capture geometric information without relying on tensor products and CG coefficients, offering computationally efficient solutions. ViSNet (Wang et al., 2024) linked inner products and Legendre polynomials via vector-scalar interactive message passing, through focusing on the first-degree steerable features. SO3KRATES (Frank et al., 2024) demonstrated that certain applications of CG coefficients are equivalent to inner products of high-degree steerable features, achieving notable performance in property prediction tasks through their equivariant transformer architecture. HEGNN (Cen et al., 2024) further developed these concepts by introducing a scalarization approach using inner products to incorporate high-degree steerable features. This approach proved capable of capturing complete angular information between edge pairs

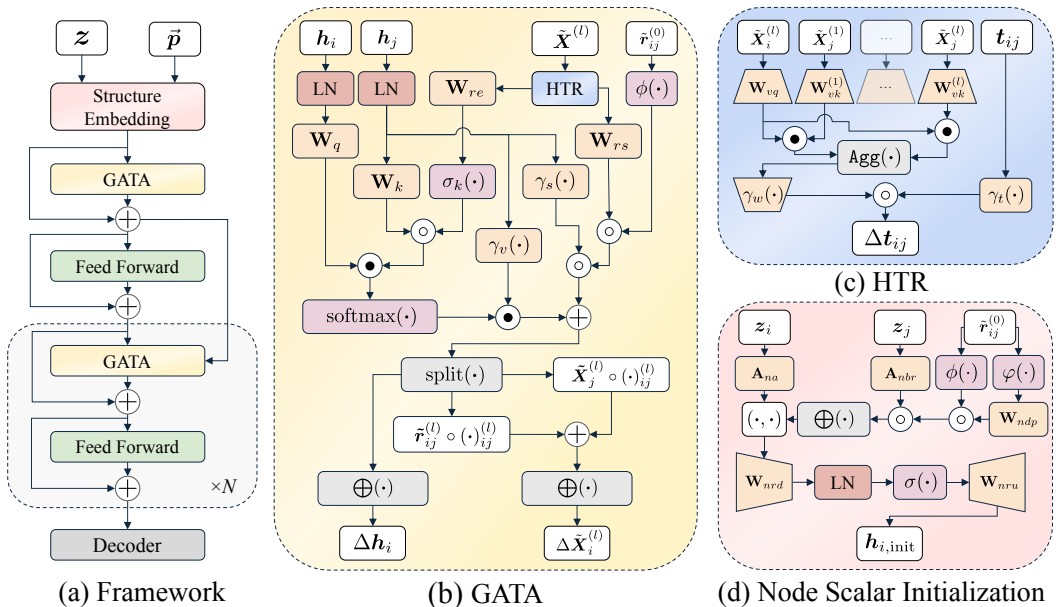

Figure 2: Architecture of GotenNet. The overall framework (a) includes an embedding, an interaction module, and a decoder; (b) shows the geometry-aware tensor attention (GATA); (c) illustrates the hierarchical tensor refinement (HTR); and (d) presents the node scalar feature initialization component, which is part of the structure embedding module. In the figure, $+$ denotes addition, $\cdot$ denotes dot product, $\oplus$ denotes aggregation, $(\cdot, \cdot)$ denotes concatenation, $\circ$ denotes element-wise (Hadamard) product with broadcasting when tensor shapes are different, LN denotes layer normalization, $\varphi$ denotes the radial basis functions, and $\gamma$ denotes differentiable functions such as MLPs.

and demonstrating enhanced model robustness in dynamics tasks. Our work advances this direction by introducing geometry-aware tensor attention, which employs a concise formulation of inner product operations combined with hierarchical refinement mechanisms. GotenNet represents a significant advancement in bridging the critical gap between scalarization-based and high-degree steerable models. The proposed spherical-scalarization model, GotenNet, achieves superior performance in real-world molecular property prediction and force field calculations across diverse datasets.

## 3 GOTENNET

In this section, we present the key components of GotenNet. Our network maintains and updates two types of node features (invariant and steerable) and edge features (invariant) throughout its operations. We first introduce an efficient initialization scheme (Sec. 3.2) that embeds geometric tensors without requiring irreps or CG transforms, significantly reducing computational complexity. The network then processes these representations through three main mechanisms: (1) a degree-wise attention-based message passing layer (Sec. 3.3) that updates both invariant and steerable features while preserving equivariance, (2) a high-degree edge refinement layer (Sec. 3.4) that updates edge features across degrees with inner products of steerable features, and (3) a node-wise feed-forward refinement layer (Sec. 3.5) that further processes both types of node features. This architecture enables accurate and scalable predictions while maintaining geometric consistency throughout the network.

### 3.1 EQUIVARIANT GEOMETRIC TENSOR REPRESENTATIONS

In our model, we distinguish between edge scalar features and edge tensor representations, employing spherical harmonics to initialize the latter. The edge tensor representation $\tilde{r}_{ij}^{(l)}$ is initialized based on the relative positions $\vec{p}_i$ and $\vec{p}_j$ of nodes $i$ and $j$, capturing spatial information from rank 0 to $L_{\max}$. Specifically, $\tilde{r}_{ij} = \{\tilde{r}^{(0)}, \tilde{r}^{(1)}, \ldots, \tilde{r}^{(L_{\max})}\}$, where each $\tilde{r}^{(l)}$ represents $l$-degree spherical harmonic functions. The components of $\tilde{r}_{ij}$ follow a hierarchical structure of increasing geometric complexity.

At the most basic level, $\tilde{\boldsymbol{r}}_{ij}^{(0)} = \|\vec{\boldsymbol{p}}_i - \vec{\boldsymbol{p}}_j\|$ captures the scalar distance between nodes, providing rotation and translation invariant information. The first-degree component $\tilde{\boldsymbol{r}}_{ij}^{(1)} = (\vec{\boldsymbol{p}}_i - \vec{\boldsymbol{p}}_j)/\|\vec{\boldsymbol{p}}_i - \vec{\boldsymbol{p}}_j\|$ encodes directional information, introducing rotational equivariance. For $l \geq 2$, each $\tilde{\boldsymbol{r}}_{ij}^{(l)}$ comprises $(2l+1)$ functions derived from spherical harmonics of degree $l$, where the degree determines the transformation behavior under rotations, and the parity of $l$ determines the behavior under inversion. These functions are chosen to capture complex spatial relationships and rotational symmetries inherent in molecular structures. Leveraging the inherent normalization property of spherical harmonics, each $\tilde{\boldsymbol{r}}^{(l)}$ for $l \geq 1$ is naturally normalized, ensuring consistent scaling across different representations.

We denote the geometric node tensors into two types of features: scalar features $\boldsymbol{h} \in \mathbb{R}^{d_{ne}}$ invariant under transformations, and high-degree steerable features $\tilde{\boldsymbol{X}}^{(l)} \in \mathbb{R}^{(1+2l) \times d_{ne}}$ whose transformations depend on their degree $l$ where $d_{ne}$ denotes node embedding dimension. We denote steerable features with a tilde, $\tilde{\cdot}$, where feature of degree $l$ is represented by $1 + 2l$ components. These representations are initialized and updated through message passing phases using the edge tensor representation $\tilde{\boldsymbol{r}}_{ij}$ and edge scalar features $\boldsymbol{t}_{ij}$ as input. The notation $\tilde{\boldsymbol{X}}$ without a specified degree $l$ refers to the collection of features with degrees from 1 to $L_{\max}$. This initialization strategy enables our model to effectively capture, process, and propagate complex structural information.

Our initialization and feature design ensure equivariance throughout the network. A geometric tensor field maps 3D points to tensor quantities that transform equivariantly under geometric transformations, combining both invariant scalars and steerable features. Each layer of GotenNet processes these tensor fields through equivariant operations while preserving $E(3)$ transformations, with the final layer producing either equivariant geometric features or invariant representations as required by the task. This composition of equivariant operations ensures that the entire network maintains equivariance, with complete proofs provided in the Appendices A, B, and C.

## 3.2 Unified Structural Embedding: Integrating Content and Geometry

Our approach introduces a unified structure embedding that captures intrinsic atomic properties and relational information through an integrated node-edge interaction mechanism. By employing a dual representation strategy, we incorporate local geometric structure through node-edge interaction. This allows the model to simultaneously process both semantic and geometric information, enabling efficient message passing for both nodes and edges.

**Node Scalar Feature Initialization.** Node scalar features are obtained through a two-step process involving message passing and representation updates. Node information is aggregated as:

$$\mathbf{m}_i = \sum_{j \in \mathcal{N}(i)} \boldsymbol{z}_j \mathbf{A}_{\text{nbr}} \circ \left( \varphi(\tilde{\boldsymbol{r}}_{ij}^{(0)}) \mathbf{W}_{\text{ndp}} \circ \phi(\tilde{\boldsymbol{r}}_{ij}^{(0)}) \right), \tag{1}$$

where $\boldsymbol{z} \in \mathbb{R}^{|\mathcal{Z}|}$ denotes the one-hot encoding of the atomic number, $\circ$ denotes element-wise product, and $\mathbf{A}_{\text{nbr}} \in \mathbb{R}^{|\mathcal{Z}| \times d_{ne}}$ is a learnable embedding matrix for neighbor atoms with maximum atomic number $|\mathcal{Z}|$. The basis functions $\varphi(\tilde{\boldsymbol{r}}_{ij}^{(0)})$ encode the distance between nodes $i$ and $j$, which are then projected through $\mathbf{W}_{\text{ndp}}$. A cutoff function $\phi(\tilde{\boldsymbol{r}}_{ij}^{(0)})$ is applied to modulate the influence of distant neighbors. It is important to note that the $\circ$ operation applies standard broadcasting rules when the shapes of the two operands are not identical. Specifically, if the tensors have mismatched shapes, the tensor with fewer dimensions or smaller size in a dimension is automatically expanded to match the shape of the other tensor before the element-wise multiplication. For example, in Equation (1), consider the shapes: $(\varphi(\tilde{\boldsymbol{r}}_{ij}^{(0)}) \mathbf{W}_{\text{ndp}}) \in \mathbb{R}^{d_{ne}}$ and $\phi(\tilde{\boldsymbol{r}}_{ij}^{(0)}) \in \mathbb{R}^1$. Here, $\phi(\tilde{\boldsymbol{r}}_{ij}^{(0)})$ is broadcast along the first dimension to conform to the shape of $(\varphi(\tilde{\boldsymbol{r}}_{ij}^{(0)}) \mathbf{W}_{\text{ndp}})$ before the element-wise product is computed. Consequently, the aggregated node information, $\mathbf{m}_i$, is produced with shape $\mathbb{R}^{d_{ne}}$.

The initial node scalar feature is defined as:

$$\boldsymbol{h}_{i,\text{init}} = \sigma\Big( \text{LN}\big( (\boldsymbol{z}_i \mathbf{A}_{\text{na}}, \mathbf{m}_i) \mathbf{W}_{\text{nrd}} \big) \Big) \mathbf{W}_{\text{nru}}. \tag{2}$$

Here, $\mathbf{A}_{\text{na}} \in \mathbb{R}^{|\mathcal{Z}| \times d_{ne}}$ is a learnable embedding matrix for node atoms, $\sigma$ denotes a non-linear activation function, and $(\cdot, \cdot)$ denotes concatenation operation. The concatenated node atom embedding and

aggregated neighbor information undergo a series of transformations: node representation projections ($\mathbf{W}_{\text{nrd}}$, $\mathbf{W}_{\text{nru}}$), and layer normalization (LN).

**Edge Scalar Feature Initialization.** To effectively model interactions, edge scalar features are computed by combining node features with distance-based edge attributes:

$$\boldsymbol{t}_{ij,\text{init}} = (\boldsymbol{h}_{i,\text{init}} + \boldsymbol{h}_{j,\text{init}}) \circ \left( \varphi(\tilde{\boldsymbol{r}}^{(0)}{}_{ij}) \mathbf{W}_{\text{erp}} \right). \tag{3}$$

Edge attributes $\boldsymbol{t}_{ij} \in \mathbb{R}^{d_{ed}}$ are processed through projection matrix $\mathbf{W}_{\text{erp}}$, enabling the integration of node-level features and spatial relationships. This formulation captures complex interactions between nodes while maintaining equivariance under molecular transformations.

**High-degree Steerable Feature Initialization.** The high-degree steerable features $\tilde{\boldsymbol{X}}$ initialized during initial interaction layer with the following formulation:

$$\begin{aligned}
\{\boldsymbol{o}^{(l)}_{ij,\text{init}}\}^{L_{\max}}_{l=1} &= \text{split}\Big( \mathbf{sea}_{ij} + (\boldsymbol{t}_{ij,\text{init}} \mathbf{W}_{rs,\text{init}}) \circ \gamma_s(\boldsymbol{h}_{j,\text{init}}) \circ \phi(\tilde{\boldsymbol{r}}^{(0)}_{ij}), d_{ne} \Big), \\
\tilde{\boldsymbol{X}}^{(l)}_{i,\text{init}} &= \bigoplus_{j \in \mathcal{N}(i)} \left( \boldsymbol{o}^{(l)}_{ij,\text{init}} \circ \tilde{\boldsymbol{r}}^{(l)}_{ij} \right),
\end{aligned} \tag{4}$$

where $\mathbf{sea}_{ij}$ is self-attention with geometric encoding, $\mathbf{W}_{rs,\text{init}} \in \mathbb{R}^{d_{ed} \times d_{ne}}$ is a learnable weight matrix, $\gamma_s : \mathbb{R}^{d_{ne}} \to \mathbb{R}^{L_{\max} \times d_{ne}}$ is a differentiable function, and the split function decomposes the input tensor into $d_{ne}$-dimensional segments. These segments are used as different coefficients for each $l$-degree steerable features. $\bigoplus$ denotes a permutation-invariant aggregation function.

### 3.3 Geometry-Aware Tensor Attention

We introduce a novel module called , which enhances the attention mechanism in graph neural networks by incorporating spatial information. GATA captures the geometric relationships between nodes to improve attention-driven message passing.

The GATA module combines self-attention with geometric encoding to generate rich node interaction representations. We compute the query ($\boldsymbol{q}$), key ($\boldsymbol{k}$), and value ($\boldsymbol{v}$) representations:

$$\boldsymbol{q}_i = \boldsymbol{h}_i \mathbf{W}_q, \quad \boldsymbol{k}_j = \boldsymbol{h}_j \mathbf{W}_k, \quad \boldsymbol{v}_j = \gamma_v(\boldsymbol{h}_j), \tag{5}$$

where $\mathbf{W}_q, \mathbf{W}_k \in \mathbb{R}^{d_{ne} \times d_{ne}}$ are learnable weight matrices, and $\gamma_v : \mathbb{R}^{d_{ne}} \to \mathbb{R}^{S \cdot d_{ne}}$ is a differentiable function (e.g., MLP). The $S$ variable introduced to generate different coefficients for each degree of steerable features and is defined as $1 + 2 \times L_{\max}$. The attention coefficients $\boldsymbol{\alpha}_{ij}$ between nodes $i$ and $j$ using the dot product of the query vector $\boldsymbol{q}_i$ and a geometry-infused key vector, which is obtained via an element-wise product of $\boldsymbol{k}_j$ and a transformed edge embedding:

$$\mathbf{sea}_{ij} = \frac{\exp(\boldsymbol{\alpha}_{ij})}{\sum_{k \in \mathcal{N}(i)} \exp(\boldsymbol{\alpha}_{ik})} \boldsymbol{v}_j, \quad \text{where} \quad \boldsymbol{\alpha}_{ij} = \boldsymbol{q}_i \big( \boldsymbol{k}_j \circ \sigma_k(\boldsymbol{t}_{ij} \mathbf{W}_{re}) \big)^{\mathrm{T}}. \tag{6}$$

Here, $\sigma_k$ denotes a non-linear activation function, and $\mathbf{W}_{re} \in \mathbb{R}^{d_{ed} \times d_{ne}}$ is a learnable weight matrix that transforms the edge scalar features. To incorporate spatial and directional information, we augment the attention mechanism with geometric encoding. The GATA operation combines self-attention with geometric features and is then split into $S$ components:

$$\boldsymbol{o}^s_{ij}, \{\boldsymbol{o}^{d,(l)}_{ij}\}^{L_{\max}}_{l=1}, \{\boldsymbol{o}^{t,(l)}_{ij}\}^{L_{\max}}_{l=1} = \text{split}(\mathbf{sea}_{ij} + (\boldsymbol{t}_{ij} \mathbf{W}_{rs}) \circ \gamma_s(\boldsymbol{h}_j) \circ \phi(\tilde{\boldsymbol{r}}^{(0)}_{ij}), d_{ne}), \tag{7}$$

where $\mathbf{W}_{rs} \in \mathbb{R}^{d_{ed} \times (S \cdot d_{ne})}$ is a learnable weight matrix, $\gamma_s : \mathbb{R}^{d_{ne}} \to \mathbb{R}^{S \cdot d_{ne}}$ is a differentiable function, and the split function decomposes the input tensor into $d_{ne}$-dimensional segments. We define $\Delta \boldsymbol{h}_i$ and $\Delta \tilde{\boldsymbol{X}}$ as the residues, which are calculated by:

$$\Delta \boldsymbol{h}_i = \bigoplus_{j \in \mathcal{N}(i)} (\boldsymbol{o}^s_{ij}), \quad \Delta \tilde{\boldsymbol{X}}^{(l)}_i = \bigoplus_{j \in \mathcal{N}(i)} \left( \boldsymbol{o}^{d,(l)}_{ij} \circ \tilde{\boldsymbol{r}}^{(l)}_{ij} + \boldsymbol{o}^{t,(l)}_{ij} \circ \tilde{\boldsymbol{X}}^{(l)}_j \right). \tag{8}$$

Here, each degree $l \in [1, L_{\max}]$ contributes its own component of steerable features weighted by their respective coefficients $\boldsymbol{o}^{d,(l)}_{ij}$ and $\boldsymbol{o}^{t,(l)}_{ij}$. Finally, representations are updated using residues with:

$$\boldsymbol{h}_i \leftarrow \boldsymbol{h}_i + \Delta \boldsymbol{h}_i, \quad \tilde{\boldsymbol{X}}^{(l)}_i \leftarrow \tilde{\boldsymbol{X}}^{(l)}_i + \Delta \tilde{\boldsymbol{X}}^{(l)}_i, \tag{9}$$

By infusing geometric information into the attention mechanism, GATA allows the model to better capture spatial dependencies and fine-grained node interactions, leading to improved performance in molecular property predictions, as demonstrated in Section 4.

### 3.4 HIERARCHICAL TENSOR REFINEMENT

The Hierarchical Tensor Refinement (HTR) component processes graph-structured data through multi-scale analysis and layer-wise refinement. High-degree steerable features are projected to query and key representations using degree-specific SO(3)-equivariant linear transformation (Deng et al., 2021; Du et al., 2023; Wang et al., 2024) as shown in Equation (10):

$$\widetilde{\boldsymbol{EQ}}_i^{(l)} = \tilde{\boldsymbol{X}}_i^{(l)} \mathbf{W}_{vq}, \quad \widetilde{\boldsymbol{EK}}_j^{(l)} = \tilde{\boldsymbol{X}}_j^{(l)} \mathbf{W}_{vk}^{(l)}, \quad \text{for } l \in \{1, \dots, L_{\max}\}, \tag{10}$$

where $\mathbf{W}_{vq}, \mathbf{W}_{vk}^{(l)} \in \mathbb{R}^{d_{ed} \times d_{xpd}}$ are tensor query and key projection matrices, respectively. Here, $\mathbf{W}_{vq}$ is a shared projection matrix across degrees, while $\mathbf{W}_{vk}^{(l)}$ is degree-specific. To preserve equivariance, uniform weights are applied across spatial dimensions. These projections aggregate angular and magnitude information between nodes across tensor degrees, defined as:

$$\boldsymbol{w}_{ij} = \texttt{Agg}_{l=1}^{L_{\max}}\left( (\widetilde{\boldsymbol{EQ}}_i^{(l)})^\top \widetilde{\boldsymbol{EK}}_j^{(l)} \right), \tag{11}$$

where $\boldsymbol{w}_{ij} \in \mathbb{R}^{d_{xpd}}$ represents the aggregated similarity between nodes $i$ and $j$, and $\texttt{Agg}_{l=1}^{L_{\max}}$ denotes an denotes the aggregation over high-degree steerable features. The aggregated information refines edge representations through a residual connection:

$$\boldsymbol{t}_{ij} \leftarrow \boldsymbol{t}_{ij} + \Delta \boldsymbol{t}_{ij}, \quad \Delta \boldsymbol{t}_{ij} = \gamma_w(\boldsymbol{w}_{ij}) \circ \gamma_t(\boldsymbol{t}_{ij}), \tag{12}$$

where $\gamma_w : \mathbb{R}^{d_{xpd}} \to \mathbb{R}^{d_{ed}}$ and $\gamma_t : \mathbb{R}^{d_{ed}} \to \mathbb{R}^{d_{ed}}$ are differentiable functions. In some cases, $d_{xpd}$ is set larger than $d_{ed}$ for richer intermediate representations.

### 3.5 EQUIVARIANT FEED-FORWARD (EQFF) NETWORKS

The EQFF blocks, employed after GATA, facilitates efficient channel-wise interaction while maintaining equivariance. By design, the module separates scalar and high-degree steerable features, allowing for specialized processing of each feature type before combining them with non-linear mappings:

$$\text{EQFF}(\boldsymbol{h}, \tilde{\boldsymbol{X}}^{(l)}) = \left( (\boldsymbol{h} + \boldsymbol{m}_1), (\tilde{\boldsymbol{X}}^{(l)} + (\boldsymbol{m}_2 \circ \tilde{\boldsymbol{X}}^{(l)} \mathbf{W}_{vu})) \right),$$
$$\text{where } \boldsymbol{m}_1, \boldsymbol{m}_2 = \text{split}_2\left( \gamma_m (||\tilde{\boldsymbol{X}}^{(l)} \mathbf{W}_{vu}||_2, \boldsymbol{h}) \right). \tag{13}$$

Here $\gamma_m$ denote differentiable functions such as MLPs, $\mathbf{W}_{vu}$ denotes learnable weight matrices, $(\cdot, \cdot)$ denotes concatenation, and $|| \cdot ||_2$ denotes $L_2$ norm. The EQFF module operates on the tensor representations while separating the scalar and high-degree steerable features and combining them through element-wise operations. The use of $\gamma_m$ enables the model to learn complex non-linear mappings, enhancing its expressiveness (Ramachandran et al., 2017; Elfwing et al., 2018).

## 4 EXPERIMENTS

In this section, we compare the performance of GotenNet with other state-of-the-art methods. Experiments were conducted with an NVIDIA A100 GPU with 80GB video memory, 512GB RAM, and an AMD EPYC 7713P CPU. We evaluated the models on QM9, rMD17, MD22, and Molecule3D datasets. The best results are **bolded** and the second best are underlined. Additional details on hyperparameters and scalability, as well as additional experiments, can be found in the Appendix E.

### 4.1 QM9 DATASET

**Dataset.** The proposed method is evaluated against a comprehensive set of baselines using the QM9 dataset (Ruddigkeit et al., 2012; Ramakrishnan et al., 2014). These baselines include Cormorant (Anderson et al., 2019), ClofNet (Du et al., 2022), NMP (Gilmer et al., 2017), EGNN (Satorras et al., 2021), SEGNN (Brandstetter et al., 2022), PaiNN (Schütt et al., 2021), DimeNet++ (Gasteiger et al., 2020a), ComENet (Wang et al., 2022), SphereNet (Liu et al., 2022), LEFTNet (Du et al., 2023), EQGAT (Le et al., 2022), ET (Thölke & De Fabritiis, 2022), HDGNN (An et al., 2024), Geoformer (Wang et al., 2023a), Equiformer (Liao & Smidt, 2023), SaVeNet (Aykent & Xia, 2023),

Table 1: Performance comparisons on QM9 dataset. † denotes using different data partitions.

| Task | $\alpha$ | $\Delta\varepsilon$ | $\varepsilon_{\text{HOMO}}$ | $\varepsilon_{\text{LUMO}}$ | $\mu$ | $C_\nu$ | $G$ | $H$ | $R^2$ | $U$ | $U_0$ | ZPVE | std. | log |
| Units | $ma_0^3$ | meV | meV | meV | mD | $\frac{\text{mcal}}{\text{mol K}}$ | meV | meV | $ma_0^2$ | meV | meV | meV | % | - |
|---|---|---|---|---|---|---|---|---|---|---|---|---|---|---|
| DimeNet++† | 44 | 32.6 | 24.6 | 19.5 | 29.7 | 23 | 7.56 | 6.53 | 331 | 6.28 | 6.32 | 1.21 | 0.98 | -5.67 |
| SphereNet† | 46 | 31.1 | 22.8 | 18.9 | 24.5 | 22 | 7.78 | 6.33 | 268 | 6.36 | 6.26 | 1.12 | 0.91 | -5.73 |
| LEFTNet | 48 | 40 | 24 | 18 | 12 | 23 | 7 | 6 | 109 | 7 | 6 | 1.33 | 0.91 | -5.82 |
| EQGAT | 53 | 32 | 20 | 16 | 11 | 24 | 23 | 24 | 382 | 25 | 25 | 2.00 | 0.86 | -5.28 |
| ET | 59 | 36.1 | 20.3 | 17.5 | 11 | 26 | 7.62 | 6.16 | 33 | 6.38 | 6.15 | 1.84 | 0.84 | -5.90 |
| HDGNN† | 46 | 32 | 18 | 16 | 17 | 23 | 11 | 10 | 342 | 8.12 | 8.34 | 1.21 | 0.80 | -5.64 |
| Geoformer | 40 | 33.8 | 18.4 | 15.4 | 10 | 22 | 6.13 | 4.39 | 28 | 4.41 | 4.43 | 1.28 | 0.75 | -6.12 |
| Equiformer | 46 | 30 | 15.4 | 14.7 | 12 | 23 | 7.63 | 6.63 | 251 | 6.74 | 6.59 | 1.26 | 0.70 | -5.82 |
| SaVeNet-B† | 39 | 24.8 | 18.4 | 16.3 | 9.3 | 23 | 6.64 | 5.43 | 58 | 5.48 | 5.43 | 1.18 | 0.69 | -6.04 |
| Equiformer$_{\text{V2}}$ | 47 | 29.0 | 14.4 | 13.3 | 9.9 | 23 | 7.57 | 6.22 | 186 | 6.49 | 6.17 | 1.47 | 0.67 | -5.87 |
| GotenNet$_S$ | **33** | **21.2** | 16.9 | 13.9 | **7.5** | **20** | **5.50** | **3.70** | 29 | **3.67** | **3.71** | 1.09 | **0.60** | -6.29 |
| GotenNet$_B$ | **32** | **20.5** | 15.2 | **13.0** | **7.2** | **19** | **5.19** | **3.44** | 27 | **3.49** | **3.43** | 1.09 | **0.56** | -6.35 |
| GotenNet$_L$ | **28** | **19.8** | **13.4** | **12.2** | **6.7** | **19** | **4.98** | **3.30** | **24** | **3.41** | **3.37** | **1.08** | **0.52** | **-6.41** |

EquiformerV2 (Liao et al., 2024), Transformer-M (Luo et al., 2022), GeoSSL-DDM (Liu et al., 2023), 3D-EMGP (Jiao et al., 2023), Coord (Zaidi et al., 2023), Frad (Ni et al., 2024a), SliDe (Ni et al., 2024b) and DenoiseVAE (Liu et al., 2025). Table 1 presents only the ten baseline methods with the lowest std. MAEs, while the complete comparison is provided in Appendix I Table 9.

**Model Performance and Size Scaling Analysis.** We evaluate three model variants - small (S), base (B), and large (L) - to analyze both performance and scaling behavior, with detailed specifications in Appendix E. As shown in Table 1, even our smallest variant GotenNet$_S$ outperforms baseline methods on nine out of twelve targets while surpassing baselines on std. MAE and log MAE. GotenNet$_B$ demonstrates further improvements, achieving best performance on eleven targets and significantly improving aggregated metrics, reducing standard MAE by over 16% and log MAE by 3% compared to the best baseline results. The largest variant GotenNet$_L$ achieves state-of-the-art performance across all metrics, although the relative improvement decreases compared to GotenNet$_B$, which suggests that dataset size may become a limiting factor for larger models. To investigate scaling to larger datasets, we conduct experiments in Section 4.2 on the Molecule3D dataset, which contains more than 3 million molecules - an order of magnitude larger than QM9. These results establish GotenNet as the new state-of-the-art while revealing important insights about model scaling behavior.

Table 2: Performance comparisons on Molecule3D dataset.

| Split | Random | | | | | | Scaffold |
| Task | $\mu$ | $\varepsilon_{\text{HOMO}}$ | $\varepsilon_{\text{LUMO}}$ | $\Delta\varepsilon$ | std. | log | $\Delta\varepsilon$ |
|---|---|---|---|---|---|---|---|
| GIN-Virtual | .0882 | .0692 | .0632 | .1036 | .0592 | -2.87 | .2371 |
| SchNet | .0532 | .0275 | .0265 | .0428 | .0263 | -3.66 | .1511 |
| DimeNet++ | .0293 | .0240 | .0190 | .0306 | .0188 | -4.01 | .1214 |
| SphereNet | .0288 | .0239 | .0183 | .0301 | .0184 | -4.03 | .1182 |
| ComENet | .0345 | .0288 | .0252 | .0326 | .0220 | -3.84 | .1273 |
| PaiNN | .0196 | .0263 | .0197 | .0307 | .0182 | -4.08 | .1208 |
| ET | .0223 | .0199 | .0194 | .0303 | .0170 | -4.13 | .1282 |
| LEFTNet | .0151 | .0183 | .0157 | .0275 | .0145 | -4.32 | .1317 |
| SaVeNet-L | .0136 | .0159 | .0143 | .0239 | .0128 | -4.44 | .1082 |
| Geoformer | - | - | - | .0202 | - | - | .1135 |
| GotenNet | **.0103** | **.0108** | **.0112** | **.0165** | **.0103** | **-4.65** | **.1002** |

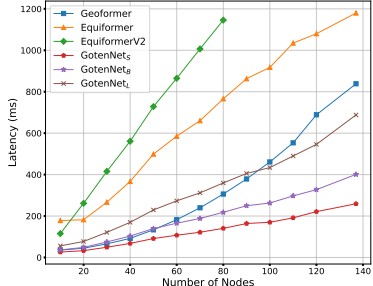

Figure 3: Comparison of training latency of the models with respect to node count on the Molecule3D dataset.

## 4.2 MOLECULE3D DATASET

**Dataset.** We further evaluate our model on the Molecule3D dataset (Xu et al., 2021). This dataset contains over $29\times$ more graphs than QM9, with approximately $1.6\times$ and $1.9\times$ increases in the average number of nodes and edges per graph, providing an ideal benchmark for both model performance and computational scaling. We compare against diverse baseline models, including GIN-Virtual (Wang et al., 2022), SchNet (Schütt et al., 2017), DimeNet++ (Gasteiger et al., 2020a), SphereNet (Liu et al., 2022), ComENet (Wang et al., 2022), PaiNN (Schütt et al., 2021), ET (Thölke & De Fabritiis, 2022), LEFTNet (Du et al., 2023), SaVeNet (Aykent & Xia, 2023), and Geoformer (Wang et al., 2023a).

Table 3: Comprehensive comparison of various molecular modeling methods on MD22 dataset. The results are reported in MAE of energy (kcal/mol) and forces (kcal/mol/Å) denoted as E and F, respectively. $|G|$ denotes the size of the graphs in terms of node count.

| Molecule | $|G|$ | | sGDML | ET | Allegro | MACE | Equiformer | ViSNet | QuinNet | E-LSRM | V-LSRM | SO3KRATES | GotenNet$_S$ | GotenNet$_B$ |
|---|---|---|---|---|---|---|---|---|---|---|---|---|---|---|
| Tetrapeptide | 42 | E | 0.3902 | 0.1121 | 0.1019 | _0.0620_ | 0.0828 | 0.0796 | 0.0840 | 0.0780 | 0.0654 | 0.337 | **0.0589** | **0.0505** |
| | | F | 0.7968 | 0.1879 | 0.1068 | 0.0876 | 0.0804 | 0.0972 | _0.0681_ | 0.0887 | 0.0902 | 0.244 | _0.0719_ | **0.0567** |
| DHA | 56 | E | 1.3117 | 0.1205 | 0.1153 | 0.1317 | 0.1788 | 0.1526 | 0.1200 | 0.0878 | _0.0873_ | 0.379 | **0.0640** | **0.0575** |
| | | F | 0.7474 | 0.1209 | 0.0732 | 0.0646 | _0.0506_ | 0.0668 | 0.0515 | 0.0534 | 0.0598 | 0.242 | **0.0496** | **0.0421** |
| Stachyose | 87 | E | 4.0497 | 0.1393 | 0.2485 | 0.1244 | 0.1404 | 0.1283 | 0.2300 | 0.1252 | _0.1055_ | 0.442 | **0.0751** | **0.0673** |
| | | F | 0.6744 | 0.1921 | 0.0971 | 0.0876 | 0.0635 | 0.0869 | _0.0543_ | 0.0632 | 0.0767 | 0.435 | **0.0512** | **0.0427** |
| AT-AT | 60 | E | 0.7235 | 0.1120 | 0.1428 | 0.1093 | 0.1309 | 0.1688 | 0.1400 | 0.1007 | _0.0772_ | 0.178 | **0.0640** | **0.0544** |
| | | F | 0.6911 | 0.2036 | 0.0952 | 0.0992 | 0.0960 | 0.1070 | _0.0687_ | 0.0881 | 0.0781 | 0.216 | **0.0632** | **0.0478** |
| AT-AT-CG-CG | 118 | E | 1.3885 | 0.2072 | 0.3933 | 0.1578 | 0.1510 | 0.1995 | 0.3800 | 0.1335 | _0.1135_ | 0.345 | **0.0964** | **0.0923** |
| | | F | 0.7028 | 0.3259 | 0.1280 | 0.1153 | 0.1252 | 0.1563 | 0.1273 | 0.1065 | _0.1063_ | 0.332 | **0.0824** | **0.0744** |
| Buckyball catcher | 148 | E | 1.1962 | 0.5188 | 0.5258 | - | _0.3978_ | 0.4421 | 0.5624 | - | 0.4220 | 0.381 | **0.3432** | **0.3032** |
| | | F | 0.6820 | 0.3318 | _0.0887_ | - | 0.1114 | 0.1335 | 0.1091 | - | 0.1026 | 0.237 | **0.0838** | **0.0789** |
| Double-walled nanotube | 370 | E | 4.0122 | 1.4732 | 2.2097 | - | 1.1945 | _1.0339_ | 1.8130 | - | 1.8230 | 0.993 | **0.9993** | **0.6641** |
| | | F | 0.5231 | 1.0031 | 0.3428 | - | _0.2747_ | 0.3959 | 0.2473 | - | 0.3391 | 0.727 | **0.2464** | **0.1888** |

**Model Performance on Large-Scale Data.** As shown in Table 2, GotenNet maintains its superior performance even on this larger Molecule3D dataset, achieving the lowest errors across all tasks, including $\mu$, $\varepsilon_{\text{HOMO}}$, $\varepsilon_{\text{LUMO}}$, and $\Delta\varepsilon$. Notably, GotenNet surpasses the previous best model, SaVeNet-L, by a significant margin of 24% in $\mu$ and more than 32% in $\varepsilon_{\text{HOMO}}$. The best log error of -4.65 in the random split further demonstrates the model's robustness on larger datasets.

**Computational Efficiency Analysis.** Beyond performance metrics, scaling to larger datasets requires efficient handling of increased graph sizes. We analyze computational efficiency by measuring training time across varying node counts (10-140 nodes per graph). Figure 3 compares GotenNet with competitive attention-based baselines including Geoformer (Wang et al., 2023a), Equiformer (Liao & Smidt, 2023), and EquiformerV2 (Liao et al., 2024). Full experimental setup details are provided in Appendix G. The $x$-axis shows the node count, while the $y$-axis shows the training time per batch in milliseconds. The results show GotenNet maintains efficient scaling at higher node counts, while baseline methods like Geoformer, despite strong performance on smaller graphs, become computationally intensive due to their dense $O(n^2)$ representations. Both GotenNet$_S$ and GotenNet$_B$ variants maintain consistent efficiency across all node counts, demonstrating their suitability for large-scale applications where computational overhead is critical.

## 4.3 MD22 DATASET

**Dataset.** The MD22 dataset (Chmiela et al., 2023) contains molecular dynamics trajectories for seven systems, with atom counts from 42 to 370, across four biomolecule and supramolecule classes. It presents challenges in system size, flexibility, and nonlocality, making it a key benchmark for scalability and accuracy in molecular force field models. Following the data splits from (Chmiela et al., 2023), we evaluate GotenNet against several baselines, including sDGML (Chmiela et al., 2018), ET (Thölke & De Fabritiis, 2022), Allegro (Musaelian et al., 2023), MACE (Batatia et al., 2022b), Equiformer (Liao & Smidt, 2023), ViSNet (Wang et al., 2024), QuinNet (Wang et al., 2023b), SO3KRATES (Frank et al., 2024), and LSRM (Li et al., 2024) along with its variants E-LSRM (Equiformer-LSRM) and V-LSRM (ViSNet-LSRM).

**Results.** The MD22 dataset poses significant challenges due to its wide range of molecule sizes, requiring accurate predictions of both energy and forces. We show results of MD22 dataset in Table 3. Our proposed model, GotenNet, consistently outperforms state-of-the-art methods across all evaluated molecules, demonstrating superior performance in both energy and force predictions.

For molecules such as Tetrapeptide and AT-AT, GotenNet achieves notable reductions in energy errors, with improvements of 18.6% and 29.5% over the previous best models, respectively. Simultaneously, force prediction errors are reduced by up to 30.4%, underscoring GotenNet's balanced performance across both metrics. In more complex cases, such as the Buckyball catcher and Double-walled nanotube, GotenNet sets new benchmarks, reducing energy errors by over 35% and force errors by up to 31.3%. These results highlight the robustness and versatility of GotenNet in handling diverse molecular structures, establishing it as a leading model in both energy and force prediction.

## 4.4 rMD17 dataset

**Dataset.** The rMD17 dataset (Christensen & Von Lilienfeld, 2020) is a revised version of the MD17 benchmark, featuring 10 small organic molecules with 100,000 conformations per molecule. It serves as a key benchmark for evaluating machine learning models' ability to predict molecular energies and forces across diverse conformations. We follow the standard split (Christensen & Von Lilienfeld, 2020) of 950 training, 50 validation, and the remaining conformations for testing. The results are averaged over five predefined splits to ensure robust evaluation.

Table 4: The table presents MAE for energy (kcal/mol) and forces (kcal/mol/Å) on the rMD17 dataset.

| Molecule | | NequIP | ACE | UNiTE | Allegro | BOTNet | MACE | TensorNet | GotenNet |
|---|---|---|---|---|---|---|---|---|---|
| Aspirin | E | 0.0530 | 0.1407 | 0.0553 | 0.0530 | 0.0530 | 0.0507 | 0.0553 | **0.0358** |
| | F | 0.1891 | 0.4128 | 0.1753 | 0.1683 | 0.1960 | 0.1522 | 0.2052 | **0.1306** |
| Azobenzene | E | 0.0161 | 0.0830 | 0.0254 | 0.0277 | 0.0161 | 0.0277 | 0.0161 | **0.0121** |
| | F | 0.0669 | 0.2514 | 0.0969 | 0.0600 | 0.0761 | 0.0692 | 0.0715 | **0.0483** |
| Benzene | E | 0.0009 | 0.0009 | 0.0016 | 0.0069 | 0.0007 | 0.0092 | **0.0005** | **0.0005** |
| | F | 0.0069 | 0.0115 | 0.0168 | **0.0046** | 0.0069 | 0.0069 | 0.0069 | 0.0047 |
| Ethanol | E | 0.0092 | 0.0277 | 0.0143 | 0.0092 | 0.0092 | 0.0092 | 0.0115 | **0.0071** |
| | F | 0.0646 | 0.1683 | 0.0853 | 0.0484 | 0.0738 | 0.0484 | 0.0807 | **0.0482** |
| Malonaldehyde | E | 0.0184 | 0.0392 | 0.0254 | 0.0138 | 0.0184 | 0.0184 | 0.0184 | **0.0129** |
| | F | 0.1176 | 0.2560 | 0.1522 | **0.0830** | 0.1338 | 0.0945 | 0.1245 | **0.0830** |
| Naphthalene | E | 0.0208 | 0.0208 | 0.0106 | 0.0046 | 0.0046 | 0.0115 | 0.0046 | **0.0039** |
| | F | 0.0300 | 0.1176 | 0.0600 | **0.0208** | 0.0415 | 0.0369 | 0.0369 | 0.0240 |
| Paracetamol | E | 0.0323 | 0.0922 | 0.0438 | 0.0346 | 0.0300 | 0.0300 | 0.0300 | **0.0212** |
| | F | 0.1361 | 0.2929 | 0.1637 | 0.1130 | 0.1338 | 0.1107 | 0.1361 | **0.0928** |
| Salicylic acid | E | 0.0161 | 0.0415 | 0.0168 | 0.0208 | 0.0184 | 0.0208 | 0.0184 | **0.0141** |
| | F | 0.0922 | 0.2145 | 0.0876 | **0.0669** | 0.0992 | 0.0715 | 0.1061 | 0.0703 |
| Toluene | E | 0.0069 | 0.0254 | 0.0104 | 0.0092 | 0.0092 | 0.0115 | 0.0069 | **0.0044** |
| | F | 0.0369 | 0.1499 | 0.0577 | 0.0415 | 0.0438 | 0.0346 | 0.0392 | **0.0261** |
| Uracil | E | 0.0092 | 0.0254 | 0.0134 | 0.0138 | 0.0092 | 0.0115 | 0.0092 | **0.0064** |
| | F | 0.0715 | 0.1522 | 0.0876 | **0.0415** | 0.0738 | 0.0484 | 0.0715 | 0.0417 |

**Results.** As shown in Table 4, GotenNet outperforms other models in 80% of tasks and ranks second in the remaining, excelling in both energy and force predictions. GotenNet sets new benchmarks for molecules such as Aspirin, Azobenzene, Ethanol, Paracetamol, and Toluene, demonstrating balanced improvements across energy and force predictions. These results highlight GotenNet's robustness and its ability to accurately model molecular properties, outperforming prior methods on rMD17 dataset.

## 4.5 Ablation Study

Table 5 presents the results of the ablation study, highlighting the impact of various components on the performance of GotenNet. The inclusion of structural embedding (SE), self-attention (SEA), geometric encoding (GE), and HTR generally leads to improved results, as shown in rows 1, 7, and 8, where the model achieves the lowest std MAE and log MAE. The removal of any one of these components results in a significant degradation in performance, particularly

Table 5: Ablation study on QM9 dataset.

| #L | $L_{max}$ | SE | SEA | GE | HTR | std | log |
|---|---|---|---|---|---|---|---|
| 4 | 2 | ✓ | ✓ | ✓ | ✓ | 0.67 | -6.21 |
| 6 | 1 | ✓ | ✓ | ✓ | ✓ | 0.68 | -6.17 |
| 6 | 2 | ✗ | ✓ | ✓ | ✓ | 0.67 | -6.17 |
| 6 | 2 | ✓ | ✗ | ✓ | ✓ | 0.65 | -6.23 |
| 6 | 2 | ✓ | ✓ | ✗ | ✓ | 0.83 | -5.96 |
| 6 | 2 | ✓ | ✓ | ✓ | ✗ | 0.64 | -6.20 |
| 6 | 2 | ✓ | ✓ | ✓ | ✓ | 0.61 | -6.26 |
| 12 | 2 | ✓ | ✓ | ✓ | ✓ | 0.56 | -6.34 |

in the cases without geometric encoding (row 4) or reducing $L_{max}$ (row 2). The full model with 12 layers (row 8) achieves the best performance, with the lowest std MAE of 0.56 and log MAE of -6.34. This demonstrates the combined effectiveness of all components for model scalability.

## 5 Conclusion

We presented GotenNet, a framework for modeling 3D molecular structures that strikes a balance between expressiveness and efficiency by integrating geometric tensor representations with innovative components, including unified structure embedding, geometry-aware tensor attention, and hierarchical tensor refinement. GotenNet consistently outperforms state-of-the-art methods across four benchmark datasets. It also demonstrates scalability and computational efficiency, making it highly suitable for large-scale molecular systems. These results establish GotenNet as a versatile and powerful framework for 3D equivariant graph neural networks. Future work could further enhance its scalability to larger molecular systems and explore applications in molecular dynamics and materials science.

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

APPENDIX

## A    PROOF: EQUIVARIANCE OF GEOMETRIC TENSOR

**Equivariance of Geometric Tensor:** Let $T : M \to \mathcal{T}$ be a geometric tensor field, and $g \in E(3)$ be an element of the Euclidean group. Then $T$ is equivariant under $g$ if:

$$(g \cdot T)(p) = \rho(g)T(g^{-1} \cdot p)$$

where $\rho$ is a representation of $E(3)$ on the space of tensors $\mathcal{T}$.

*Proof.* To prove the equivariance of the geometric tensor field, we start by recalling the definition of equivariance. A tensor field $T : M \to \mathcal{T}$ is equivariant under the group action of $g \in E(3)$ if, for all points $p \in M$,

$$T(g \cdot p) = \rho(g)T(p),$$

where $\rho(g)$ is a representation of the group element $g$ on the space $\mathcal{T}$.

Now, consider the action of the group element $g \in E(3)$ on $T$. By the definition of the group action of $g$ on the tensor field $T$, we have:

$$(g \cdot T)(p) = T(g^{-1} \cdot p).$$

Next, we apply the representation $\rho(g)$ to the transformed tensor $T(g^{-1} \cdot p)$. By the equivariance condition, we require that:

$$(g \cdot T)(p) = \rho(g)T(g^{-1} \cdot p).$$

This completes the proof, as we have shown that the transformed tensor field $g \cdot T$ is related to the original tensor field $T$ by the representation $\rho(g)$, satisfying the equivariance condition.  $\square$

## B    PROOF: GOTENNET PRESERVE THE EQUIVARIANCE PROPERTY

*Proof.* We will prove that layer operations in GotenNet preserve the equivariance property of geometric tensor fields under the action of the Euclidean group $E(3)$. To this end we will prove that each component used in GotenNet preserves the invariance/equivariance properties.

First we will prove that the initial node and edge representations are invariant under the action of the Euclidean group $E(3)$. We'll consider each component separately. The node embedding process consists of two main steps: message passing and representation update. We'll show that both steps are invariant under $E(3)$. The message passing equation in node embedding is:

$$\mathbf{m}_i = \sum_{j \in \mathcal{N}(i)} \boldsymbol{z}_j \mathbf{A}_{\text{nbr}} \circ \left( \varphi(\tilde{\boldsymbol{r}}^{(0)}{}_{ij}) \mathbf{W}_{\text{ndp}} \circ \phi(\tilde{\boldsymbol{r}}^{(0)}{}_{ij}) \right).$$

Under the action of $g \in E(3)$ $\boldsymbol{z}_j$ is invariant as it's an atomic number. $\tilde{r}_{ij,\text{init}}$ is invariant as it's the distance between nodes $i$ and $j$. $\mathcal{N}(i)$ is invariant as the set of neighbors doesn't change under rigid transformations. Therefore, $\mathbf{m}_i$ is invariant under $E(3)$. The node representation update is defined as:

$$\boldsymbol{h}_{i,\text{init}} = \left( \sigma\left( \text{LN}((\boldsymbol{z}_i \mathbf{A}_{\text{na}}, \mathbf{m}_i) \mathbf{W}_{\text{nrd}}) \right) \right) \mathbf{W}_{\text{nru}}.$$

Here, $\boldsymbol{z}_i$ is invariant, $\mathbf{m}_i$ is invariant (as shown above), and all other operations (concatenation, linear transformations, layer normalization, and activation) are invariant. Thus, $\boldsymbol{h}_i$ is invariant under $E(3)$. The node embedding is followed by edge embedding and the initial edge representation is computed as:

$$\boldsymbol{t}_{ij,\text{init}} = (\boldsymbol{h}_{i,\text{init}} + \boldsymbol{h}_{j,\text{init}}) \circ \left( \varphi(\tilde{\boldsymbol{r}}^{(0)}{}_{ij}) \mathbf{W}_{\text{erp}} \right).$$

We've already shown that $h_{\text{init}}$ is invariant. The distance $r_{ij,\text{init}}$ is also invariant under $E(3)$. The operations $\varphi$, and linear transformations are all invariant. Therefore, $\boldsymbol{t}_{ij,\text{init}}$ is invariant under $E(3)$. Thus, we have shown that both the initial node representations $h_{i,\text{init}}$ and edge representations $t_{ij,\text{init}}$ are invariant under the action of the Euclidean group $E(3)$.

After the initialization step the next module is GATA. Therefore, we will prove that the GATA module and the subsequent operations in the interaction layer preserve equivariance under the action of the Euclidean group $E(3)$. The query, key, and value Computation as defined in Equation (5), we have:

$$\boldsymbol{q}_i = \boldsymbol{h}_i\mathbf{W}_q, \quad \boldsymbol{k}_j = \boldsymbol{h}_j\mathbf{W}_k, \quad \boldsymbol{v}_j = \gamma_v(\boldsymbol{h}_j),$$

$\boldsymbol{h}_i$ and $\boldsymbol{h}_j$ are scalar (0-degree steerable features) representations, which are invariant under $E(3)$. The linear transformations $\mathbf{W}_q$ and $\mathbf{W}_k$, and the function $\gamma_v$, preserve this invariance. Thus, $\boldsymbol{q}_i$, $\boldsymbol{k}_j$, and $\boldsymbol{v}_j$ are invariant under $E(3)$. The attention coefficients are defined in Equation (6), the attention coefficient $\boldsymbol{\alpha}_{ij}$ is computed as:

$$\boldsymbol{\alpha}_{ij} = \boldsymbol{q}_i\big(\boldsymbol{k}_j \circ \sigma_k(\boldsymbol{t}_{ij}\mathbf{W}_{re})\big)^T.$$

We've shown that $\boldsymbol{q}_i$ and $\boldsymbol{k}_j$ are invariant. $\boldsymbol{t}_{ij}$ is an edge embedding, which is invariant under $E(3)$. The operations $\sigma_k$, $\circ$, and matrix multiplication preserve invariance. Therefore, $\boldsymbol{\alpha}_{ij}$ is invariant under $E(3)$. The self-attention operation in Equation (6) defined as:

$$\mathbf{sea}_{ij} = \frac{\exp(\boldsymbol{\alpha}_{ij})}{\sum_{k \in \mathcal{N}(i)} \exp(\boldsymbol{\alpha}_{ik})}\boldsymbol{v}_j.$$

Since $\boldsymbol{\alpha}_{ij}$ and $\boldsymbol{v}_j$ are invariant, and the softmax operation preserves invariance, $\mathbf{sea}_{ij}$ is invariant under $E(3)$. Geometric Encoding defined in Equation (7), we have:

$$\boldsymbol{o}_{ij}^s, \{\boldsymbol{o}_{ij}^{d,(l)}\}_{l=1}^{L_{\max}}, \{\boldsymbol{o}_{ij}^{t,(l)}\}_{l=1}^{L_{\max}} = \mathrm{split}(\mathbf{sea}_{ij} + (\boldsymbol{t}_{ij}\mathbf{W}_{rs}) \circ \gamma_s(\boldsymbol{h}_j) \circ \phi(\tilde{\boldsymbol{r}}_{ij}^{(0)}), d_{ne}).$$

$\mathbf{sea}_{ij}$, $\boldsymbol{t}_{ij}$, $\boldsymbol{h}_j$, and $\tilde{\boldsymbol{r}}_{ij}^{(0)}$ are all invariant under $E(3)$. The operations $\mathbf{W}_{rs}$, $\gamma_s$, $\phi$, $\circ$, and split preserve this invariance. Therefore, $\boldsymbol{o}_{ij}^s$, $\{\boldsymbol{o}_{ij}^{d,(l)}\}_{l=1}^{L_{\max}}$ and $\{\boldsymbol{o}_{ij}^{t,(l)}\}_{l=1}^{L_{\max}}$ are invariant under $E(3)$. Finally, node tensor representation updated with Equation (8), we have:

$$\Delta\boldsymbol{h}_i = \bigoplus_{j \in \mathcal{N}(i)} (\boldsymbol{o}_{ij}^s), \quad \Delta\tilde{\boldsymbol{X}}_i^{(l)} = \bigoplus_{j \in \mathcal{N}(i)} \left(\boldsymbol{o}_{ij}^{d,(l)} \circ \tilde{\boldsymbol{r}}_{ij}^{(l)} + \boldsymbol{o}_{ij}^{t,(l)} \circ \tilde{\boldsymbol{X}}_j^{(l)}\right).$$

Here, $\boldsymbol{h}_i$, $\boldsymbol{o}_{ij}^s$, $\{\boldsymbol{o}_{ij}^{d,(l)}\}_{l=1}^{L_{\max}}$ and $\{\boldsymbol{o}_{ij}^{t,(l)}\}_{l=1}^{L_{\max}}$ are invariant under $E(3)$. Specifically, $\{\boldsymbol{o}_{ij}^{d,(l)}\}_{l=1}^{Lmax}$ and $\{\boldsymbol{o}_{ij}^{t,(l)}\}_{l=1}^{L_{\max}}$ are sets of invariant coefficients, where for each degree $l$ of steerable features, we have distinct invariant scalars $\boldsymbol{o}_{ij}^{d,(l)}$ and $\boldsymbol{o}_{ij}^{t,(l)}$. $\tilde{\boldsymbol{r}}_{ij}^{(l)}$ and $\tilde{\boldsymbol{X}}_j$ are high-degree steerable features that transform equivariantly under $E(3)$. The $\circ$ operation between invariant scalars and equivariant tensors preserves equivariance. Specifically, since $\circ$ multiplies the same invariant values over the spatial dimension of the high-degree tensors, this operation preserves equivariance. To see this, let $g \in E(3)$ be a transformation, $s$ be an invariant scalar, and $T$ be an equivariant tensor. Then:

$$g(s \circ T) = g(sT) = sg(T) = s \circ g(T).$$

This shows that the $\circ$ operation commutes with the group action, preserving equivariance. The permutation-invariant aggregation function $\bigoplus$ (such as summation or averaging) preserves equivariance because it operates independently on each degree of the tensor, maintaining their transformation properties under $E(3)$. Therefore, the scalar $\Delta\boldsymbol{h}_i$ remains invariant, while the high-degree steerable features $\tilde{\boldsymbol{X}}^{(l)}$ transform equivariantly under $E(3)$. Thus, we have shown that the interaction layer preserves the equivariance of the input representations under the action of the Euclidean group $E(3)$. Thus, we have shown that the interaction layer preserves the equivariance of the input representations under the action of the Euclidean group $E(3)$.

Next, we will prove that the HTR component preserves equivariance under the action of the Euclidean group $E(3)$. We'll consider each operation in the HTR component. Tensor projections as defined in Equation (10), we have:

$$\widetilde{\boldsymbol{EQ}}_i^{(l)} = \tilde{\boldsymbol{X}}_i^{(l)}\mathbf{W}_{vq}, \quad \widetilde{\boldsymbol{EK}}_j^{(l)} = \tilde{\boldsymbol{X}}_j^{(l)}\mathbf{W}_{vk}^{(l)}, \quad \text{for } l \in \{1, \ldots, L_{\max}\},$$

$\tilde{\boldsymbol{X}}_i^{(l)}$ and $\tilde{\boldsymbol{X}}_j^{(l)}$ are high-degree tensors that transform equivariantly under $E(3)$. The projection matrices $\mathbf{W}_{vq}$ and $\mathbf{W}_{vk}^{(l)}$ apply uniform weights across the spatial dimensions for each representation

dimension. This uniform application preserves equivariance because it commutes with the action of $E(3)$. Let $g \in E(3)$ be a transformation:

$$g(\tilde{\boldsymbol{X}}_i^{(l)} \mathbf{W}_{vq}) = (g(\tilde{\boldsymbol{X}}_i^{(l)}))(\mathbf{W}_{vq}) = (g(\tilde{\boldsymbol{X}}_i^{(l)} \mathbf{W}_{vq})).$$

The same holds for $\widetilde{\boldsymbol{EK}}_j^{(l)}$. Therefore, $\widetilde{\boldsymbol{EQ}}_i^{(l)}$ and $\widetilde{\boldsymbol{EK}}_j^{(l)}$ transform equivariantly under $E(3)$. Aggregation of angular and magnitude information from Equation (11), we have:

$$\boldsymbol{w}_{ij} = \text{Agg}_{l=1}^{L_{\max}}\left((\widetilde{\boldsymbol{EQ}}_i^{(l)})^\top \widetilde{\boldsymbol{EK}}_j^{(l)}\right).$$

The operation $(\widetilde{\boldsymbol{EQ}}_i^{(l)})^\top \widetilde{\boldsymbol{EK}}_j^{(l)}$ is an inner product between equivariant tensors. Crucially, both $\widetilde{\boldsymbol{EQ}}_i^{(l)}$ and $\widetilde{\boldsymbol{EK}}_j^{(l)}$ are subject to the same global rotation and translation under the action of $E(3)$. Let $g \in E(3)$ be a transformation. Then:

$$\begin{aligned}
(g(\widetilde{\boldsymbol{EQ}}_i^{(l)}))^\top g(\widetilde{\boldsymbol{EK}}_j^{(l)}) &= (\boldsymbol{D}^{(l)}(r) \cdot \widetilde{\boldsymbol{EQ}}_i^{(l)})^\top (\boldsymbol{D}^{(l)}(r) \cdot \widetilde{\boldsymbol{EK}}_j^{(l)}) \\
&= (\widetilde{\boldsymbol{EQ}}_i^{(l)})^\top (\boldsymbol{D}^{(l)}(r))^\top \boldsymbol{D}^{(l)}(r) \cdot \widetilde{\boldsymbol{EK}}_j^{(l)} \\
&= (\widetilde{\boldsymbol{EQ}}_i^{(l)})^\top \widetilde{\boldsymbol{EK}}_j^{(l)} \cdot \eta^l.
\end{aligned}$$

Here, $\boldsymbol{D}^{(l)}(r)$ represents the Wigner D-matrix of degree $l$, which is the appropriate representation for the transformation of spherical tensors under rotations. The matrix $\boldsymbol{D}^{(l)}(r)$ is unitary, meaning $(D^{(l)}(r))^\top \boldsymbol{D}^{(l)}(r) = I$. The factor $\eta^l = \pm 1$ accounts for the parity transformation, where $\eta = -1$ for improper rotations and $\eta = 1$ for proper rotations, with the exponent $l$ determining the overall sign based on the spherical harmonic degree. This demonstrates that the inner product transforms covariantly under the full $O(3)$ symmetry group, with the parity factor properly accounting for improper rotations. The aggregation $\bigoplus$ over these covariant scalars preserves the transformation properties, ensuring that $w_{ij}$ transforms appropriately under both rotations and inversions. The edge representation update is given by:

$$\boldsymbol{t}_{ij} \leftarrow \boldsymbol{t}_{ij} + \Delta\boldsymbol{t}_{ij}, \quad \Delta\boldsymbol{t}_{ij} = \gamma_w(\boldsymbol{w}_{ij}) \circ \gamma_t(\boldsymbol{t}_{ij}),$$

$w_{ij}$ is invariant, as shown in the previous step, and $\boldsymbol{t}_{ij}$ is an edge embedding, which is also invariant under $E(3)$. Since both $\boldsymbol{w}_{ij}$ and $\boldsymbol{t}_{ij}$ are invariant, the application of the differentiable functions $\gamma_w$ and $\gamma_t$ preserves this invariance. Moreover, the Hadamard product of invariant quantities is itself invariant. Hence, we have

$$g(\Delta\boldsymbol{t}_{ij}) = g(\gamma_w(\boldsymbol{w}_{ij}) \circ \gamma_t(\boldsymbol{t}_{ij})) = \gamma_w(\boldsymbol{w}_{ij}) \circ \gamma_t(\boldsymbol{t}_{ij}) = \Delta\boldsymbol{t}_{ij}$$

Thus, $\Delta\boldsymbol{t}_{ij}$ is invariant under $E(3)$. Thus, we have shown that the Hierarchical Tensor Refinement component preserves the equivariance of the input representations under the action of the Euclidean group $E(3)$. The high-degree tensors transform equivariantly, while the scalar quantities and edge embeddings remain invariant.

Finally, we will prove that the EQFF (Equivariant Feed-Forward) blocks preserve equivariance under the action of the Euclidean group $E(3)$. Consider each operation in the EQFF component as described in Equation (13). The input scalars $\boldsymbol{h}$ is invariant under $E(3)$ as it represents scalar features, while $\tilde{\boldsymbol{X}}$ transforms equivariantly under $E(3)$ as it represents high-degree steerable features. Next, we examine the computation of $\boldsymbol{m}_1$ and $\boldsymbol{m}_2$:

$$\boldsymbol{m}_1, \boldsymbol{m}_2 = \text{split}_2(\gamma_m(||\tilde{\boldsymbol{X}}^{(l)} \mathbf{W}_{vu}||_2, \boldsymbol{h})).$$

The operation $\tilde{\boldsymbol{X}}^{(l)} \mathbf{W}_{vu}$ preserves equivariance as it applies the same linear transformation across all spatial dimensions. The $L_2$ norm of this equivariant tensor field, $||\tilde{\boldsymbol{X}}^{(l)} \mathbf{W}_{vu}||_2$, is invariant under $E(3)$. To see this, let $g \in E(3)$ be a transformation. Then $||g(\tilde{\boldsymbol{X}}^{(l)} \mathbf{W}_{vu})||_2 = ||R \cdot (\tilde{\boldsymbol{X}}^{(l)} \mathbf{W}_{vu})||_2 = ||\tilde{\boldsymbol{X}}^{(l)} \mathbf{W}_{vu}||_2$, where $R$ is the rotation matrix corresponding to $g$. The translation component doesn't affect the norm. The concatenation $(||\tilde{\boldsymbol{X}}^{(l)} \mathbf{W}_{vu}||_2, \boldsymbol{h})$ is of two invariant quantities, resulting in an invariant vector. As $\gamma_m$ is applied to an invariant input, its output is also invariant. Finally, splitting this invariant vector results in invariant components $\boldsymbol{m}_1$ and $\boldsymbol{m}_2$. Now, we analyze the EQFF operation:

$$\text{EQFF}(\boldsymbol{h}, \tilde{\boldsymbol{X}}^{(l)}) = \left((\boldsymbol{h} + \boldsymbol{m}_1), (\tilde{\boldsymbol{X}}^{(l)} + (\boldsymbol{m}_2 \circ \tilde{\boldsymbol{X}}^{(l)} \mathbf{W}_{vu}))\right).$$

The term $\boldsymbol{h} + \boldsymbol{m}_1$ is a sum of two invariant quantities, resulting in an invariant scalar. The operation $\boldsymbol{m}_2 \circ \tilde{\boldsymbol{X}}^{(l)}\mathbf{W}_{vu}$ preserves equivariance because $\boldsymbol{m}_2$ is an invariant scalar and $\tilde{\boldsymbol{X}}^{(l)}\mathbf{W}_{vu}$ is equivariant. The element-wise product of an invariant scalar with an equivariant tensor is equivariant. Consequently, $\tilde{\boldsymbol{X}}^{(l)} + (\boldsymbol{m}_2 \circ \tilde{\boldsymbol{X}}^{(l)}\mathbf{W}_{vu})$ is a sum of two equivariant tensors, resulting in an equivariant tensor. Hence, EQFF returns tuple of updated representations for scalar and high-degree steerable features.

Thus, we have shown that the EQFF operation preserves the equivariance of the input representations under the action of the Euclidean group $E(3)$. The scalar part remains invariant, while the high-degree steerable features transform equivariantly.

Hence, we have shown that each major component of the GotenNet architecture preserves the equivariance property under the action of the Euclidean group $E(3)$. The initial node and edge embeddings are invariant under $E(3)$. The GATA module, including its self-attention mechanism and geometric encoding, maintains the invariance of scalar quantities and the equivariance of high-degree steerable tensors. The Hierarchical Tensor Refinement (HTR) component preserves equivariance in its tensor projections and ensures that edge updates remain invariant. Finally, the EQFF blocks maintain the overall equivariance structure by preserving the invariance of scalar parts and the equivariance of high-degree steerable features. In each of these components, we have demonstrated how the various operations interact with the group action of $E(3)$ to preserve the required invariance and equivariance properties.

$\square$

## C  PROOF: EQUIVARIANCE OF GOTENNET

**Equivariance of GotenNet:** If all layers of GotenNet are equivariant, then the entire network is equivariant.

*Proof.* We will prove this by induction on the number of layers in the network.

**Base Case (Layer 1):** Let $T^1 : M \to \mathcal{T}^1$ represent the output of the first layer of the network. Since the first layer is equivariant by assumption, we have that for all $g \in E(3)$,

$$(g \cdot T^1)(p) = \rho(g)T^1(g^{-1} \cdot p).$$

Thus, the first layer preserves equivariance.

**Inductive Step:** Assume that the output of layer $l$, denoted by $T^l$, is equivariant. That is, for all $g \in E(3)$,

$$(g \cdot T^l)(p) = \rho(g)T^l(g^{-1} \cdot p).$$

We need to show that layer $l + 1$, denoted by $T^{l+1}$, is also equivariant. Let $\Phi^{l+1}$ represent the operation of the $(l + 1)$-th layer. Then $T^{l+1} = \Phi^{l+1} \circ T^l$. Since the $(l + 1)$-th layer is also assumed to be equivariant, we have:

$$(g \cdot T^{l+1})(p) = \rho(g)T^{l+1}(g^{-1} \cdot p).$$

The equivariance of $T^{l+1}$ follows from the equivariance of $T^l$ and $\Phi^{l+1}$, as the composition of equivariant functions is also equivariant. Explicitly:

$$\begin{aligned}
(g \cdot T^{l+1})(p) &= (g \cdot (\Phi^{l+1} \circ T^l))(p) \\
&= \Phi^{l+1}((g \cdot T^l)(p)) \\
&= \Phi^{l+1}(\rho(g)T^l(g^{-1} \cdot p)) \\
&= \rho(g)\Phi^{l+1}(T^l(g^{-1} \cdot p)) \\
&= \rho(g)T^{l+1}(g^{-1} \cdot p)
\end{aligned}$$

Hence, by the principle of mathematical induction, we have shown that if all individual layers in the GotenNet are equivariant, then for any number of layers, the final output of the network, being the composition of these equivariant layers, is also equivariant. This result relies on the property

that the composition of equivariant functions is itself equivariant. Specifically, we have shown that $T^{l+1} = \Phi^{l+1} \circ T^l$, and the equivariance of both $\Phi^{l+1}$ and $T^l$ ensures the equivariance of $T^{l+1}$. Hence, the entire network is equivariant. This theorem, combined with the previous proof of component-wise equivariance (see Appendix B), establishes the overall equivariance of the GotenNet architecture, ensuring its consistency under Euclidean transformations of the input space. □

## D    REPRODUCIBILITY STATEMENT

The details on components of the architecture, hyper-parameters, and model variations are outlined in Section E. The code used to reproduce the experiments is available. All datasets used in this study are publicly available; the access instructions will be included in the source code. We have included information on the computational resources used for our experiments, including hardware specifications and software versions, to facilitate reproducibility of our results.

Table 6: Hyper-parameters for the datasets GotenNet compared against the baselines. The parameters are for GotenNet$_B$ if multiple variations exists.

| Hyper-parameters | QM9 | Molecule3D | MD22 | rMD17 |
|---|---|---|---|---|
| Optimizer | AdamW | AdamW | AdamW | AdamW |
| Learning rate scheduling | Linear warmup with reduce on plateau | | | |
| Warmup steps | 10,000 | 5,000 | 1,000 | 1,000 |
| Maximum learning rate | [6e−5, 1e−4] | 1e−4 | [4e−5, 1e−4] | 2e−4 |
| Learning rate decay | 0.8 | 0.8 | 0.8 | 0.8 |
| Learning rate patience | 15 | 5 | 30 | 30 |
| Loss function | MSE | $L_1$ | MSE | MSE |
| Gradient clipping | 10 | - | 5 | 10 |
| Batch size | 32 | 256 | 4 | 4 |
| Number of epochs | 1,000 | 300 | 3,000 | 3,000 |
| Weight decay | 0.01 | 0.01 | 0.01 | 0.01 |
| Dropout rate | 0.1 | 0.1 | 0.1 | 0.1 |
| Node dimension ($d_{ne}$) | 256 | 384 | [256, 384] | 192 |
| Edge dimension ($d_{ed}$) | 256 | 384 | [256, 384] | 192 |
| Edge refinement dimension ($d_{xpd}$) | 256 | 384 | 768 | 768 |
| $L_{\max}$ | 2 | 2 | 2 | 2 |
| Number of Layers | 6 | 12 | [6, 8] | 12 |
| Number of RBFs | 64 | 32 | 32 | 32 |
| Number of Attention Heads | 8 | 8 | 8 | 8 |
| Cutoff radius | 5.0 | 5.0 | [4.0, 5.0] | 5.0 |

## E    TRAINING DETAILS AND HYPER-PARAMETERS

Table 6 presents the comprehensive set of hyper-parameters employed in our experiments across various datasets. These parameters were carefully selected to optimize model performance and ensure fair comparisons with baseline methods. For the GotenNet architecture, we primarily report the parameters for the base (GotenNet$_B$) variation where multiple model sizes exist. The optimization process utilized the AdamW optimizer across all datasets, coupled with a linear warmup strategy and learning rate reduction on plateau. This adaptive learning rate approach allows for more stable training and improved convergence. Furthermore, spherical harmonics were computed using the e3nn (Geiger & Smidt, 2022) library.

The learning rates were fine-tuned for each dataset, with QM9 and MD22 employing a range of maximum learning rates to account for the diverse nature of their target properties. Molecule3D and rMD17 datasets, on the other hand, used fixed maximum learning rates of 1e−4 and 2e−4, respectively. To mitigate overfitting and promote generalization, we implemented weight decay (0.01) and dropout (0.1) consistently across all datasets. The choice of loss function varied, with mean squared error (MSE) being the predominant choice, except for Molecule3D, which utilized the L1 loss.

Table 7: Performance comparisons of GotenNet variations on QM9 dataset.

| Task
Units | $\alpha$
$ma_0^3$ | $\Delta\varepsilon$
meV | $\varepsilon_{\text{HOMO}}$
meV | $\varepsilon_{\text{LUMO}}$
meV | $\mu$
mD | $C_\nu$
$\frac{\text{mcal}}{\text{mol K}}$ | $G$
meV | $H$
meV | $R^2$
$ma_0^2$ | $U$
meV | $U_0$
meV | ZPVE
meV | std.
% | log
- |
|---|---|---|---|---|---|---|---|---|---|---|---|---|---|---|
| GotenNet$_{\hat{S}}$ | 37 | 25.4 | 18.4 | 15.7 | 7.5 | 21 | 5.67 | 4.17 | 33 | 3.97 | 3.89 | 1.16 | 0.67 | -6.21 |
| GotenNet$_{S}$ | 33 | 21.2 | 16.9 | 13.9 | 7.5 | 20 | 5.50 | 3.70 | 29 | 3.67 | 3.71 | 1.09 | 0.60 | -6.29 |
| % Improvement | 10.8 | 16.5 | 8.2 | 11.5 | 0.0 | 4.8 | 3.0 | 11.3 | 12.1 | 7.6 | 4.6 | 6.0 | 10.5 | 1.3 |
| GotenNet$_{\hat{B}}$ | 33 | 23.0 | 16.4 | 14.4 | 7.8 | 20 | 5.42 | 3.74 | 32 | 3.76 | 3.76 | 1.10 | 0.61 | -6.26 |
| GotenNet$_{B}$ | 32 | 20.5 | 15.2 | 13.0 | 7.2 | 19 | 5.19 | 3.44 | 27 | 3.49 | 3.43 | 1.09 | 0.56 | -6.35 |
| % Improvement | 3.0 | 10.9 | 7.3 | 9.7 | 7.7 | 5.0 | 4.2 | 8.0 | 15.6 | 7.2 | 8.8 | 0.9 | 8.2 | 1.4 |
| GotenNet$_{\hat{L}}$ | 30 | 20.7 | 14.3 | 13.3 | 7.7 | 19 | 5.27 | 3.47 | 25 | 3.58 | 3.67 | 1.08 | 0.56 | -6.34 |
| GotenNet$_{L}$ | 28 | 19.8 | 13.4 | 12.2 | 6.7 | 19 | 4.98 | 3.30 | 24 | 3.41 | 3.37 | 1.08 | 0.52 | -6.41 |
| % Improvement | 6.7 | 4.4 | 6.3 | 8.3 | 13.0 | 0.0 | 5.5 | 4.9 | 4.0 | 4.8 | 8.2 | 0.0 | 7.1 | 1.1 |

The model architecture parameters, such as $d_{ne}$, $d_{ed}$, and $d_{xpd}$, were adjusted based on the complexity of the dataset and the specific prediction tasks. For instance, MD22 and rMD17 datasets, which involve more complex molecular dynamics simulations, employed larger edge refinement dimensions (768) compared to QM9 and Molecule3D (256 and 384, respectively). The number of layers in the model also varied, with Molecule3D and rMD17 using deeper architectures (12 layers) compared to QM9 and MD22 (6 layers for the base model).

It is worth noting that for QM9, we experimented with different model sizes by varying the number of layers (4, 6, and 12 for S, B, and L variations, respectively). Similarly, for MD22, we explored a more compact model variation by reducing the number of layers to 4 and halving the representation dimensions. These variations allow us to investigate the trade-offs between model complexity and performance across different molecular property prediction tasks.

The consistent use of 8 attention heads and a $L_{\text{max}}$ of 2 across all datasets suggests that these parameters provide a good balance between computational efficiency and model expressiveness for a wide range of molecular modeling tasks. The cutoff radius, predominantly set at 5.0 Å (with some variations in MD22), was chosen to capture relevant atomic interactions while maintaining computational feasibility.

## F  ABLATION STUDY: SHARED AND INDIVIDUAL COEFFICIENTS

In this section, we investigate the impact of using shared coefficients for spherical harmonics as compared to learning individual coefficients for each degree. Table 7 presents a comprehensive performance comparison on the QM9 dataset for the two configurations. Variants denoted with a hat (e.g., GotenNet$_{\hat{S}}$, GotenNet$_{\hat{B}}$, GotenNet$_{\hat{L}}$) utilize shared coefficients across all degrees, whereas the corresponding non-hat variants (e.g., GotenNet$_{S}$, GotenNet$_{B}$, GotenNet$_{L}$) learn separate coefficients for each spherical harmonic degree.

The results indicate that the models with individual coefficients consistently yield lower error metrics across several tasks. For instance, the GotenNet$_{S}$ variant outperforms its shared-coefficient counterpart in terms of $\alpha$, $\Delta\varepsilon$, and molecular orbital energies, with similar trends observed for the base and large model configurations. This suggests that allowing independent coefficient learning provides greater expressiveness in capturing geometric nuances.

However, it is important to note that the increased flexibility of individual coefficients comes at the cost of a higher number of parameters. In resource-constrained settings, the shared coefficient approach may offer a favorable trade-off between model complexity and performance. Overall, these findings highlight the need to balance expressiveness and computational efficiency when selecting the appropriate configuration for a given application.

## G  EFFICIENCY EXPERIMENT DETAILS

The efficiency experiments are conducted by sampling graphs from the Molecule3D dataset. Using a real-world dataset provides realistic neighborhoods for the nodes computed with a radius graph, which is more beneficial than using a synthetic dataset. The graph sampling process involves selecting

Table 8: Computational complexity comparison of different methods. Training and inference latency are measured with batch size of 128 samples. Training time is reported in GPU days (min/avg/max/limit). Time per epoch is in seconds. Latency measurements are in milliseconds. Best results are in **bold**.

| Model | Batch Size | Time per Epoch (s) | Training Time (GPU days) | | | | Training Latency | Inference Latency | Trainable Parameters | Std. MAE | Log MAE |
|---|---|---|---|---|---|---|---|---|---|---|---|
| | | | min | avg | max | limit | | | | | |
| Equiformer | 128 | 425 | 1.48 | 1.48 | 1.48 | 1.48 | 421 | 150 | 3.5M | 0.70 | -5.82 |
| EquiformerV2 | 64 | 821 | 2.85 | 2.85 | 2.85 | 2.65 | 918 | 341 | 11.2M | 0.67 | -5.87 |
| EquiformerV2 | 48 | 847 | 2.94 | 2.94 | 2.94 | 2.65 | 918 | 341 | 11.2M | 0.67 | -5.87 |
| Geoformer | 32 | 436 | - | - | - | 5.05 | 759 | 264 | 50.6M | 0.75 | -6.12 |
| GotenNet$_{\hat{S}}$ | 32 | **117** | **0.41** | **0.75** | **1.34** | **1.35** | **80** | **37** | 6.1M | **0.67** | **-6.21** |
| GotenNet$_{\hat{B}}$ | 32 | 180 | 0.75 | 1.15 | 1.92 | 2.08 | 120 | 56 | 9.2M | **0.61** | **-6.26** |
| GotenNet$_{\hat{L}}$ | 32 | 291 | 1.37 | 1.87 | 2.33 | 3.37 | 244 | 112 | 18.3M | **0.56** | **-6.34** |

graphs with a predetermined node count. If no graph is available for a given node count, we sample the larger graphs with the closest node count and subsample the nodes to satisfy the node limitation. If the number of graphs is less than desired, we oversample the graphs for experimentation.

After creating a dataset for each node count, we conduct experiments by first warming up the models without timing for 5,000 steps. Then, we start timing the forward and backward passes for the subsequent 5,000 steps. The final values are obtained by averaging the timings over the total number of steps. It is important to note that the timings are measured per batch. Each batch consists of a fixed number of graphs, and the reported timings represent the average time taken to process a single batch during the training process.

This experimental setup ensures a fair comparison of the scalability properties of GotenNet and the baseline models, providing insights into their efficiency in handling graphs of varying sizes. By measuring the training time per batch, we can accurately assess how the computational overhead scales with increasing graph sizes, independent of the number of epochs or the total dataset size.

## H COMPUTATIONAL COMPLEXITY ANALYSIS

We present a comprehensive analysis of computational efficiency across state-of-the-art models, examining training requirements, inference speed, and model complexity. Here, we analyze the performance of GotenNet under a shared-coefficient paradigm for the spherical harmonic outputs $\{o_{ij}^d\}_{l=1}^{L_{\max}}$ and $\{o_{ij}^t\}_{l=1}^{L_{\max}}$ (denoted as GotenNet$_{(\cdot)}$ in Section F), highlighting its computational efficiency and effectiveness. Table 8 compares these metrics across different architectures under standardized conditions.

Our analysis reveals several key insights about computational efficiency across models. First, training protocols show significant variation across architectures - from Equiformer's relatively lightweight approach to Geoformer's more intensive requirements. Equiformer (Liao & Smidt, 2023) and EquiformerV2 (Liao et al., 2024) employ 300 epochs with batch sizes of 128 and 48/64 respectively, requiring 1.48 and 2.85/2.94 GPU days for completion. Geoformer (Wang et al., 2023a) utilizes a batch size of 32 for up to 600 epochs with early stopping, theoretically requiring 5.05 GPU days for full training. This variation reflects different trade-offs between computational demands and model expressiveness.

GotenNet demonstrates superior efficiency across multiple metrics. The smallest variant, GotenNet$_S$, achieves competitive performance with just 6.1M parameters while requiring minimal computational resources (0.75 GPU days average training time). This efficiency extends to both training and inference latencies, with GotenNet$_S$ achieving the lowest latencies in both categories (80ms and 37ms respectively).

Notably, even as model capacity increases, GotenNet maintains its efficiency advantages. GotenNet$_L$ (18.3M parameters) demonstrates remarkable scalability, requiring only 1.87 GPU days average training time while achieving 42% faster inference and 25% faster training compared to the closest competitor, Equiformer.

These results demonstrate that GotenNet's architectural innovations - particularly its efficient handling of geometric tensor representations - translate to practical advantages across all operational metrics. The consistent performance improvements across model scales suggest that GotenNet's approach to balancing expressiveness and efficiency is fundamentally sound, making it particularly suitable for real-world applications where computational resources are constrained.

## I  COMPREHENSIVE PERFORMANCE COMPARISON ON QM9 DATASET

Table 9 provides a comprehensive comparison of the performance of various baseline models against our proposed GotenNet on the QM9 dataset. This extended table includes a full list of baseline methods, offering a detailed assessment across all molecular property prediction tasks. The results showcase how GotenNet consistently surpasses existing models in both energy and force predictions, further highlighting its robustness and scalability. By including a wider range of baseline comparisons in this appendix, we aim to give a clearer view of GotenNet's advantages in different metrics and provide a more exhaustive evaluation for the QM9 dataset.

Table 9: Performance comparisons on QM9 dataset. † denotes using different data partitions.

| Task | $\alpha$ | $\Delta\varepsilon$ | $\varepsilon_{\text{HOMO}}$ | $\varepsilon_{\text{LUMO}}$ | $\mu$ | $C_\nu$ | $G$ | $H$ | $R^2$ | $U$ | $U_0$ | ZPVE | std. | log |
|---|---|---|---|---|---|---|---|---|---|---|---|---|---|---|
| Units | $ma_0^3$ | meV | meV | meV | mD | $\frac{\text{mcal}}{\text{mol K}}$ | meV | meV | $ma_0^2$ | meV | meV | meV | % | - |
| **Invariant models** | | | | | | | | | | | | | | |
| Cormorant | 85 | 61 | 34 | 38 | 38 | 26 | 20 | 21 | 961 | 21 | 22 | 2.03 | 2.14 | -4.75 |
| NMP | 92 | 69 | 43 | 38 | 30 | 40 | 19 | 17 | 180 | 20 | 20 | 1.50 | 1.78 | -5.08 |
| DimeNet++† | 44 | 32.6 | 24.6 | 19.5 | 29.7 | 23 | 7.56 | 6.53 | 331 | 6.28 | 6.32 | 1.21 | 0.98 | -5.67 |
| ComENet† | 45 | 32.4 | 23.1 | 19.8 | 24.5 | 22 | 7.98 | 6.86 | 259 | 6.82 | 6.69 | 1.20 | 0.93 | -5.69 |
| SphereNet† | 46 | 31.1 | 22.8 | 18.9 | 24.5 | 22 | 7.78 | 6.33 | 268 | 6.36 | 6.26 | 1.12 | 0.91 | -5.73 |
| **Scalarization-based models** | | | | | | | | | | | | | | |
| ClofNet | 63 | 53 | 33 | 25 | 40 | 27 | 9 | 9 | 610 | 9 | 8 | 1.23 | 1.37 | -5.37 |
| EGNN | 71 | 48 | 29 | 25 | 29 | 31 | 12 | 12 | 106 | 12 | 11 | 1.55 | 1.23 | -5.43 |
| PaiNN† | 45 | 45.7 | 27.6 | 20.4 | 12.0 | 24 | 7.35 | 5.98 | 66 | 5.83 | 5.85 | 1.28 | 1.01 | -5.85 |
| LEFTNet | 48 | 40 | 24 | 18 | 12 | 23 | 7 | 6 | 109 | 7 | 6 | 1.33 | 0.91 | -5.82 |
| EQGAT | 53 | 32 | 20 | 16 | 11 | 24 | 23 | 24 | 382 | 25 | 25 | 2.00 | 0.86 | -5.28 |
| ET | 59 | 36.1 | 20.3 | 17.5 | 11 | 26 | 7.62 | 6.16 | 33 | 6.38 | 6.15 | 1.84 | 0.84 | -5.90 |
| Geoformer | 40 | 33.8 | 18.4 | 15.4 | 10 | 22 | 6.13 | 4.39 | 28 | 4.41 | 4.43 | 1.28 | 0.75 | -6.12 |
| SaVeNet-B† | 39 | 24.8 | 18.4 | 16.3 | 9.3 | 23 | 6.64 | 5.43 | 58 | 5.48 | 5.43 | 1.18 | 0.69 | -6.04 |
| **High-degree steerable models** | | | | | | | | | | | | | | |
| SEGNN | 60 | 42 | 24 | 21 | 23 | 31 | 15 | 16 | 660 | 13 | 15 | 1.62 | 1.08 | -5.27 |
| HDGNN† | 46 | 32 | 18 | 16 | 17 | 23 | 11 | 10 | 342 | 8.12 | 8.34 | 1.21 | 0.80 | -5.64 |
| Equiformer | 46 | 30 | 15.4 | 14.7 | 12 | 23 | 7.63 | 6.63 | 251 | 6.74 | 6.59 | 1.26 | 0.70 | -5.82 |
| Equiformer$_{V2}$ | 47 | 29.0 | 14.4 | 13.3 | 9.9 | 23 | 7.57 | 6.22 | 186 | 6.49 | 6.17 | 1.47 | 0.67 | -5.87 |
| **Pre-trained models** | | | | | | | | | | | | | | |
| SE(3)-DDM | 46 | 40.2 | 23.5 | 19.5 | 15 | 24 | 7.65 | 7.09 | 122 | 6.99 | 6.92 | 1.31 | 0.93 | -5.76 |
| 3D-EMGP | 57 | 37.1 | 21.3 | 18.2 | 20 | 26 | 9.30 | 8.70 | 92 | 8.60 | 8.60 | 1.38 | 0.92 | -5.68 |
| Transformer-M | 41 | 27.4 | 17.5 | 16.2 | 37 | 22 | 9.63 | 9.39 | 75 | 9.41 | 9.37 | 1.18 | 0.86 | -5.74 |
| Coord | 52 | 31.8 | 17.7 | 14.3 | 12 | 20 | 6.91 | 6.45 | 450 | 6.11 | 6.57 | 1.71 | 0.76 | -5.75 |
| Frad(VRN) | 42 | 27.7 | 17.9 | 13.8 | 11 | 21 | 6.03 | 6.01 | 354 | 5.35 | 5.41 | 1.63 | 0.71 | -5.85 |
| Frad(RN) | 37 | 27.8 | 15.3 | 13.7 | 10 | 20 | 6.19 | 5.55 | 342 | 5.62 | 5.33 | 1.42 | 0.66 | -5.91 |
| DenoiseVAE | 65 | 26.0 | 14.2 | 11.9 | 7.9 | 15 | 5.35 | 4.19 | 62 | 4.03 | 4.31 | 1.03 | 0.61 | -6.18 |
| SliDe | 37 | 26.2 | 13.6 | 12.3 | 8.7 | 19 | 5.37 | 4.26 | 341 | 4.29 | 4.28 | 1.52 | 0.60 | -6.02 |
| **Spherical-scalarization models** | | | | | | | | | | | | | | |
| GotenNet$_S$ | 33 | 21.2 | 16.9 | 13.9 | 7.5 | 20 | 5.50 | 3.70 | 29 | 3.67 | 3.71 | 1.09 | 0.60 | -6.29 |
| GotenNet$_B$ | 32 | 20.5 | 15.2 | 13.0 | 7.2 | 19 | 5.19 | 3.44 | 27 | 3.49 | 3.43 | 1.09 | 0.56 | -6.35 |
| GotenNet$_L$ | **28** | **19.8** | **13.4** | **12.2** | **6.7** | **19** | **4.98** | **3.30** | **24** | **3.41** | **3.37** | **1.08** | **0.52** | **-6.41** |

## J  MD22 VISUALIZATIONS

Figure 4 visualizes the mean absolute errors (MAE) for energy and forces across molecules in the MD22 dataset. The $x$-axis represents the energy error (kcal/mol), and the $y$-axis denotes the force

error (kcal/mol/Å). Each point corresponds to a model, with performance improving as the point approaches the origin (0,0), where lower values indicate better performance for both metrics. Our proposed GotenNet consistently outperforms the baseline models, achieving the best performance across all molecules, as evidenced by its closer proximity to the origin compared to competing methods.

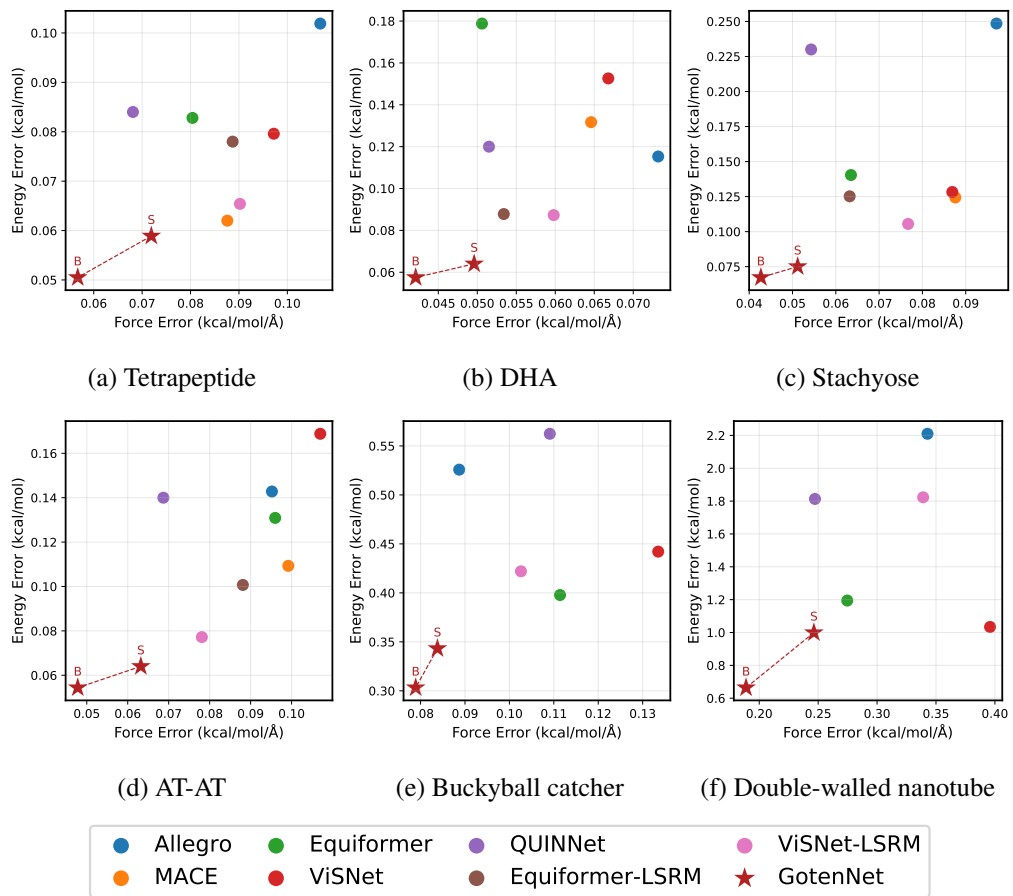

Figure 4: Mean absolute error of the molecules for energy and forces.

## K    rMD17 Visualizations

Figure 5 presents the MAE for energy and force predictions across nine molecules in the rMD17 dataset, including Aspirin, Azobenzene, Benzene, Ethanol, Malonaldehyde, Naphthalene, Paracetamol, Uracil, and Toluene. The $x$-axis represents energy error (mkcal/mol), and the $y$-axis denotes force error (mkcal/mol/Å). Each point corresponds to a model's performance on a specific molecule, where better performance is indicated by proximity to the origin (0,0) — reflecting lower errors in both metrics. Our proposed GotenNet demonstrates consistent superiority over baseline models, achieving the lowest errors across all nine molecules, as evidenced by its closer alignment with the origin compared to other methods.

## L    Future Work and Extensions

While GotenNet has demonstrated strong performance in molecular property prediction, several promising directions exist for future research and extensions. The architecture can naturally extend to other spatial data where geometric relationships significantly influence node interactions, such as point cloud processing, protein structure analysis, and dynamic molecular simulations. These applications

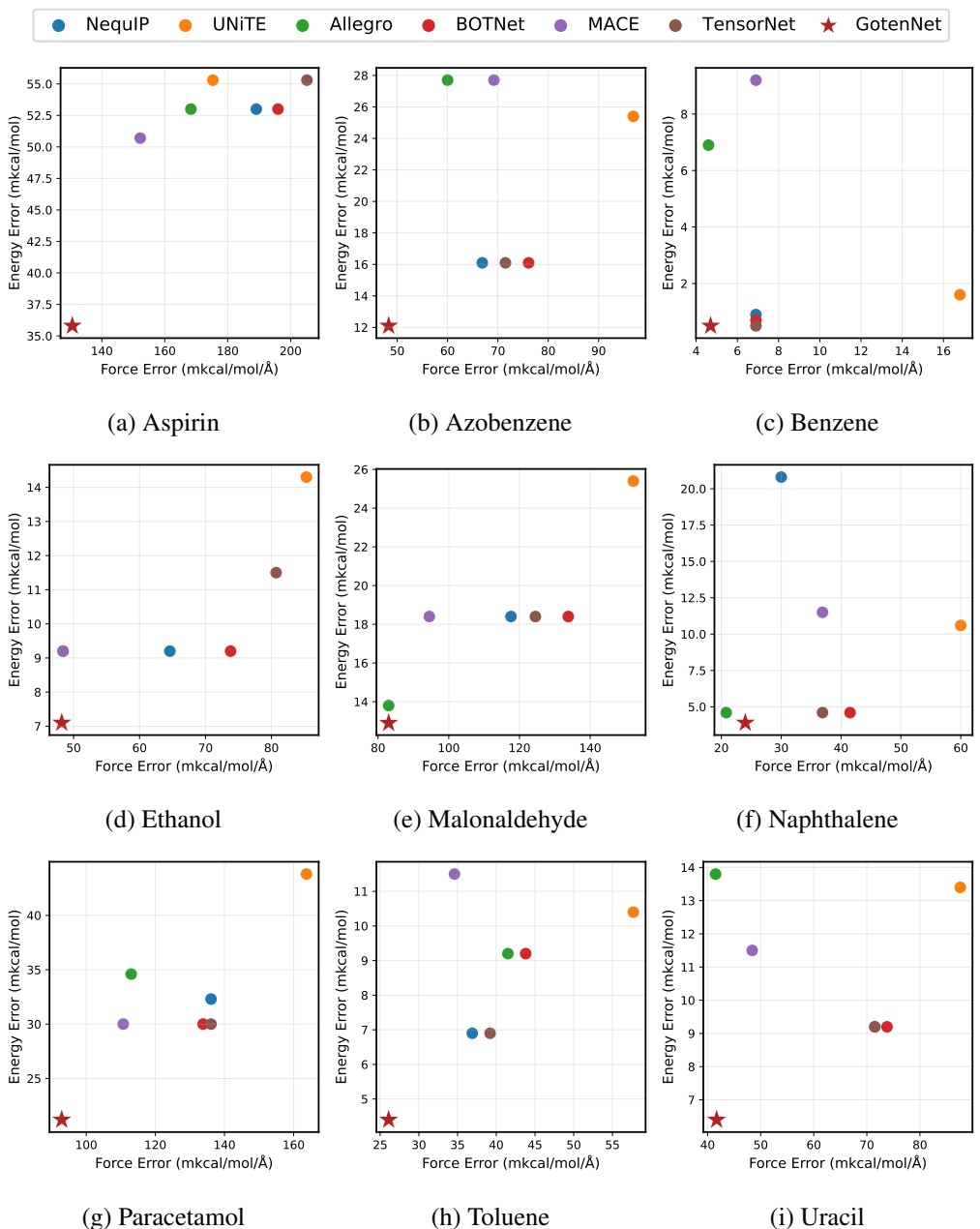

Figure 5: Mean absolute error of the molecules on rMD17 dataset for energy and forces.

share the fundamental requirement of processing geometric relationships while preserving symmetries, making them natural candidates for our framework. From an architectural perspective, the model could be enhanced through the incorporation of scale equivariance, exploration of higher-order features beyond second degree, and development of sparse implementations for larger systems. Memory efficiency improvements could also enable applications to even larger-scale systems. On the theoretical front, future work could focus on developing formal analyses of the expressiveness-efficiency trade-off, understanding generalization properties of geometric tensor representations. These potential extensions maintain GotenNet's core principle of balancing expressiveness and efficiency while broadening its applicability across different domains of geometric deep learning.

