# Efficient Equivariant Graph Neural Networks for 3D Atomistic Systems

## Abstract

Understanding complex three-dimensional (3D) structures of graphs is essential for accurately modeling various properties, yet many existing approaches struggle with fully capturing the intricate spatial relationships and symmetries inherent in such systems, especially in large-scale, dynamic molecular datasets. These methods often must balance trade-offs between expressiveness and computational efficiency, limiting their scalability. To address this gap, we propose a novel Geometric Tensor Network (GotenNet) that effectively models the geometric intricacies of 3D graphs while ensuring strict equivariance under the Euclidean group E(3). Our approach directly tackles the expressiveness-efficiency trade-off by leveraging effective geometric tensor representations without relying on irreducible representations or Clebsch-Gordan transforms, thereby reducing computational overhead. We introduce a unified structural embedding, incorporating geometry-aware tensor attention and hierarchical tensor refinement that iteratively updates edge representations through inner product operations on high-degree steerable features, allowing for flexible and efficient representations for various tasks. We evaluated models on QM9, rMD17, MD22, and Molecule3D datasets, where the proposed model consistently outperforms state-of-the-art methods in both scalar and high-degree property predictions, demonstrating exceptional robustness across diverse datasets, and establishes GotenNet as a versatile and scalable framework for 3D equivariant Graph Neural Networks.

## 1 Introduction

Accurately modeling 3D molecular systems is increasingly crucial in areas such as drug discovery (Jing et al., 2021; Nguyen et al., 2021; Huang et al., 2020; Chen et al., 2020; Yang et al., 2022; Zhang et al., 2022; 2021), materials science (Pablo-García et al., 2023; Reiser

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

 more computationally efficient solutions. ViSNet (Wang et al., 2024) established connections between inner products and Legendre polynomials through vector-scalar interactive message passing, though focusing on first-degree steerable features. SO3KRATES (Frank et al., 2024) demonstrated that certain applications of CG coefficients are equivalent to inner products of high-degree steerable features, achieving notable performance in property prediction tasks through their equivariant transformer architecture. HEGNN (Cen et al., 2024) further developed these concepts by introducing a scalarization approach using inner products to incorporate high-degree steerable features. This approach proved capable of capturing complete angular information between edge pairs and demonstrating enhanced model robustness in dynamics tasks. Our work advances this direction by introducing geometry-aware tensor attention, which employs a concise formulation of inner product operations combined with hierarchical refinement mechanisms. GotenNet represents a significant advancement in bridging the critical gap between scalarization-based and high-degree steerable models. Through its novel architectural design, GotenNet achieves superior

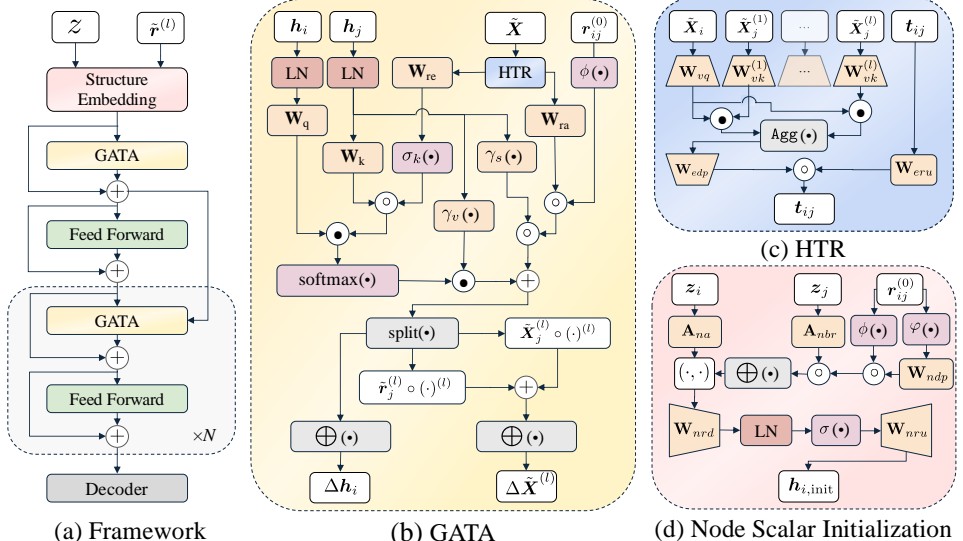

Figure 2: Architecture of GotenNet. The overall framework (a) includes an embedding, an interaction module, and decoder; (b) shows the geometry-aware tensor attention (GATA); (c) illustrates the hierarchical tensor refinement (HTR); and (d) presents the node embedding. In the figure, $+$ denotes addition, $\cdot$ denotes dot product, $\oplus$ denotes aggregation, $(\cdot, \cdot)$ denotes concatenation, $\circ$ denotes element-wise (Hadamard) product, LN denotes layer normalization, $\varphi$ denotes the radial basis functions, and $\gamma$ denotes differentiable functions such as MLPs.

performance in real-world molecular property prediction and force field calculations across diverse datasets.

## 3 GOTENNET

In this section, we introduce the key components of GotenNet. We first outline the efficient initialization and embedding design tailored for geometric tensors, which eliminates the need for irreps and CG transforms, reducing computational overhead. Next, we present our geometry-aware tensor attention and hierarchical tensor refinement mechanisms, which refine edge representations and enhance traditional dot product attention, enabling accurate and scalable predictions of molecular properties.

### 3.1 EQUIVARIANT GEOMETRIC TENSOR REPRESENTATIONS

Molecular property modeling requires accurate representation of both 3D spatial relationships and chemical interactions. The main challenges lie in encoding node-level and edge-level information while preserving the geometric and topological features of molecular structures. Our tensor-based geometric approach addresses these challenges by creating representations that maintain rotational and translational symmetries, efficiently capturing both local atomic interactions and global molecular patterns.

**Geometric Tensor Notations.** In our model, we distinguish between edge scalar features and edge tensor representations, employing spherical harmonics to initialize the latter. The edge tensor representation $\tilde{r}_{ij}^{(l)}$ is initialized based on the relative positions $\vec{p}_i$ and $\vec{p}_j$ of nodes $i$ and $j$, capturing spatial information from rank 0 to $L_{\max}$. Specifically, $\tilde{r}_{ij} = \{\tilde{r}^{(0)}, \tilde{r}^{(1)}, \ldots, \tilde{r}^{(L_{\max})}\}$, where each $\tilde{r}^{(l)}$ represents $l$-degree spherical harmonic functions. The components of $\tilde{r}_{ij}$ follow a hierarchical structure of increasing geometric complexity. At the most basic level, $\tilde{r}_{ij}^{(0)} = \|\vec{p}_i - \vec{p}_j\|$ captures the scalar distance between nodes, providing rotation and translation invariant information. The first-degree component $\tilde{r}_{ij}^{(1)} = (\vec{p}_i - \vec{p}_j)/\|\vec{p}_i - \vec{p}_j\|$ encodes directional information, introducing rotational equivariance. For $l \geq 2$, each $\tilde{r}_{ij}^{(l)}$ comprises $(2l + 1)$ functions derived from spherical

harmonics of degree $l$, where the degree determines the transformation behavior under rotations, and the parity of $l$ determines the behavior under inversion. These functions chosen to capture complex spatial relationships and rotational symmetries inherent in molecular structures. Leveraging the inherent normalization property of spherical harmonics, each $\tilde{r}^{(l)}$ for $l \geq 1$ is naturally normalized, ensuring consistent scaling across different representations.

We denote the geometric node tensors into two types of features: scalar features $h \in \mathbb{R}^{d_{ne}}$ which remain invariant under transformations, and high-degree steerable features $\tilde{X}^{(l)} \in \mathbb{R}^{(1+2l) \times d_{ne}}$ whose transformations depend on their degree $l$ where $d_{ne}$ denotes node embedding dimension. These representations are initialized and updated through message passing phases using the edge tensor representation $\tilde{r}_{ij}$ and edge scalar features $t_{ij}$ as input. The notation $\tilde{X}$ without a specified degree $l$ refers to the collection of features with degrees from 1 to $L_{\max}$. This initialization strategy enables our model to effectively capture, process, and propagate complex structural information.

Our initialization and feature design ensure equivariance throughout the network. A geometric tensor field maps 3D points to tensor quantities that transform equivariantly under geometric transformations, combining both invariant scalars and steerable features. Each layer of GotenNet processes these tensor fields through equivariant operations while preserving $E(3)$ transformations, with the final layer producing either equivariant geometric features or invariant representations as required by the task. This composition of equivariant operations ensures that the entire network maintains equivariance, with complete proofs provided in the Appendices A, B, and C.

## 3.2 Unified Structural Embedding: Integrating Content and Geometry

Our approach introduces a unified structure embedding that captures intrinsic atomic properties and relational information through an integrated node-edge interaction mechanism. By employing a dual representation strategy, we incorporate local geometric structure through node-edge interaction. This allows the model to simultaneously process both semantic and geometric information, enabling efficient message passing for both nodes and edges.

**Node Scalar Feature Initialization.** Node scalar features are obtained through a two-step process involving message passing and representation updates. Information from neighboring nodes is aggregated as:

$$\mathbf{m}_i = \sum_{j \in \mathcal{N}(i)} z_j \mathbf{A}_{\text{nbr}} \circ \left( \varphi(\tilde{r}_{ij}^{(0)}) \mathbf{W}_{\text{ndp}} \circ \phi(\tilde{r}_{ij}^{(0)}) \right), \tag{1}$$

where $z$ denotes the one-hot encoding of the atomic number, $\circ$ denotes element-wise product, and $\mathbf{A}_{\text{nbr}} \in \mathbb{R}^{|\mathcal{Z}| \times d_{ne}}$ is a learnable embedding matrix for neighbor atoms with maximum atomic number $|\mathcal{Z}|$. The radial basis functions $\varphi(\tilde{r}_{ij}^{(0)})$ encode the distance between nodes $i$ and $j$, which are then projected through $\mathbf{W}_{\text{ndp}}$. A cutoff function $\phi(\tilde{r}_{ij}^{(0)})$ is applied to modulate the influence of distant neighbors.

The initial node scalar feature is defined as:

$$h_{i,\text{init}} = \left( \sigma \Big( \text{LN}\big( (z_i \mathbf{A}_{\text{na}}, \mathbf{m}_i) \mathbf{W}_{\text{nrd}} \big) \Big) \right) \mathbf{W}_{\text{nru}}. \tag{2}$$

Here, $\mathbf{A}_{\text{na}} \in \mathbb{R}^{|\mathcal{Z}| \times d_{ne}}$ is a learnable embedding matrix for node atoms, $\sigma$ denotes a non-linear activation function and $(\cdot, \cdot)$ denotes concatenation operation. The concatenated node atom embedding and aggregated neighbor information undergo a series of transformations: node representation projections ($\mathbf{W}_{\text{nrd}}$, $\mathbf{W}_{\text{nru}}$), and layer normalization (LN).

**Edge Scalar Feature Initialization.** Edge scalar features are computed by combining node features with distance-based edge attributes:

$$t_{ij,\text{init}} = (h_{i,\text{init}} + h_{j,\text{init}}) \circ \left( \sigma \Big( \text{LN}\big( \varphi(\tilde{r}^{(0)}_{ij}) \mathbf{W}_{\text{erd}} \big) \Big) \right) \mathbf{W}_{\text{eru}}. \tag{3}$$

Edge attributes are processed through down-projection $\mathbf{W}_{\text{erd}}$ and up-projection $\mathbf{W}_{\text{eru}}$, enabling the integration of node-level features and spatial relationships. This formulation captures complex interactions between nodes while maintaining equivariance under molecular transformations.

**High-degree Steerable Feature Initialization.** The high-degree steerable features $\tilde{X}$ initialized during initial interaction layer with the following formulation:

$$\{o_{ij,\text{init}}^{(l)}\}_{l=1}^{L_{\max}} = \text{split}\Big(\mathbf{sea}_{ij} + (t_{ij,\text{init}}\mathbf{W}_{rs,\text{init}}) \circ \gamma_s(h_{j,\text{init}}) \circ \phi(\tilde{r}_{ij}^{(0)}), d_{ne}\Big),$$

$$\tilde{X}_{i,\text{init}}^{(l)} = \bigoplus_{j \in \mathcal{N}(i)} \Big(o_{ij,\text{init}}^{(l)} \circ \tilde{r}_{ij}^{(l)}\Big), \tag{4}$$

where $\mathbf{sea}_{ij}$ is self-attention with geometric encoding, $\mathbf{W}_{rs,\text{init}} \in \mathbb{R}^{d_{ed} \times d_{ne}}$ is a learnable weight matrix, $\gamma_s : \mathbb{R}^{d_{ne}} \to \mathbb{R}^{L_{\max} \times d_{ne}}$ is a differentiable function, and the split function decomposes the input tensor into $d_{ne}$-dimensional segments. These segments are used as different coefficients for each $l$-degree steerable features. $\bigoplus$ denotes a permutation-invariant aggregation function.

## 3.3 GEOMETRY-AWARE TENSOR ATTENTION

We introduce a novel module called Geometry-Aware Tensor Attention (GATA), which enhances the attention mechanism in graph neural networks by incorporating spatial information. GATA captures the geometric relationships between nodes to improve attention-driven message passing.

The GATA module combines self-attention with geometric encoding to generate rich node interaction representations. We compute the query ($q$), key ($k$), and value ($v$) representations:

$$q_i = h_i\mathbf{W}_q, \quad k_j = h_j\mathbf{W}_k, \quad v_j = \gamma_v(h_j), \tag{5}$$

where $\mathbf{W}_q, \mathbf{W}_k \in \mathbb{R}^{d_{ne} \times d_{ne}}$ are learnable weight matrices, and $\gamma_v : \mathbb{R}^{d_{ne}} \to \mathbb{R}^{S \times d_{ne}}$ is a differentiable function (e.g., MLP). The $S$ variable introduced to generate different coefficients for each degree of steerable features and formulated as $(1 + 2 \times L_{\max})$. The attention coefficients $\alpha_{ij}$ between nodes $i$ and $j$ using the dot product of the query vector $q_i$ and a geometry-infused key vector, which is obtained via an element-wise product of $k_j$ and a transformed edge embedding:

$$\mathbf{sea}_{ij} = \frac{\exp(\alpha_{ij})}{\sum_{k \in \mathcal{N}(i)} \exp(\alpha_{ik})} v_j, \quad \text{where} \quad \alpha_{ij} = q_i\big(k_j \circ \sigma_k(t_{ij}\mathbf{W}_{re})\big)^T. \tag{6}$$

Here, $\sigma_k$ denotes a non-linear activation function, and $\mathbf{W}_{re} \in \mathbb{R}^{d_{ed} \times C_{ne}}$ is a learnable weight matrix that transforms the edge scalar features. To incorporate spatial and directional information, we augment the attention mechanism with geometric encoding. The GATA operation combines self-attention with geometric features and is then split into $S$ components:

$$o_{ij}^s, \{o_{ij}^{d,(l)}\}_{l=1}^{L_{\max}}, \{o_{ij}^{t,(l)}\}_{l=1}^{L_{\max}} = \text{split}(\mathbf{sea}_{ij} + (t_{ij}\mathbf{W}_{rs}) \circ \gamma_s(h_j) \circ \phi(\tilde{r}_{ij}^{(0)}), d_{ne}), \tag{7}$$

where $\mathbf{W}_{rs} \in \mathbb{R}^{d_{ed} \times (S \cdot d_{ne})}$ is a learnable weight matrix, $\gamma_s : \mathbb{R}^{d_{ne}} \to \mathbb{R}^{S \times d_{ne}}$ is a differentiable function, and the split function decomposes the input tensor into $d_{ne}$-dimensional segments. We define $\Delta h_i$ and $\Delta \tilde{X}$ as the residues, which are calculated by: high-degree steerable features:

$$\Delta h_i = \bigoplus_{j \in \mathcal{N}(i)} (o_{ij}^s), \quad \Delta \tilde{X}_i^{(l)} = \bigoplus_{j \in \mathcal{N}(i)} \Big(o_{ij}^{d,(l)} \circ \tilde{r}_{ij}^{(l)} + o_{ij}^{t,(l)} \circ \tilde{X}_j^{(l)}\Big). \tag{8}$$

Here, each degree $l \in [1, L_{\max}]$ contributes its own component of steerable features weighted by their respective coefficients $o_{ij}^{d,(l)}$ and $o_{ij}^{t,(l)}$. Finally updated representations using residues calculated with:

$$h_i = h_i + \Delta h_i, \quad \tilde{X}_i^{(l)} = \tilde{X}_i^{(l)} + \Delta \tilde{X}_i^{(l)}, \tag{9}$$

By infusing geometric information into the attention mechanism, GATA allows the model to better capture spatial dependencies and fine-grained node interactions, leading to improved performance in molecular property predictions, as demonstrated in Section 4.

## 3.4 HIERARCHICAL TENSOR REFINEMENT

The Hierarchical Tensor Refinement (HTR) component processes graph-structured data through multi-scale analysis and layer-wise refinement. High-degree steerable features are projected to query

and key representations using degree-specific SO(3)-equivariant linear transformation (Deng et al., 2021; Du et al., 2023; Wang et al., 2024) as shown in Equation (10):

$$\widetilde{\boldsymbol{EQ}}_i^{(l)} = \tilde{\boldsymbol{X}}_i^{(l)} \mathbf{W}_{vq}, \quad \widetilde{\boldsymbol{EK}}_j^{(l)} = \tilde{\boldsymbol{X}}_j^{(l)} \mathbf{W}_{vk}^{(l)}, \quad \text{for } l \in \{1, \ldots, L_{\max}\}, \tag{10}$$

where $\mathbf{W}_{vq}, \mathbf{W}_{vk}^{(l)} \in \mathbb{R}^{d_{ed} \times d_{xpd}}$ are tensor query and key projection matrices, respectively. Here, $\mathbf{W}_{vq}$ is a shared projection matrix across degrees, while $\mathbf{W}_{vk}^{(l)}$ is degree-spesific. To maintain equivariance, we apply uniform weights across the spatial dimensions for each representation dimension. These projections aggregate angular and magnitude information between nodes across tensor degrees, defined as:

$$\boldsymbol{w}_{ij} = \text{Agg}_{l=1}^{L_{\max}}\left((\widetilde{\boldsymbol{EQ}}_i^{(l)})^\top \widetilde{\boldsymbol{EK}}_j^{(l)}\right), \tag{11}$$

where $\boldsymbol{w}_{ij} \in \mathbb{R}^{d_{xpd}}$ represents the aggregated similarity between nodes $i$ and $j$, and $\text{Agg}_{l=1}^{L_{\max}}$ denotes an aggregation operation. The aggregated information updates edge representations through a residual connection:

$$\boldsymbol{t}_{ij} = \boldsymbol{w}_{ij} \mathbf{W}_{edp} + \boldsymbol{t}_{ij} \mathbf{W}_{eru}, \tag{12}$$

where $\mathbf{W}_{edp} \in \mathbb{R}^{d_{xpd} \times d_{ed}}$ is an edge projection matrix, $\mathbf{W}_{eru} \in \mathbb{R}^{d_{ed} \times d_{ed}}$ is a residual update projection matrix. $d_{xpd}$ is chosen larger than $d_{ed}$ for richer intermediate representations. The weight matrices $\mathbf{W}_{edp}$ and $\mathbf{W}_{eru}$ apply the same values across the tensor dimensions, ensuring the model's equivariance is preserved.

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

 | 40 | 33.8 | 18.4 | 15.4 | 10 | 22 | 6.13 | 4.39 | 28 | 4.41 | 4.43 | 1.28 | 0.75 | -6.12 |
| **High-order steerable models** | | | | | | | | | | | | | | |
| SEGNN | 60 | 42 | 24 | 21 | 23 | 31 | 15 | 16 | 660 | 13 | 15 | 1.62 | 1.08 | -5.27 |
| HDGNN† | 46 | 32 | 18 | 16 | 17 | 23 | 11 | 10 | 342 | 8.12 | 8.34 | 1.21 | 0.80 | -5.64 |
| Equiformer | 46 | 30 | 15.4 | 14.7 | 12 | 23 | 7.63 | 6.63 | 251 | 6.74 | 6.59 | 1.26 | 0.70 | -5.82 |
| Equiformer$_{\text{V2}}$ | 47 | 29.0 | 14.4 | 13.3 | 9.9 | 23 | 7.57 | 6.22 | 186 | 6.49 | 6.17 | 1.47 | 0.67 | -5.87 |
| **Pre-trained models** | | | | | | | | | | | | | | |
| Transformer-M | 37 | 27.4 | 17.5 | 16.2 | 37 | 22 | 9.63 | 9.39 | 75 | 9.41 | 9.37 | 1.18 | 0.86 | -5.74 |
| SE(3)-DDM | 46 | 40.2 | 23.5 | 19.5 | 15 | 24 | 7.65 | 7.09 | 122 | 6.99 | 6.92 | 1.31 | 0.93 | -5.76 |
| 3D-EMGP | 57 | 37.1 | 21.3 | 18.2 | 20 | 26 | 9.30 | 8.70 | 92 | 8.60 | 8.60 | 1.38 | 0.92 | -5.68 |
| Coord | 52 | 31.8 | 17.7 | 14.3 | 12 | 20 | 6.91 | 6.45 | 450 | 6.11 | 6.57 | 1.71 | 0.76 | -5.75 |
| Frad(VRN) | 42 | 27.7 | 17.9 | 13.8 | 11 | 21 | 6.03 | 6.01 | 354 | 5.35 | 5.41 | 1.63 | 0.71 | -5.85 |
| Frad(RN) | 37 | 27.8 | 15.3 | 13.7 | 10 | 20 | 6.19 | 5.55 | 342 | 5.62 | 5.33 | 1.42 | 0.66 | -5.91 |
| **Hybrid geometric models** | | | | | | | | | | | | | | |
| GotenNet$_S$ | 35 | 23.2 | 16.3 | 14.7 | 7.5 | 20 | 5.51 | 3.86 | 26 | 3.76 | 3.82 | 1.15 | 0.62 | -6.27 |
| GotenNet$_B$ | 33 | 21.3 | 15.2 | 13.5 | 7.3 | 20 | 5.33 | 3.52 | 25 | 3.49 | 3.49 | 1.10 | 0.58 | -6.33 |
| GotenNet$_L$ | **30** | **19.9** | **13.7** | **12.2** | **7.7** | **19** | **4.98** | **3.36** | **21** | **3.33** | **3.37** | **1.08** | **0.54** | **-6.39** |

# K FUTURE WORK AND EXTENSIONS

While GotenNet has demonstrated strong performance in molecular property prediction, several promising directions exist for future research and extensions. The architecture can naturally extend to other spatial data where geometric relationships significantly influence node interactions, such as point cloud processing, protein structure analysis, and dynamic molecular simulations. These applications share the fundamental requirement of processing geometric relationships while preserving symmetries, making them natural candidates for our framework. From an architectural perspective, the model could be enhanced through the incorporation of scale equivariance, exploration of higher-order features beyond second degree, and development of sparse implementations for larger systems. Memory efficiency improvements could also enable applications to even larger-scale systems. On the theoretical front, future work could focus on developing formal analyses of the expressiveness-efficiency trade-off, understanding generalization properties of geometric tensor representations. These potential extensions maintain GotenNet's core principle of balancing expressiveness and efficiency while broadening its applicability across different domains of geometric deep learning.

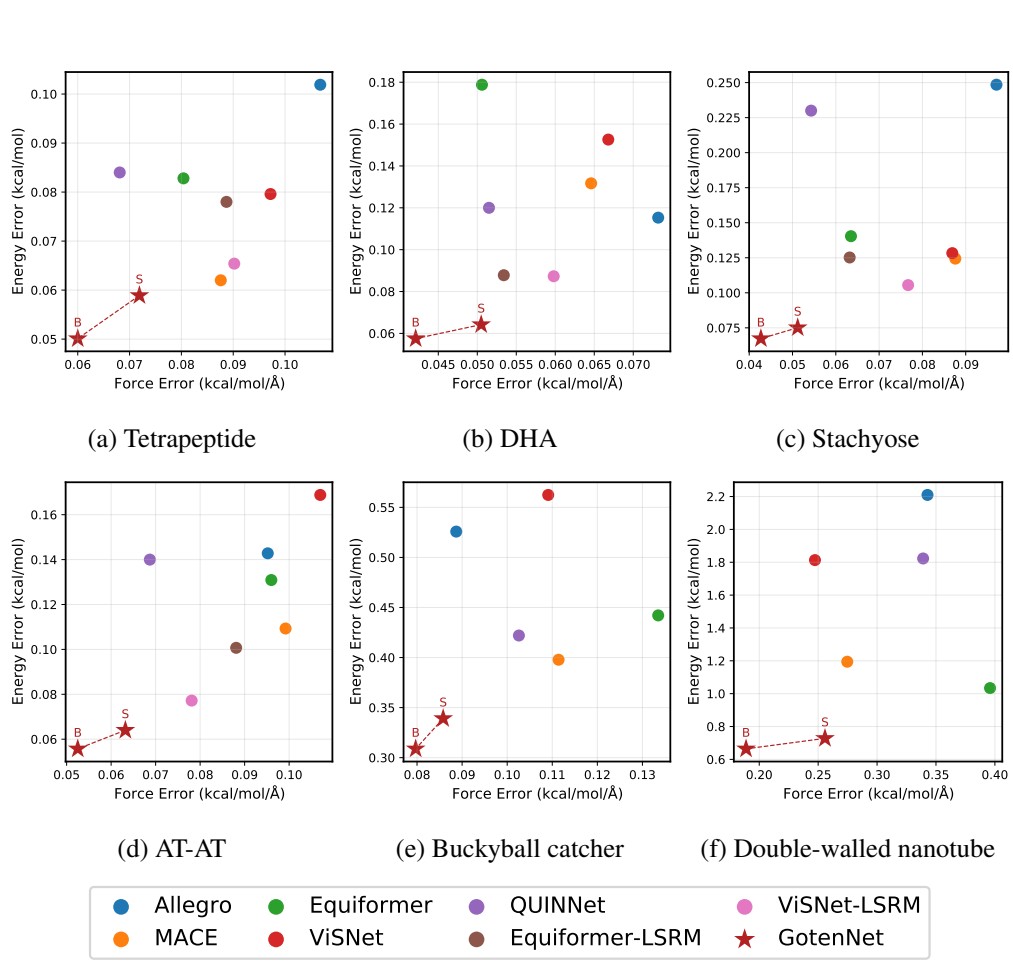

Figure 4: Mean absolute error of the molecules for energy and forces.

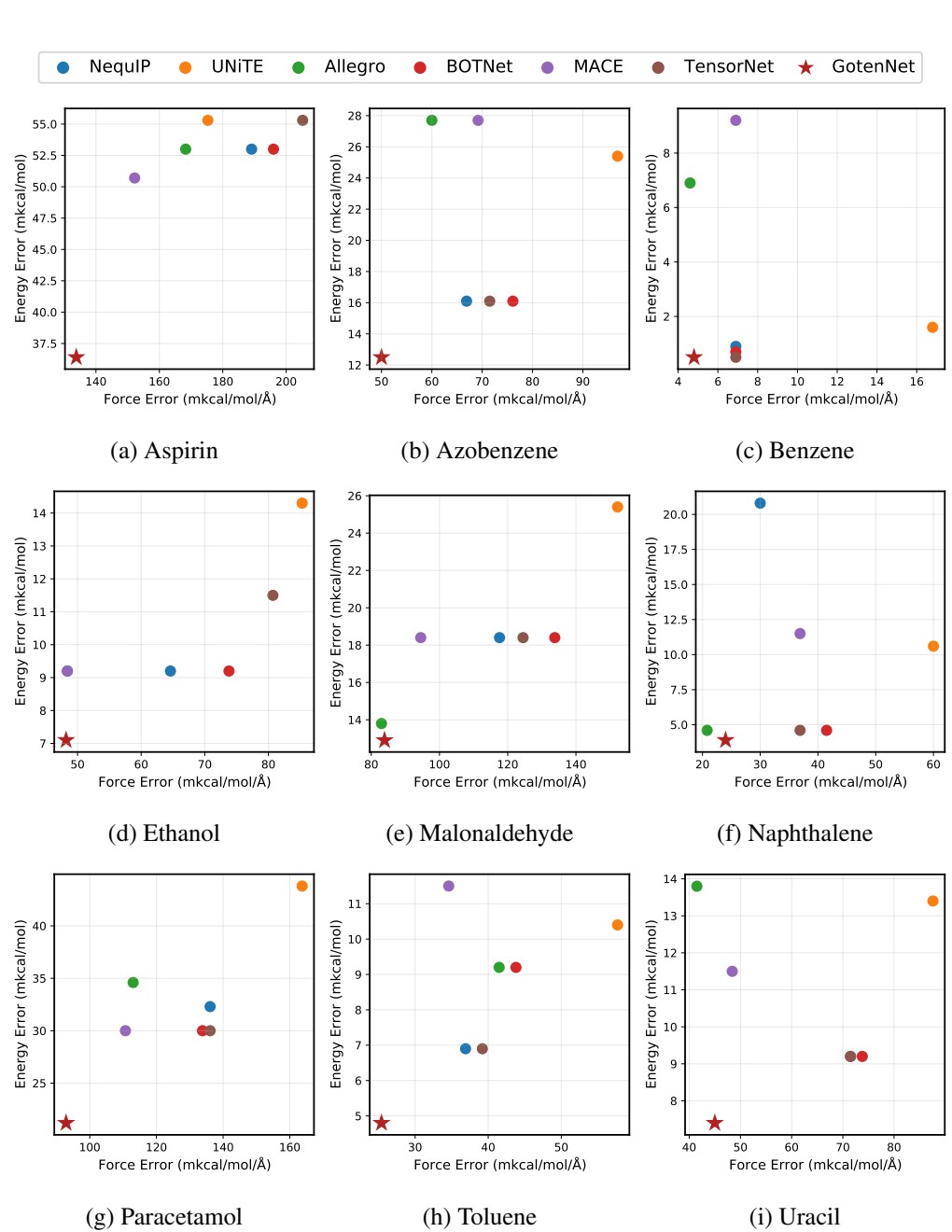

Figure 5: Mean absolute error of the molecules on rMD17 dataset for energy and forces.