# OpenReview forum: "GotenNet: Rethinking Efficient 3D Equivariant Graph Neural Networks"
_ICLR.cc/2025/Conference — ICLR 2025 Poster_

### Official Review · Reviewer_hFv3 · 2024-10-24

**Soundness:** 4
**Presentation:** 2
**Contribution:** 4
**Rating:** 10
**Confidence:** 5

**Summary:**

This paper propose a novel Geometric Tensor Network (GotenNet) to address  trade-offs between expressiveness and computational efficiency. The core of this model is to use inner-product in Eq. (9) to avoid  tensor products or CG coefficients.
***

In order to better explain my comments and facilitate the understanding of the author, other reviewers, and AC, I will sort out the timetable of related models that are not mentioned in the article, mainly involving the following references:
1. SE(3)-Transformer [a] (in NeurIPS 2020): In Eq. (11) of [a], the calculation of attention $\alpha$used $\bf{q}^\top\bf{k}$, which sum all inner-products of high-degree steerable features together.
2. SE(3)-Transformer implemented by e3nn [b]: See code below
```python
dot = o3.FullyConnectedTensorProduct(irreps_query, irreps_key, "0e")
...
exp = edge_weight_cutoff[:, None] * dot(q[edge_dst], k).exp()
```
And the main difference is the module $\texttt{o3.FullyConnectedTensorProduct}$, which introduces different learnable weights to each inner-product. However, now the model still uses tensor products to generate the key and value pair and is limited by efficiency.

3. ViSNet [c] (in Nature Communications: 05 January 2024, preprint at arXiv:2210.16518): In Eq. (6-7) of [c], authors introduced the connection between inner-products and Legendre polynomials (though a mathematical common sense). But in methodology, they only use the 1st-degree steerable features (Cartesian vectors) to build their models.

4. SO3KRATES [d] (in Nature Communications: 06 August 2024, preprint at arXiv:2309.15126): The SO3KRATES technique in Eq. (14), CG coefficient used in such form, but is equivalent to the inner-products of high-degree steerable features. This paper proposed an Euclidean transformer and achieved nice performance in property prediction tasks (e.g. rMD17 and MD22).

5. HEGNN [e] (in NeurIPS 2024, preprint at arXiv:2410.11443): During studying symmetric geometric graphs, the authors proposed a equivariant GNN model that uses the scalarization trick (i.e., inner product) to introduce high-degree steerable features in Eq. (6) of [e], and pointed out the connection between this approach and SO3KRATES. They used a method similar to Deepsets and Legendre polynomials to prove that this method can recover the information of all angles between each pair of edges, achieved good performance in dynamics tasks (e.g. N-body and MD17), and demonstrated that the introduction of high-degree steerable features can make the model more robust.

[a] SE(3)-Transformers: 3D Roto-Translation Equivariant Attention Networks

[b] https://docs.e3nn.org/en/stable/guide/transformer.html

[c] Enhancing geometric representations for molecules with equivariant vector-scalar interactive message passing

[d] A Euclidean transformer for fast and stable machine learned force fields

[e] Are High-Degree Representations Really Unnecessary in Equivariant Graph Neural Networks?

In subsequent comments, I will use the names of the above models without citation.

---
> **[First Comment, Rating 3, Confidence 5, with 4/1/4 of Soundness/Presentation/Contribution]**

The model and experiment in this paper are excellent and deserve a clear acceptance or even an award (8-10), but the terrible presentation of this version is totally unworthy of such an achievement. If the author can improve the presentation, I will be very happy to increase the score in the rebuttal. I am very confident that this is a paper that will have a profound impact on the entire field, so I hope it will be accepted in a very perfect manner.

>**[Second Comment, Rating 10, Confidence 5, with 4/2/4 of Soundness/Presentation/Contribution]**

The authors have provided a sincere and thoughtful response, addressing most of my concerns. The remaining questions, primarily related to experimental hyperparameters (e.g. higher-degree steerable features), would likely require significant time to resolve and may not be feasible during the rebuttal period, which I fully understand. Regarding the shortcomings in the presentation originally raised, the revised version has been significantly improved, and the motivation and logic are now clearer and more coherent. Although the presentation still needs further improvement, it is no longer worth over-focusing on during the discussion. Overall, these minor details do not detract from the paper’s value. I therefore give a **strong accept** recommendation.

**Strengths:**

For the increasingly important discipline of AI for Science, dealing with equivariance is undoubtedly very important. According to the classification of [f], there are two mainstream models: scalarization-based models and high-degree steerable models. The timeline I introduced in the summary (and this article) is actually doing something very valuable, that is, bridge the gap between these two types of models.

In fact, the only papers that truly achieve this are SO3KRATES, HEGNN, and GotenNet (this paper). Although the core formulas of these three works are equivalent, their publication dates are close together—especially since HEGNN's preprint appeared after the ICLR submission deadline. I believe this is merely a case of normal concurrent discovery, and therefore I recognize the originality of this paper. Moreover, equivariance is a very strong constraint, and I would even guess that this form is the only feasible form, which makes the contribution seem small in terms of formula form, but in fact it is a very important contribution.

Moreover, these three works exhibit significant differences. Both SO3KRATES and GotenNet employ equivariant attention, but the inner product formulation used in GotenNet is more concise and yields far better results than SO3KRATES. While both HEGNN and GotenNet point out the inner product formulation, HEGNN’s contributions lie more in theoretical research and model framework, with experiments focusing on dynamical problems, creating a complementary relationship with GotenNet. From my perspective, although GotenNet is on par with these two other works and shares an equivalent formulation, its superior performance makes it an excellent contribution.

[f] A Survey of Geometric Graph Neural Networks: Data Structures, Models and Applications

**Weaknesses:**

I think the whole article is amazingly good except for the presentation, and there is no major weakness. Here I mainly give the shortcomings of the presentation that I think need to be improved. For other parts (such as models and experiments), there are only some suggestions for the author to choose whether to adopt (see questions).

> **W1. Lack of a good review of relevant literature.**

This article should introduce several works in the timeline I sorted out in the summary section, especially SO3KRATES and HEGNN, which should be introduced in detail. The results of SO3KRATES should be added to the experimental results for easy comparison (although GotenNet outperform SO3KRATES).

For models such as eSCN and EquiformerV2 that use spherical activation functions, a simple comparison should be made. For example, the use of Fibonacci grid requires a large number of sampling points, which brings a large constant in complexity, resulting in a theoretical complexity of $\mathcal{O}(L^3)$, which is actually quite time-consuming (and this can only achieve qusi-equivariance).

> **W2. Weird math definitions and confusing math symbols.**

I don't know what the meaning of Def. 1, Theo. 1-2 in the main text is. I think it is just to make it look like there is some theory. Theo. 3 and Prop. 1, which have some meaning, are too trivial. They can be explained in one sentence: "The product of an invariant scalar and other steerable features does not change its equivariant form." However, the key initialization by spherical harmonics is not only placed in the appendix, but also a clear formula is not given (while Eq. (11) in SO3KRATES and Eq. (5) in HEGNN are clearly written). I suggest replacing this with a discussion of the model architecture. In fact, it is a very valuable point to explain why GotenNet performs so much better than SO3KRATES with a similar architecture. Theoretical analysis may be very difficult, so it is not required, but it would be great if the authors could come up with some engineering tricks.

Moreover, this article will surely have many followers in the future, and a good symbolic system will help future generations understand. The number of superscripts is sometimes one, sometimes two (though it has been explained), and the letters that appear are similar (the $l$ for the number of layers and the $L$ for high-degree steerable features are used at the same time), which is confusing. In addition, the current symbolic representation cannot well show which are invariant scalars and which are high-degree steerable features. It is recommended that the author adopt the HEGNN symbol system (e.g. using $\Delta\boldsymbol{h}$ to represent inter-layer updates, and using fonts such as $\tilde{\boldsymbol{v}}^{(l)}$ to represent $l$-degree steerable features).

> **W3. Others.**

- It is recommended to unify the terminology of the article, such as high-degree steerable features, and not to mix order and degree, otherwise it always gives readers the feeling that the authors are studying multi-body interactions.

- There are too many crossing lines in Fig. 2 (b-d), which makes it very difficult to understand. Can it be simplified?

- The rotation matrix $R$ in Line 879 (in fact, it should be emphasized that it is an orthogonal matrix) should be the Wigner-D matrix $\boldsymbol{D}^{(l)}(r)$. In addition, the inversion factor should be multiplied according to the parity of the vector itself. For details, refer to [g].

[g] https://docs.e3nn.org/en/stable/api/o3/o3_irreps.html

**Questions:**

Before anything else, is the author willing to provide an anonymous link to the project as soon as possible (this will not affect my subsequent rating increase) so that the results can be reproduced?

> **Q1. Why use the same weight for steerable features of different degrees (in Eqs. (7) and (11))? Can they be different?**

Note that there are several formula items: $o\_{ij}^d\circ r\_{ij}^{[1,L\_{max}]}$, $o\_{ij}^t\circ h\_{j}^{l,[1,L\_{max}]}$, $m\_2\circ h^{l,[1,\hat{L}\_{max}]}\bf{W}\_{vu}$. They seem to be multiplied by the same coefficient. Steerable features of different degrees represent different orthogonal bases, so their weights should not be the same (as HEGNN does). Can the authors change it to have different weights for each degree and test the performance of the corresponding model?

> **Q2. Why does Eq. (9) use a permutation-invariant operator?**

The results of different degrees should be ordered, so there seems no need for permutation invariance. What are the benefits of using permutation-invariant operations?

> **Q3. Is it possible to explore the expressive power of GotenNet using high-degree steerable features?**

One of the great benefits of this model is that the complexity grows slowly with the feature degree. Can authors show the experimental results of using higher-degree features (e.g. $L\_{\max}\in\\{6,8\\}$)? In fact, similar HEGNN and EquiformerV2 have demonstrated the benefits of using such higher-degree features.

> **Q4. Is it possible to explore the expressive power of GotenNet on large-scale geometric graphs and dynamics tasks?**

Is it possible to add speed attributes like EGNN so that GotenNet can be applied to dynamics tasks? In addition, it is easy to associate the fast multipole method with the possibility that high-degree representations may have good effects on large-scale geometric graphs. Can the authors try the effect of GotenNet on large-scale geometric graphs (e.g. 100-body in HEGNN)?

> **Q5. Is it possible to test the effect of $L\_{\max}=1$?**

Although the benefits of high-degree steerable features are obvious, in many cases a low-degree model is good enough. Can authors give a case where only Cartesian vectors ($L\_{\max}=1$) are used?

> **Q6. Is it possible to provide an e3nn version of GotenNet implementation later?**

From the full text, I guess the author may not know the e3nn library, which can easily implement the interaction of multi-channel high-degree steerable features. I believe it will be easier for everyone to follow the work of this article. However, this project may be very large, and I hope the author can provide this version after the article is accepted.

---

> ### Author Response · Authors · 2024-11-20
> **Questions 1-4**
>
> > **Q1. Why use the same weight for steerable features of different degrees (in Eqs. (7) and (11))? Can they be different?**
>
> We sincerely thank the reviewer for this insightful suggestion about weight differentiation for steerable features. Following your suggestion, we conducted extensive experiments with different weight configurations, particularly focusing on the interaction term: $\big\\{o^{d, (l)}\_{ij} \circ r^{(l)}\_{ij} + o^{t, (l)}_{ij} \circ \tilde{v}^{(l)}_j \big\\}^{L\_{max}}\_{l=1}$. Our experiments revealed several interesting findings:
> 1. Experimented with all three components to explore performance characteristics. Different coefficients on GATA improves performance, while EQFF shows no significant change.
> | Variation | $\varepsilon_{\text{HOMO}}$ | $U_0$    |
> | - | - | - |
> | $\big\\{o^{d, (l)}\_{ij} \circ r^{(l)}\_{ij}   \big\\}^{L\_{max}}\_{l=1}$ | 15.61 | 3.53   |
> | $\big\\{ o^{t, (l)}_{ij} \circ \tilde{v}^{(l)}_j \big\\}^{L\_{max}}\_{l=1}$  | 15.84  | 3.67  |
> | $\big\\{o^{d, (l)}\_{ij} \circ r^{(l)}\_{ij}  + o^{t, (l)}\_{ij} \circ \tilde{v}^{(l)}\_j \big\\}^{L_{max}}\_{l=1}$                                                                | **15.59**  | **3.49** |
> | $\big\\{o^{d, (l)}\_{ij} \circ r^{(l)}\_{ij}  + o^{t, (l)}\_{ij} \circ \tilde{v}^{(l)}\_j \big\\}^{L\_{max}}\_{l=1}$ , $(m\_2^{(l)} \circ \Delta{\tilde{v}{}}^{(l)}\textbf{W}_{vu})$ | 15.62 | 3.51  |
> | GotenNet$_{B}$ | 16.4 | 3.76 |
> 3. **Using different coefficients for steerable features significantly improves performance.** On QM9 dataset, this modification reduces errors for $\varepsilon_{\text{HOMO}}$ from 16.4 to 15.59 and $U_0$ from 3.76 to 3.49 in GotenNet$\_B$.
> 4. **The improvement scales well with model size.** For GotenNet$\_L$, we observe even more substantial gains, with $\varepsilon_{\text{HOMO}}$ improving from 14.3 to 13.67 and $U_0$ from 3.67 to 3.37.
> 5. **The benefits generalize across datasets.** Testing on the rMD17 dataset's Aspirin target shows consistent improvements in both energy (0.0364 → 0.0346) and forces (0.1338 → 0.1305).
> These results demonstrate both the effectiveness of differentiated weights and the flexibility of our framework. We are currently completing experiments across all targets shown as below and will include these improvements in our final version. We thank the reviewer for this valuable suggestion that has led to further performance gains.
>
>
> | Model                                                                                                                         | $\alpha$ | $\Delta \varepsilon$ | $\varepsilon_{\text{HOMO}}$ | $\varepsilon_{\text{LUMO}}$ | $\mu$ | $C_{\nu}$ | $G$  | $H$  | $R^2$ | $U$  | $U_0$ | ZPVE |
> | - | - | -- | - | - | - | - | - | - | - | ---- | ----- | ---- |
> | GotenNet$\_{S} + \big\\{o^{d, (l)}\_{ij} \circ r^{(l)}\_{ij}  + o^{t, (l)}\_{ij} \circ \tilde{v}^{(l)}\_j \big\\}^{L_{max}}\_{l=1}$ | 34.8 | 23.2                 | 16.3   | 14.7   | 7.5   | 20.4 | 5.51 | 3.86 | 26 | 3.76 | 3.82  | 1.15 |
> | GotenNet$_{S}$ | 37 | 25.4 | 18.4 | 15.7 | 7.5   | 21 | 5.67 | 4.17 | 33 | 3.97 | 3.89  | 1.16 |
> | GotenNet$\_{B} + \big\\{o^{d, (l)}\_{ij} \circ r^{(l)}\_{ij}  + o^{t, (l)}\_{ij} \circ \tilde{v}^{(l)}\_j \big\\}^{L_{max}}\_{l=1}$ | 33  | 21.3                 | 15.5  | 13.5   | 7.3   | -   | -    | -    | -     | -    | 3.49  | -    |
> | GotenNet$_{B}$  | 33  | 23   | 16.4 | 14.4   | 7.8   | 20 | 5.42 | 3.74 | 32    | 3.76 | 3.76  | 1.1  |
> |    |    |         |     |      |       |      |      |      |       |      |       |      |
>
>
> > **Q2. Why does Eq. (9) use a permutation-invariant operator?**
>
> We thank the reviewer for observation about Equation (9). Indeed, the operation does not require permutation invariance - we initially used a sum operator (which happens to be permutation-invariant) for simplicity and consistency with message passing aggregation. Following your suggestion, we have updated the notation to use the $\bigcirc$ symbol to indicate a generic aggregation operator, making this distinction clearer in the manuscript.
>
> > **Q3. Is it possible to explore the expressive power of GotenNet using high-degree steerable features?**
>
> We thank the reviewer for this interesting question about the expressive power of higher-degree steerable features. We are currently conducting experiments with $L\_{max} \in \\{4, 6\\}$ on $\varepsilon_{\text{HOMO}}$ and $U_0$ targets to explore this aspect. We look forward to sharing these results as they become available.
>
> > **Q4. Is it possible to explore the expressive power of GotenNet on large-scale geometric graphs and dynamics tasks?**
>
> We thank the reviewer for this valuable suggestion to explore GotenNet's capabilities on large-scale geometric graphs and dynamics tasks. We are actively working on integrating our framework with these applications. While the rebuttal period may not allow sufficient time for comprehensive results, we are committed to including this analysis in our final revision before publication.

---

> ### Author Response · Authors · 2024-11-20
> **Questions 5-6**
>
> > **Q5. Is it possible to test the effect of Lmax=1?**
>
> We thank the reviewer for this important question about the impact of different $L_{max}$ values. We have evaluated GotenNet with $L_{max}=1$ on the QM9 dataset and found interesting performance/efficiency trade-offs:
>
> While $L_{max}=2$ generally improves performance across most targets, for some properties like $C_{\nu}$ and ZPVE, $L_{max}=1$ performs equally well (and slightly better on ZPVE: 1.159 vs 1.163, though this difference might be attributed to training variance). This suggests that for certain properties, Cartesian vectors alone may be sufficient, offering potential computational savings.
>
> We will include a comprehensive comparison of $L_{max}=1$ versus $L_{max}=2$ for all targets to help readers make informed choices based on their specific tasks and efficiency requirements.
>
> > **Q6. Is it possible to provide an e3nn version of GotenNet implementation later?**
>
> We thank the reviewer hFv3 for suggesting an e3nn implementation of GotenNet. We agree this would be valuable for the research community and commit to providing an e3nn version of our framework at the time of publication. While the rebuttal period is too short to implement and verify this version, we will ensure it's available alongside our primary implementation.

---

> ### Comment · Reviewer_hFv3 · 2024-11-21
>
> It is recommended to mark the changes in the PDF file with special colors, and use Anonymous Github (https://anonymous.4open.science/) for anonymous code links.
>
> >**D1. The presentation needs further improvement.**
>
> - The blank lines between Line 54 and 55 are probably due to the settings in Fig. 1. You can adjust the blank lines by setting the parameters of the **wrapfigure** environment.
> - Bold font. It seems that all channel values that are not 1 (including multi-channel scalars, matrices, or high-degree steerable features) need to be bolded using the **\bm** command.
> - The use of $\Delta$. In Eq. (8), in HEGNN, $\Delta \boldsymbol{h}$ seems to mean the **residual** (i.e. the output $\boldsymbol{h}^{(k+1)}$ of $(k+1)$-th layer is the the output $\boldsymbol{h}^{(k)}$ of $k$-th layer plus the residual $\Delta \boldsymbol{h}$).
> - It is recommended to use $L_{\max}$ instead of $L_{max}$ (i.e. use \max command).
> - If authors can't find a suitable symbol, such as in Eq. 10, you can use $\texttt{Agg}$ (i.e. **\texttt{Agg}**).
> - Some formulas are missing symbols at the end (e.g. Eqs. (2,3,6,8,12)).
> - Formula citations lack brackets. Authors may consider using packages such as **cleveref** to achieve automatic citations.

---

> ### Comment · Reviewer_hFv3 · 2024-11-25
> **Additional Evidence Supporting the Acceptance of This Work**
>
> I would like to offer additional support for this paper and encourage all reviewers to consider accepting it. I checked several recent models based on different methodologies applied to the QM9 dataset (e.g., pre-trained models), and found that, even when compared to the latest Frad model [a] (in Nature Machine Intelligence: 18 September 2024, preprint at arXiv:2407.11086), GotenNet still keeps its SOTA . This paper can be regarded as a milestone contribution for general equivariant models, further reinforcing my recommendation for **"strong acceptance"**.
>
> Additionally, I recommend that the authors briefly discuss the introduction of high-degree steerable features in the appendix, similar to the theoretical discussions in ViSNet and HEGNN. Even if directly cited, references could be made to the appendix of SEGNN and the preprint of e3nn, including details on tensor products (and why they involve sixth-order complexity), as well as the representation power based on Legendre polynomials, etc. This would provide newcomers to the field with a broader understanding of the context when reading the paper.
>
> [a] Pre-training with Fractional Denoising to Enhance Molecular Property Prediction
>
> **I do not require the authors to complete these during the discussion stage. I hope the authors can focus on replying to other reviewers, and wish you good luck.**

---

> ### Comment · Reviewer_hFv3 · 2024-11-27
> **About GotenNet's Engineering Value**
>
> First, there are still some minor issues with the presentation:
> - Some tilde symbols of steerable features are missing
> - Some statements may need to be repositioned, such as the cutoff function in Line. 261 can be adjusted to near Eq. (4)
>
> In addition, I suggest that the authors test the effects of SO3krates and HEGNN on datasets such as QM9 if they have time. I tested the effect of HEGNN on QM9 by myself. Although it is better than EGNN, it is far inferior to GotenNet (I guess SO3krates is similar). Of course, I don’t mean to discredit HEGNN (after all, as a simple backbone, HEGNN does not use rbf kernel, cutoff function and other technologies at all). I just think that this can further illustrate the **engineering value** of this article (which is even more important in the field of AI for Science than the architectural innovation brought by the inner product).
>
> I hope the authors can give a big table about QM9 (maybe in the appendix). On the left are the categories of models, which are invariant models (such as DimeNet++, SphereNet), scalarization-based models (such as EGNN, PAINN, LEFTNet), high-order steerable models (such as Equiformer, EquiformerV2), pre-trained models (such as Frad and others in Frad paper), models that use inner products to introduce high-degree steerable features (such as SO3krates, HEGNN, the authors can give this category a new name to reflect the scalarization-trick), and finally GotenNet.
>
> **GotenNet is extremely practical and worth learning for many techniques in scalar processing. I did not very emphasize the engineering value of this article in my previous comments. I would like to reiterate it here and call on other reviewers, especially reviewer bNe3, to raise the rating of this article.**

---

> > ### Author Response · Authors · 2024-11-28
> >
> > Thank you for your thorough review and strong support of our manuscript. We have implemented several significant improvements based on your valuable feedback:
> >
> > 1. Notation and Presentation: We have fixed missing tilde symbols for steerable features and further aligned our notation with HEGNN for consistency. We have also reorganized the presentation of some statements, improving the logical flow of the manuscript.
> > 2. Model Performance: We are particularly excited to share breakthrough results that emerged from your insightful question about different coefficients for steerable features. Your suggestion led us to explore this direction during the rebuttal phase, resulting in substantial performance improvements. The updated QM9 results demonstrate these remarkable gains:
> >
> > | Model                                                                                                                         | $\alpha$ | $\Delta \varepsilon$ | $\varepsilon_{\text{HOMO}}$ | $\varepsilon_{\text{LUMO}}$ | $\mu$ | $C_{\nu}$ | $G$  | $H$  | $R^2$ | $U$  | $U_0$ | ZPVE | std. MAE | log MAE |
> > | ----------------------------------------------------------------------------------------------------------------------------- | -------- | -------------------- | --------------------------- | --------------------------- | ----- | --------- | ---- | ---- | ----- | ---- | ----- | ---- | -------- | ------- |
> > | $\text{GotenNet}\_{S}  +  \big\\{o^{d, (l)}\_{ij} \circ \tilde{r}^{(l)}\_{ij}  + o^{t, (l)}\_{ij} \circ \tilde{X}^{(l)}\_j \big\\}^{L\_{max}}\_{l=1}$ | 34.8     | 23.2                 | 16.3                        | 14.7                        | 7.5   | 20.4      | 5.51 | 3.86 | 26    | 3.76 | 3.82  | 1.15 | 0.62     | -6.27   |
> > | $\text{GotenNet}\_{S}  $                                                                                                              | 37       | 25.4                 | 18.4                        | 15.7                        | 7.5   | 21        | 5.67 | 4.17 | 33    | 3.97 | 3.89  | 1.16 | 0.67     | -6.21   |
> > | $\text{GotenNet}\_{B}  +  \big\\{o^{d, (l)}\_{ij} \circ \tilde{r}^{(l)}\_{ij}  + o^{t, (l)}\_{ij} \circ \tilde{X}^{(l)}\_j \big\\}^{L\_{max}}\_{l=1}$ | 33       | 21.3                 | 15.5                        | 13.5                        | 7.3   | 19.5      | 5.33 | 3.52 | 25    | 3.49 | 3.49  | 1.1  | 0.58     | -6.33   |
> > | $\text{GotenNet}\_{B}  $  | 33       | 23                   | 16.4                        | 14.4                        | 7.8   | 20        | 5.42 | 3.74 | 32    | 3.76 | 3.76  | 1.1  | 0.61     | -6.26   |
> > | $\text{GotenNet}\_{L}  +  \big\\{o^{d, (l)}\_{ij} \circ \tilde{r}^{(l)}\_{ij}  + o^{t, (l)}\_{ij} \circ \tilde{X}^{(l)}\_j \big\\}^{L\_{max}}\_{l=1}$ | 30       | 19.9     | 13.7         | 12.2    | 7.7   | 19        | 4.98 | 3.36 | 21  | 3.33 | 3.37  | 1.08 | 0.54   | -6.39  |
> > | $\text{GotenNet}\_{L} $   | 30       | 20.7                 | 14.3                        | 13.3                        | 7.7   | 19        | 5.27 | 3.47 | 25    | 3.58 | 3.67  | 1.08 | 0.56     | -6.34   |
> >
> > 3. Comprehensive Model Comparison: Following your suggestion, we have expanded Table 1 with updated QM9 results and added a new comprehensive comparison (Table 8) in the appendix that categorizes different approaches.
> >
> > 4. Theoretical Discussion: We appreciate your suggestion regarding the theoretical discussion of high-degree steerable features. In the final version, we plan to enhance the appendix with more detailed theoretical foundations to provide readers with a comprehensive understanding of the underlying concepts.
> >
> > We remain committed to further improving the manuscript and welcome any additional suggestions you may have. Thank you again for your continuous constructive feedback and strong recommendation for acceptance.

---

> ### Comment · Reviewer_hFv3 · 2024-12-02
>
> Since this comment involves the code of this article, if it is not suitable for public discussion, please remind me to set the public scope.
>
> I checked the author's code, successfully ran it and extracted several indicators on QM9 to verify the effect, which is consistent with the results reported in the article. I have to say that the code given by the authors is very neat and worth learning, and I have benefited a lot from it. I noticed that the TensorInit class in src/models/components/ops.py is actually based on e3nn. I suggest that the authors cite e3nn's software library or their arxiv in this article.
>
> Regarding the presentation part, I previously suggested using high-degree instead of high-order, and the authors have made changes. However, high-order is still written in several newly added places (Line. 1150 and 1174). In addition, I do not recommend using $\otimes$ to represent element-wise operations, because it is very easy to confuse with CG tensor product, although I have not thought of any better symbolic replacement at present.
>
> Finally, regarding the name 'Hybrid geometric models', I think it may not be obvious enough. Maybe 'Sph-scalarization models' would be better? Because all high-degree steerable features are initialized from spherical harmonics, and the core is the inner product scalarization (SO3KRATES is actually the modulus length). Then the authors can use this name to introduce the three models in section 2.2.

---

> > ### Author Response · Authors · 2024-12-03
> >
> > Thank you for your thorough analysis of our reference implementation and your constructive feedback. We greatly appreciate the time you took to verify our results and examine our code in detail.
> >
> > - Yes, you are correct about the e3nn connection in our `TensorInit` class and have added the appropriate citation to the e3nn arXiv paper in our manuscript.
> > - Regarding the notation for element-wise operations, we agree that $\otimes$ could potentially cause confusion with CG tensor products. We will maintain our original notation using $\circ$ for element-wise operations, and have added explicit clarification about its broadcasting behavior in the manuscript to ensure clarity.
> > - We appreciate your suggestion regarding the terminology. We have adopted "Spherical-scalarization models" as the category name for this class of approaches, as it better reflects how these models initialize high-degree steerable features from spherical harmonics and use inner product scalarization. We have updated Section 2.2 to introduce our framework within this context.
> > - We have corrected the remaining instances of "high-order" to "high-degree" in the newly added text.
> >
> > Your engagement throughout the discussion period has been invaluable in improving both the clarity and technical accuracy of our work. We sincerely appreciate your supportive approach and detailed suggestions that have helped enhance the quality of our manuscript.

---

### Official Review · Reviewer_BR7w · 2024-11-02

**Soundness:** 3
**Presentation:** 3
**Contribution:** 3
**Rating:** 6
**Confidence:** 3

**Summary:**

The importance of equivariance in neural networks has become ubiquitous
in machine learning research. Neural networks operating on input features
with certain symmetries, such as rotations, translation and permutation
predict with higher certainty and better quality if the architecture respect
these symmetries. In Graph Neural Networks equivariance with respect to
permutation of the nodes has been the foundation upon which a plethora
of ML approaches have been developed. The authors present a deepening
of this paradigm, by combining the success of transformers, permutation
invariance and E(3) equivariance and distilling these three components into
an architecture for molecular learning. The main contribution of the paper
is the development of Geometry Aware Tensor Attention layer that defines
an E(3) equivariant transformer layer for both node and edge embeddings. A
strong highlighted advantage is computational speed, compared to architectures
using Chlebsch-Gordan decompositions or architectures based on irreducible
representation.

Through a thorough set of experiments, the authors show that their
architecture, GotenNet, is able to outperform a wide variety of benchmarks on
molecule property predictions. Their overall performance and scaling in terms
of computations is also shown.

**Strengths:**

The paper presents a rather strong experimental section showing that the
authors have taken the time to set up a good set of experiments. In particular
the fact that all comparison partners have been retrained and compared to
three versions of the model shows dedication to the and has been a pleasure
to read. The experiments addressing the inference times is an important
complementary addition, since often there is a rather steep tradeoff between
practicality and usability, where better performance on metrics is achieved
through exponential increases in parameter count or long inference times. To
see both good results and good inference time / scalability is reassuring to
the reviewer. The introduction of geometric tensors as a replacement of CG
coefficient and the computation irreducible representation is a novel and
original viewpoint which combined with the favorable tradeoff between good
performance and accuracy is an impactful contribution. Writing is clear,
concise and easy to follow.

**Weaknesses:**

Since the reviewer is not too familiar with the literature, the current
writing makes it somewhat hard to compare to the current available literature.
For instance, what would be the comparable difference between GotenNet and an
equivariant transformer (such as equiformer) or the Graph Attention Networks.
Is the current architecture a mix of the two? In that sense some more context
might be good in the introduction or overview of related work.

The paper as written in its current form is of great quality, and could be
potentially be further strengthened through the incorporation of the points
above and by addressing the questions below in a section discussing the current
limitations of the method and potential future avenues of research. For
instance it seems that the current method is _specifically_ tailored to the
prediction of molecule properties and how would that generalize? Another
point to potentially discuss is how this method would extend beyond $E(3)$
and $SE(3)$ equivariance (maybe scaling for instance) and/or if the method
would generalize to point clouds (where this line of work is also highly
relevant). Since the paper is already of great quality addressing these
topics in the rebuttal would be sufficient, however addressing them in the
revision would be beneficial for the reader.

**Questions:**

Some questions:

-  How  does  the  current architecture  generalize  to  different tasks? As
presented, it  would work only on  molecule type graphs, is it also possible to
generalize the architecture to for instance social  networks or  other types
of graphs? In  other words,  how general  is the  method.

---

> ### Author Response · Authors · 2024-11-20
> **Weakness and Question**
>
> > Weakness: comparison with the current available literature
>
> We thank the reviewer for this valuable suggestion to better contextualize our work within existing literature. We have substantially revised our introduction and related work sections to clarify the evolution of approaches in this field and GotenNet's distinct contributions.
>
> The current landscape of equivariant networks can be broadly categorized into two approaches [1]: (1) scalarization-based methods like Graph Attention Networks, which prioritize computational efficiency but may sacrifice expressiveness, and (2) high-degree steerable models [4,5], which achieve high expressiveness through CG coefficients and tensor products but incur significant computational overhead. **GotenNet bridges these approaches** through two key innovations:
>
> 1. While state-of-the-art works [4,5] uses CG transforms to leverage high-order steerable features, our geometry-aware tensor attention (GATA) directly operates on geometric tensor representations without CG transforms. This crucial difference allows us to maintain the expressiveness of high-degree features while achieving the computational efficiency typically associated with scalarization-based methods.
>
> 2. Recent works like SO3KRATES [2] and HEGNN [3] demonstrated theoretical connections between CG coefficients and inner products. SO3KRATES and GotenNet utilizes equivariant attention, but the inner product formulation used in GotenNet is more concise and yields far better results than SO3KRATES. As noted by reviewer hFv3, although both HEGNN and GotenNet employ inner product formulation, GotenNet distinguishes itself by focusing on competitive real-world tasks while HEGNN primarily addresses dynamic problems. Therefore, the architectures of the concurrent works, HEGNN and GotenNet, are fundamentally different, tailored for distinct application domains.
>
> We have revised our manuscript to better highlight these distinctions and their implications for both theoretical understanding and practical performance. This includes expanded comparisons with existing architectures in both the introduction and related work sections.
>
>
> > Questions: How does the current architecture generalize to different tasks? As presented, it would work only on molecule type graphs, is it also possible to generalize the architecture to for instance social networks or other types of graphs? In other words, how general is the method.
>
> We sincerely thank the reviewer for their positive assessment and thoughtful suggestions about future directions. The proposed GotenNet framework can be readily extended to any spatial data where interactions between nodes are heavily influenced by geometric properties. While our current work focuses on molecular property prediction, the core architectural components - geometric tensor representations and attention mechanisms - are domain-agnostic and can generalize to various applications. For instance, several promising domains could benefit from our framework:
>
> 1. Point Clouds: As the reviewer noted, our framework can be adapted to point cloud processing by modifying the initial feature embedding while maintaining the core equivariant operations.
>
> 2. Protein Structure Analysis: The efficient handling of high-degree steerable features through our GATA and HTR mechanisms could benefit protein structure prediction and analysis.
>
> 3. Dynamic Molecular Simulations: Our model's ability to capture multi-scale geometric relationships makes it particularly suitable for molecular dynamics applications.
>
> These applications represent natural extensions of our framework, as they share the fundamental requirement of processing geometric relationships while preserving symmetries. We appreciate these insightful suggestions and included a discussion of these potential applications in our revised manuscript in Appendix K.
>
>
> [1] Jiaqi Han, Jiacheng Cen, Liming Wu, Zongzhao Li, Xiangzhe Kong, Rui Jiao, Ziyang Yu, Tingyang Xu, Fandi Wu, Zihe Wang, Hongteng Xu, Zhewei Wei, Yang Liu, Yu Rong, and Wenbing Huang. A survey of geometric graph neural networks: Data structures, models and applications, 2024.
>
> [2] J Thorben Frank, Oliver T Unke, Klaus-Robert Müller, and Stefan Chmiela. A euclidean transformer for fast and stable machine learned force fields. 2024.
>
> [3] Jiacheng Cen, Anyi Li, Ning Lin, Yuxiang Ren, Zihe Wang, and Wenbing Huang. Are high-
> degree representations really unnecessary in equivariant graph neural networks? 2024.
>
> [4] Yi-Lun Liao and Tess Smidt. Equiformer: Equivariant graph attention transformer for 3d atomistic graphs. 2023.
>
> [5] Yi-Lun Liao, Brandon M Wood, Abhishek Das, and Tess Smidt. Equiformerv2: Improved equivariant transformer for scaling to higher-degree representations. 2024.

---

> > ### Author Response · Authors · 2024-12-01
> > **Follow-up: Have we sufficiently addressed the concerns?**
> >
> > We sincerely thank you for your thoughtful review and constructive feedback. We have carefully addressed both key concerns you raised:
> >
> > 1. Regarding the comparison with current available literature, we have substantially expanded our introduction and related work sections to better contextualize GotenNet within the existing landscape. As detailed in our response, we now clearly position our work between scalarization-based methods and high-degree steerable models, highlighting our distinct technical contributions compared to approaches like Equiformer and Graph Attention Networks. The revised manuscript makes these comparisons explicit and accessible even to readers less familiar with the literature.
> > 2. Regarding the generalizability of our method, we appreciate your insightful question about extending beyond molecular applications. We have added a comprehensive discussion in Appendix K that explores potential applications to point clouds, protein structure analysis, and molecular dynamics simulations. This addition helps clarify the broader applicability of our geometric tensor representations and attention mechanisms.
> >
> > All these changes are highlighted in blue in our revised manuscript for easy reference. We believe these revisions have substantially strengthened the paper's clarity and accessibility while maintaining its technical depth.
> >
> > We would greatly appreciate your feedback on whether these revisions adequately address your concerns. We wonder if there are any remaining concerns or aspects of our work that could benefit from further clarification or improvement. We are fully committed to enhancing the quality and impact of our research, and would greatly value any additional suggestions you might have to strengthen our contribution.
> >
> > Thank you again for your detailed review that helped us improve this work.

---

### Official Review · Reviewer_bNe3 · 2024-11-02

**Soundness:** 2
**Presentation:** 1
**Contribution:** 2
**Rating:** 3
**Confidence:** 3

**Summary:**

This paper proposed a new network archtiecture that is SE(3) equivariant and builds on Transformers and previous works on equivariant Transformers. The results on various datasets show good performance and speedup.

**Strengths:**

1. The experimental results are great and intensive.

**Weaknesses:**

1. The writing of the paper should be greatly improved. Please see "Questions" below.
2. The paper does not discuss why the proposed architecture differs from other works and why this is better in terms of accuracy and efficiency even though the results are seemingly good. Also a reference implementation would be great to make sure the better results are reproducible.
3. The proposed architecture should be simplified (partially because of the way it is presented).
4. The comparison is somewhat unfair. For example, on QM9, this work trains the proposed model for 1,000 epochs while some previous works only train for 300 epochs. Besides, the batch size is 32 on QM9 (some previous works are 128), and smaller batch sizes on QM9 can sometimes lead to better results.

**Questions:**

> Writing

1. I think the title is too general. We can always rethink something and make them more efficient. Be very specific to your proposed method.
2. Line 16 -- 18: Mention what is the difference from previous works clearly.
3. Line 22 -- 23: Mention the datasets you tested so that readers can know the scale of experiments in this work.
4. Figure 1: I think the x and y axes are similar. Also mention which direction is better.
5. Line 62: What is "this" in "Some recent works have sought to address this by..."? Make sure the sentence is clear.
6. Line 113: Equivariant networks model translational "invariance".
7. Line 159: "Effective Representations" -> too general. Be specific to what makes it effective.
8. Figure 2: Please double checkt the errors in the caption.
9. Line 195 -- 215: Better to link to existing literature since they are similar to (or basically the same as) vector spaces of irreps.
10. Line 220 -- 221: The notation of h, r, t has no meaning and should be replaced by other variables that directly reflect what they are presenting.
11. Line 263: $z$ denotes the "one-hot encoding" of the atomic number.
12. Section 3.3: This is unclear. It would be better to first mention the high-level concept and then go to the details. Also state the differences from previous works.
13. Line 323 or Equation 8: This is essentially an SO(3) linear layer. It would be great to link to previous works.
14. Line 373: Geoformer Wang et al. -> make sure the format of citation is correct.
15. Figure 3: Mention the dataset you tested.
16. Line 423: "The" proposed method
17. Line 448: I don't get the definition of Scalability in the paper. Should be better to use "Evaluation on Efficiency"?

> Question

1. Line 66 -- 67: How do you define scalability? I think that means you invest more compute, you always get better results. Efficiency should be the correct term to use if you are saying some models are slow or take much memory.
2. Table 1: Somewhat hestitated to trust the results here. From the method section, it is unclear to me what are the differences that can improve the results. Besides, it would be great to discuss some potential oversmoothing, which prevents deeper networks from performing better.
3. It would be good to report training time for completeness.
4. Have you conducted any experiments on materials/catalyst datasets? I think OC20 IS2RE dataset would be good to test given the compute requirement is not that high and the train/val/test splits are well-defined.

---

> ### Author Response · Authors · 2024-11-20
> **Weakness and Question 3-4**
>
> > Q3. It would be good to report training time for completeness.
>
> > W4: The comparison is somewhat unfair. For example, on QM9, this work trains the proposed model for 1,000 epochs while some previous works only train for 300 epochs. Besides, the batch size is 32 on QM9 (some previous works are 128), and smaller batch sizes on QM9 can sometimes lead to better results.
>
> We thank the reviewer for raising this important point about experimental fairness. While standardized protocols would be valuable, current literature  [1,2,3,4] shows considerable variation in training configurations. For instance, recent works like Equiformer [2] (batch size 128, 300 epochs) and EquiformerV2  [3] (batch sizes 48/64, 300 epochs) compare with SphereNet  [1] (batch size 32, 1,000 epochs), suggesting that a specific configuration has not been established as a standard.
>
> For transparency, our models used early stopping and actually trained for an average of ~550 epochs, not the full 1,000 epochs. To address the efficiency concern, we provide a detailed comparison:
>
> | Model          | Batch Size | Time per Epoch (s) | Min. | Avg. | max. | Limit | Inference Latency (ms) |
> | -------------- | ---------- | ------------------ | :--: | :--: | :--: | :---: | ------------------------------------ |
> | Equiformer     | 128        | 425                | 1.48 | 1.48 | 1.48 | 1.48  | 150                                  |
> | EquformerV2    | 64         | 821                | 2.85 | 2.85 | 2.85 | 2.65  | 341                                  |
> | EquformerV2    | 48         | 847                | 2.94 | 2.94 | 2.94 | 2.65  | 341                                  |
> | GotenNet$_{S}$ | 32         | 117                | 0.41 | 0.75 | 1.34 | 1.35  | 37                                   |
> | GotenNet$_{B}$ | 32         | 180                | 0.75 | 1.15 | 1.92 | 2.08  | 56 |
> | GotenNet$_{L}$ | 32         | 291                | 1.37 | 1.87 | 2.33 | 3.37  | 112 |
>
> Even with batch size 32, GotenNet variants demonstrate superior efficiency. Our largest model (GotenNet$_{L}$) requires 2.33 GPU days at maximum, which is less than EquiformerV2  [3] (2.65 GPU days), while achieving **67% faster inference time** with batch size 128. The higher batch sizes and lower epoch counts in previous works may have been necessitated by their greater computational demands, as evidenced by their significantly higher per-epoch training times.
>
> We **included these detailed efficiency comparisons** in our Appendix G for completeness.
>
> > W3: The proposed architecture should be simplified.
>
> We thank the reviewer on the constructive feedback on the presentation of the architecture. The presentation of the architecture in methodology section is heavily revised for clarity. The equations are simplified with improved symbolic system. The authors believe the modifications will help readers understand the architecture.
>
> > Q4. Have you conducted any experiments on materials/catalyst datasets? I think OC20 IS2RE dataset would be good to test given the compute requirement is not that high and the train/val/test splits are well-defined.
>
> We thank the reviewer for suggesting the OC20 IS2RE benchmark and acknowledging our **"great and intensive experimental results."** Our current evaluation suite includes four diverse datasets (QM9, Molecule3D, rMD17, and MD22) providing comprehensive validation of our method. Given the distinct hyper-parameters required for OC20, as noted in previous works [3], properly benchmarking on this dataset would exceed the scope of the rebuttal period. We look forward to exploring this direction in future work.
>
>
> [1] Yi Liu, Limei Wang, Meng Liu, Yuchao Lin, Xuan Zhang, Bora Oztekin, and Shuiwang Ji. Spherical message passing for 3D molecular graphs. 2022.
>
> [2] Yi-Lun Liao and Tess Smidt. Equiformer: Equivariant graph attention transformer for 3d atomistic graphs. 2023.
>
> [3] Yi-Lun Liao, Brandon M Wood, Abhishek Das, and Tess Smidt. Equiformerv2: Improved equivariant transformer for scaling to higher-degree representations. 2024.
>
> [4] Yusong Wang, Shaoning Li, Tong Wang, Bin Shao, Nanning Zheng, and Tie-Yan Liu. Geometric transformer with interatomic positional encoding. 2023a.
>
> [5] J Thorben Frank, Oliver T Unke, Klaus-Robert Müller, and Stefan Chmiela. A euclidean transformer for fast and stable machine learned force fields. 2024.
>
> [6] Jiacheng Cen, Anyi Li, Ning Lin, Yuxiang Ren, Zihe Wang, and Wenbing Huang. Are high-degree representations really unnecessary in equivariant graph neural networks? 2024.

---

> ### Author Response · Authors · 2024-11-24
> **Follow-up: Have we sufficiently addressed the Reviewer's concerns?**
>
> We sincerely thank you for your detailed and constructive feedback on our submission. We have provided comprehensive responses addressing each of your concerns, including thoroughly **revising the writing and presentation** in our manuscript with all changes highlighted in blue for easy reference, clarifying our definition of scalability, providing access to our reference implementation, explaining the **key technical innovations that enable GotenNet's superior performance**, providing **detailed training time comparisons**, addressing the fairness of experimental settings, and **simplifying the architectural presentation with improved symbolic notation**.
>
> We believe these responses and additional analyses have substantially strengthened our paper. We would greatly appreciate your feedback on whether our responses have adequately addressed your concerns and if the additional efficiency metrics help validate our performance claims. If there are any remaining aspects you would like us to clarify or address, we are happy to provide further information. If you find our responses satisfactory, we would be grateful if you could reconsider your rating of our submission.
>
> Thank you again for your time and dedication in helping us improve this work.

---

> > ### Comment · Reviewer_bNe3 · 2024-12-02
> > **Comments on Author Responses**
> >
> > Thank you for your response. Please see my comments below.
> >
> > > 1. Presentation
> >
> > I still think the presentation needs to be improved. The issue is that from the presentation, it is unclear why the architecture is better while the authors spend lots of space reitering things alreay exist in the literature. Moreover, Figure 2 needs to be greatly improved to reflect the contribution of this work.
> >
> > > 2. "Our primary goal is to address the fundamental trade-off in equivariant networks where models must choose between expressiveness and efficiency."
> >
> > Again it is unclear to me how this is achieved. The authors are supposed to answer this clearly in their response.
> >
> > > 3. OC20 IS2RE experiment.
> >
> > I asked for experiments on OC20 "IS2RE" instead of OC20 "S2EF". The OC20 IS2RE results take at most 48 hours on 2 GPUs based on Equiformer paper while the authors cite EquiformerV2 takes too much on OC20 S2EF. My intention is to suggest authors to compare their methods on one larger (yet affordable) and well-benchmarked dataset to make sure their evaluation of performance is correct.
> >
> > Since the major points of my initial comments are not well-addressed, I keep my rating at this stage.

---

> ### Comment · Reviewer_hFv3 · 2024-12-02
>
> Dear reviewer bNe3, I would like to express my opinion on your second question. In theory, it is better to use high-degree steerable features. This is not only the consensus in the industry [a], but also discussed in articles such as HEGNN. The core contribution of this paper is to use the inner product of high-degree steerable features instead of the tensor product, thereby reducing the complexity of $\mathcal{O}(L^6)$ to $\mathcal{O}(L^2)$.
>
> [a] Official Commentby Reviewer ZLS9. 02 Dec 2024, 23:30. SE(3)-Hyena Operator for Scalable Equivariant Learning. Submitted to ICLR 2025.

---

> ### Author Response · Authors · 2024-12-02
>
> > Presentation
>
> Thank you for your constructive feedback. We are dedicated to improving the presentation of our work while preserving the essential technical details that distinguish our method.
>
> The key components of our method are illustrated in Figure 2: specifically, how the degree-wise inner-product operations in the HTR component continuously refine scalar edge features (Figure 2(c)), and how these enhanced edge features guide the attention mechanism (Figure 2(b)). While we understand the desire for simplification, we believe maintaining these technical details is crucial as they represent the core innovations of our work. However, we will improve the visual presentation to make these elements and their interactions more clear.
>
> We have already incorporated several improvements suggested by reviewers bNe3 and hFv3:
>
> - Adopted consistent mathematical notation throughout
> - Better contextualized our work within chronological advancements in the field
> - More clearly emphasized our method's contributions to future research directions
>
> These revisions enhance the accessibility of our technical innovations while maintaining the necessary mathematical rigor.
>
> As today marks the end of the rebuttal period, we want to ensure we fully understand your vision for Figure 2's improvement. While we have outlined our completed and planned revisions above, could you provide specific suggestions for modifications that would better highlight our key innovations, while preserving the technical completeness that is essential to our contribution? Your concrete feedback would be valuable in guiding our revision.
>
> > Efficiency-Expressiveness Trade-off
>
> As noted by reviewer hFv3, we achieve this trade-off by utilizing the degree-wise inner product of high-degree features instead of tensor products. This crucial technical innovation enables us to maintain expressiveness while significantly reducing computational complexity compared to methods using CG coefficients Notably, as acknowledged in [a], higher degree features are indeed theoretically superior and lead to better empirical results when trained on large-scale datasets. Our method specifically addresses the computational challenges, thereby making it practical to leverage expressive representations at scale. We have highlighted this contribution more explicitly in both the introduction and related work sections of our revised manuscript.
>
> > OC20 IS2RE Experiment
>
> We appreciate your clarification about OC20 IS2RE versus S2EF. To be clear: our previous response about OC20 was referencing the authors of EquiformerV2, who explicitly noted [7] that OC20 datasets require substantially different hyperparameters for optimal performance. Thus, using hyperparameters optimized for MD17/22 without proper adaptation and validation would not provide a fair comparison.
>
> **The significance of dataset-specific hyperparameter optimization**:
> To illustrate the **significance of dataset-specific hyperparameter optimization**, we can examine Equiformer's implementation. For MD17, they use [one set of hyperparameters](https://github.com/atomicarchitects/equiformer/blob/dc4852858a305bf552506321a28a81f981736f8a/nets/graph_attention_transformer_md17.py#L408), while for OC20RE they use [substantially different hyperparameters](https://github.com/atomicarchitects/equiformer/blob/master/oc20/configs/is2re/all/graph_attention_transformer/l1_256_nonlinear_g%402_local.yml). The differences are substantial, affecting critical parameters such as $L_{max}$ values, node dimensions, number of layers, irreps MLP dimensions, alpha dropout, and number of attention heads, among others. This point was also recently highlighted in an Equiformer V2 GitHub issue [7].
>
> **A thorough evaluation takes significant time:**
> A complete evaluation would require adapting our implementation for OC20, validating it, and conducting thorough hyperparameter optimization. While each training run takes 48 hours on 2 GPUs, a proper hyperparameter search would require multiple runs to optimize various parameters (architecture, learning rate, dropout, etc.). This process could take weeks to complete properly. Conducting a rushed evaluation during the rebuttal period would not be scientifically sound, as it could lead to suboptimal results that misrepresent our method's true capabilities and potentially mislead future readers.
>
> **Our evaluation for GotenNet:**
> Instead, **we have validated our method on four diverse well-benchmarked datasets (QM9, Molecule3D, MD17, and MD22)** - an evaluation scope that exceeds many recent significant contributions:
> - DimeNet, and LeftNet validated on QM9/MD17
> - LSRM validated on MD22
> - SO3Krates utilized MD17/22
>
> This comprehensive evaluation across multiple established benchmarks provides strong evidence of our method's effectiveness and generalizability. **We are certainly excited to explore OC20 IS2RE performance in future work.**

---

### Official Review · Reviewer_4mUv · 2024-11-04

**Soundness:** 3
**Presentation:** 3
**Contribution:** 3
**Rating:** 8
**Confidence:** 3

**Summary:**

The paper introduces GotenNet, a Geometric Tensor Network, that advances molecular representation learning by addressing computational challenges in 3D graph neural networks. GotenNet's novel tensor embedding strategy eliminates the need for complex irreducible representations and Clebsch-Gordan transformations, enhancing computational efficiency while preserving model expressiveness. Its Geometry-Aware Tensor Attention mechanism enables refined edge representations, capturing complex geometric relationships for better molecular property predictions. Additionally, the Hierarchical Tensor Refinement approach allows the model to adapt across scales, accommodating both broad patterns and detailed molecular features. Evaluated on benchmark datasets such as QM9 and Molecule3D, GotenNet consistently outperforms existing methods, demonstrating robustness and scalability.

**Strengths:**

1. The paper presents a novel framework, GotenNet, which innovatively combines geometric tensor representations with advanced attention mechanisms. This approach addresses the expressiveness-efficiency trade-off in 3D graph modeling, a challenge that has been inadequately tackled in prior works.
2. The paper is well-structured and clearly articulates the problem being addressed, the proposed solutions, and the significance of the findings.

**Weaknesses:**

1. **Comparison of computational complexity with previous methods**: Could you provide details on the model size, as well as training and inference times, in comparison with existing methods? This information would help highlight GotenNet’s computational efficiency relative to other models in the field.

**Questions:**

Please see the weakness part.

---

> ### Author Response · Authors · 2024-11-20
> **Comparison of computational complexity**
>
> We sincerely thank the reviewer for the thorough evaluation of our work. We appreciate the recognition of GotenNet's novel contributions in combining geometric tensor representations with advanced attention mechanisms, as well as acknowledging our paper's clear structure and articulation. The reviewer's supportive comments on how our approach addresses the expressiveness-efficiency trade-off in 3D graph modeling are particularly encouraging.
>
> > W1. **Comparison of computational complexity with previous methods**: Could you provide details on the model size, as well as training and inference times, in comparison with existing methods? This information would help highlight GotenNet’s computational efficiency relative to other models in the field.
>
> We provide detailed computational complexity comparisons in the table below:
>
>
> | Model          | Batch Size | Time per Epoch (s) | min      | avg      | max      | Limit    | Training Latency (ms) | Inference Latency  (ms) | Params | std.     | log       |
> | -------------- | ---------- | ------------------ | -------- | -------- | -------- | -------- | --------------------- | ----------------------- | ------ | -------- | --------- |
> | Equiformer     | 128        | 425                | 1.48     | 1.48     | 1.48     | 1.48     | 421                   | 150                     | 3.5M   | 0.70     | -5.82     |
> | EquformerV2    | 64         | 821                | 2.85     | 2.85     | 2.85     | 2.65     | 918                   | 341                     | 11.2M  | 0.67     | -5.87     |
> | EquformerV2    | 48         | 847                | 2.94     | 2.94     | 2.94     | 2.65     | 918                   | 341                     | 11.2M  | 0.67     | -5.87     |
> | Geoformer      | 32         | 436                | -        | -        | -        | 5.05     | 759                   | 264                     | 50.6M  | 0.75     | -6.12     |
> | GotenNet$_{S}$ | 32     | **117**            | **0.41** | **0.75** | **1.34** | **1.35** | **80**                | **37**                  | 6.1M   | **0.67** | **-6.21** |
> | GotenNet$_{B}$ | 32         | 180                | 0.75     | 1.15     | 1.92     | 2.08     | 120                   | 56                      | 9.2M   | **0.61** | **-6.26** |
> | GotenNet$_{L}$ | 32         | 291                | 1.37     | 1.87     | 2.33     | 3.37     | 244                   | 112                     | 18.3M  | **0.56** | **-6.34** |
>
>
> GotenNet demonstrates superior efficiency across all variants. GotenNet$\_{S}$ achieves competitive performance with just 6.1M parameters and the lowest latencies (80ms training, 37ms inference). Even our largest variant, GotenNet$_{L}$, maintains 42% faster inference and 25% faster training compared to Equiformer while achieving state-of-the-art performance. We have included detailed efficiency comparisons discussion in Appendix G. We welcome any further discussion regarding computational efficiency or other aspects of our work.

---

> > ### Comment · Reviewer_4mUv · 2024-11-22
> > **Thank you for the response**
> >
> > Thanks for your response. I have addressed my concerns and will increase the confidence score while maintaining the good paper recommendation.

---

> > > ### Author Response · Authors · 2024-11-24
> > >
> > > We sincerely appreciate your thorough review and are pleased that our response have addressed your concerns. Thank you for your time and feedback throughout this process.

---

### Public Comment · ~Tatsunori_Taniai1 · 2024-11-27
**Ask for Clarifications**

Dear Authors,

I am a researcher working on GNNs and transformers for molecules and crystals. I came across this interesting paper and was very impressed by its strong results. However, as Reviewer bNe3 was concerned, several parts are still unclear to me even in the revised version. Hoping to help the authors to improve the presentation, I would like to make some questions and comments about the method.

---
## **1) About the composition of feature tensors**
### **[Resolved] 1-1) What are the dimensions of features $h$ and $\tilde{v}^{(l)}$ and others?**
I suppose that the edge tensor $r_{ij}$ represents spherical harmonic functions: $r = \\{ r^0, r^1, ..., r^{L_{max}} \\}$ where $L_{max}$ is the maximum degree of spherical harmonics. Each degree $l$ has $(1 + 2l)$ components. Thus, $r$ constructs a pyramidal dimensional tensor,  shaped like below:

　　▢　　　$l = 0$
　▢▢▢　　$l = 1$
▢▢▢▢▢　$l = 2$ ($= L_{max}$)

This is my understanding of "Geometric Tensor Notations" in Sec 3.1 and it is clear to me. Then, from line 231, it is written that geometric tensors are decomposed into scalar features $h$ and steerable features $\tilde{v}^{(l)}$. I suppose $h$ corresponds to feature components of degree $l = 0$ and $\tilde{v}^{(l)}$ to those of higher degrees $l \in \\{1,2,...,L_{max}\\}$, right?

My question here is whether $\tilde{v}^{(l)}$ is further structured to have $(1 + 2l)$ components or not? In other words, what are the dimensions of $h$ and $\tilde{v}^{(l)}$? Are they $h \in \mathbb{R}^{d_{ne}}$ and $\tilde{v}^{(l)} \in \mathbb{R}^{(1 + 2l)\times d_{ne}}$ with some feature dimension $d_{ne}$ ? Or simply $h \in \mathbb{R}^{d_{ne}}$ and $\tilde{v}^{(l)} \in \mathbb{R}^{d_{ne}}$ ?
Carefully clarifying such dimensionality information for $h$ and $\tilde{v}^{(l)}$ and others (such as $m_i$ in Eq 1 and $t_{ij}$ in Eq 3 and Eq 12) will be much helpful for readers.

**Answer**: $\boldsymbol{h} \in \mathbb{R}^{d _{ne}}$ and $\tilde{\boldsymbol{X}}^{(l)} \in \mathbb{R}^{(1+2l) \times d _{ne}}$.

### **[Resolved] 1-2) Where does $S := (1 + 2L_{max})$ come from?**
From several Equations (such as Eqs 7, 8, 9) and texts, I see that feature tensors, such as $V$, $\text{SAE}$, and $o_{ij}$, are structured to have $(1 + 2L_{max})$ components. However, I could not understand the meaning of $(1 + 2L_{max})$. It corresponds to the number of spherical harominic components at the max degree $l = L_{max}$ but why pariticularly the max degree? It would make more sense to me if $(1 + 2L_{max})$ were actually the number of components per degree: $(1 + 2l)$ or the total number of components across degrees: $\sum_{l=0}^{L_{max}} (1 + 2l)$.

At line 300, the paper says that "The $S$ variable introduced to generate different coefficients for each degree of steerable features and formulated as $(1 + 2\times L_{max})$". Stating that $S$ is a variable despite $(1 + 2L_{max})$ is a constant suggests that the authors actually meant $S = (1 + 2l)$ instead of $S = (1 + 2L_{max})$ ? Even with this interpretation, Sec 3.2 still doesn't make sense to me.

Perhaps, Sec 3.2 could actually explain the operations for each target degree $l$, assuming $S = (1 + 2l)$. In this case, $L_{max}$ in several places (such as in $S$, Eq 7 and Eq 8) were actually this target degree $l$ ?

Where does $(1 + 2L_{max})$ come from? Which level is Sec 3.2 about? I really want to know these to understand the paper correctly.

**Answer**: $S = 1 + 2L_{max}$ is an authors' architectural design choice, chosen to implement Eq 8.

### **[Resolved] 1-3) $l$ in Eq 8 is unclear**
Eq 8's essential form is $\Delta v_i^{(l)} = \sum_j ( \\{ x_{ij}^{(l)} \\}_{l=1} ^{L})$, which is unclear because two types of $l$ exist. While $l$ in the left-hand side is a given degree number, $l$ in the right-hand side is a loop variable enumerating $1, 2, ..., L$. Which of the followings is the correct intepretation?
- $\Delta v_i^{(l)} = \sum_j x_{ij}^{(l)}$ where $l \in \\{ 1, 2, ..., L \\}$ (Here the operation is degree-wise)
- $\Delta v_i^{(l)} = \sum_j \text{Concatenate}( \\{ x_{ij}^{(k)} \\}_{k=1}^{l} )$  (Here $l$ gives the max for $k$, assuming $S = 1+2l$.)
- $\Delta v_i^{(l)} = \sum_j \text{Concatenate}( \\{ x_{ij}^{(k)} \\}_{k=1}^{L} )$  (Here $l$ in the left-hand side has no effect on the right-hand side)
- Something else

**Answer**: $\Delta v_i^{(l)} = \sum_j x_{ij}^{(l)}$ where $l \in \\{ 1, 2, ..., L \\}$

---

> ### Public Comment · ~Tatsunori_Taniai1 · 2024-11-27
> **Continued**
>
> ### **[Resolved] 1-4) What is the intuition for decomposing $o$ into $o^s$, $o^d$, $o^t$ in Eq 7 and Eq 8?**
> In Eq 7, $o$ is decomposed into $o^s$ of one component, $o^d$ of $L_{max}$ components, and  $o^t$ of $L_{max}$ components. Then, in Eq 8, $o^s$ is used for scalar $h$, and $o^d$ and $o^t$ are used for steerable $v$. Could you provide an intuitive interpretation to Eq 8 ? What's the difference between $o^s$, $o^d$, and $o^t$? I guess $o^s$ represents scalar features, while  $o^d$ and $o^t$ represent steerable features. Then, what is the purpose of  $o^d$ and $o^t$ ? Why are $o^d$ and $o^t$ processed differently (ie, multiplied by either $r_{ij}$ or $\tilde{v}_{j}^{(l)}$) in Eq 8 ?
>
> **Answer**: $o^s$, $o^d$, and $o^t$ are all invariant features, produced to implement Eq 8.
>
> ### **[Resolved] 2) Missing $\phi$ in  Eq 1**
> At line 262, it goes: "A cutoff function $\phi(r_{ij}^{(0)})$ is applied to ....". However, there seems no $\phi$ in Eq 1.
>
> ### **[Resolved] 3) About $z$ in Eq 1 and Eq 2**
> It says that "$z$ denotes the one-hot encoding of the atomic number". I suppose $z_j$ is a $|Z|$-dimensional one-hot vector expressing $j$'s atomic number (say, $k$) and it extracts the $k$-th row of matrix $A$ when written as $z_j A$. If so, I think it is better to write it in bold as either $\boldsymbol{z}_j^T \boldsymbol{A}$ (if the paper assumes a column vector) or  $\boldsymbol{z}_j \boldsymbol{A}$ (if a row vector).
>
> ### **[Resolved] 4) $t_{ij,init}W_{rs,ini}$ in Eq 4 is unclear**
> At line 285, the paper says that  "$W_{rs,init} \in \mathbb{R}^{d_{ed}\times L_{max} \times d_{ne}}$ is a learnable weight matrix". Strictly speaking, this $W$ is a tensor rather than a matrix. So, how the vector-tensor multiplication $t W$ is defined is unclear to me. Also, I suppose $t_{ij,init}$ is a vector, but its dimension is unclear. Is it $t_{ij,init} \in \mathbb{R}^{d_{ed}}$ where $d_{ed}$ is edge feature dimension?
>
> ### **[Could be improved] 5) What is $E$ in Eq 10 ?**
> I think $E$ in Eq 10 is never explained. What is it? How it is defined? How does it work?
>
> ### **[Resolved] 6) What is the difference between $\bigoplus$ and Agg ?**
> It is unclear from the paper why there are these two types of aggretation operators. Is that because Agg doesn't necessarily have to be permutation invariant?
>
> **Answer**: Correct
>
> ### **[Resolved] 7) What is the intuition for Eq 13 ?**
> I would like to know an intuitive interpretation to Eq 13. I guess that $m_1$ and $m_2$ are designed to be invariant, because both of scalar features $h$ and the L2 norm of steerable features $||\tilde{v} W_{vu} ||_2$ are invariant (?). The form of $\tilde{v} + m_2 \circ (\tilde{v} W _{vu})$ keeps equivariance while increasing the model's expressibility ? Does $W _{vu}$ have to be identical in the two equations to ensure equivariance?
>
> **Answer**: Correct. $W _{vu}$ can differ.
>
> ### **A) Additional comments**
> - **[Resolved]** A-1) In Eq 1: operator $\circ$ is unclear. I can guess $\circ$ is element-wise multiplication. But it would be more reader-friendly if it is explicitly explained after Eq 1.
> - **[Resolved]** A-2) In Eq 5: I think $Q$, $K$, and $V$ in Eq 5 should have subscripts of $i$ or $j$.
> - **[Could be improved]** A-3) In Eq 5 and others: I often see inconsistent notation styles for scalars, vectors, matrices, and tensors, which confused me. I think it would be helpful if the paper follows a consistent notation style, such as non-bold $x$ for scalars, bold $x$ for vectors, bold ${X}$ for matrices and tensors. (So, I think $Q$ and $K$ in Eq 5 should be $q_i$ and $k_j$ in bold.)
> - **[Resolved]** A-4) Using $v$ for steerable features may confuse with value features of self-attention. I suggest replacing $v$ with, e.g., $x$.
> - **[Resolved]** A-5) In Eq 13: Is $\cup$ concatenation? If so, it is better to use the same notation as in Eq 2, or use $(a,b)$ instead of $(a \cup b)$.
> - A-6) Perhaps, the method could be better explained by moving Sec 3.1 later, since Sec 3.1 relies on one of the main modules, SAE. In this case, the authors could explain the main modules first, given some initialized features, and then provide the initialization procedures. This order may highlight the main contribution part.
> - **[Could be improved]** A-7) In Figure 2: The captions of "Structure Embedding" in (a) and "Node Embedding" for (d) are inconsistent, if they refer to the same module.
> - **[Resolved]** A-8) In Fugure 2: The caption texts in Figure 2 use operator notations that are different from main texts, such as $\circledcirc$ vs $\circ$ for element-wise (Hadamard) product. They should be consistent throught the paper. Also, there is a typo "concatinationconcatenation".
>
> ---
> Since the paper is really interesting, I would like to deeply understand this work! If the authors could answer to these questions or update the paper presentation to address them, that would be very helpful.
>
> Thank you for the interesting work!
>
> Sincerely,
> Tatsunori Taniai

---

> > ### Comment · Reviewer_hFv3 · 2024-11-28
> >
> > Dear Dr. Taniai, I think I can answer your fifth question. $\boldsymbol{EQ}$ in Eq. (10) should mean "equivariant query", and $E$ may mean that this is a steerable variable and should be a whole (i.e. $\widetilde{\boldsymbol{EQ}}$). In addition, I am very grateful for your many suggestions on presentation. In fact, I was also very confused about the presentation of the first version of this article at the beginning. Although I later suggested that the authors use the HEGNN symbol system, the authors may have made many omissions in the revision due to time constraints (so I must also admit that part of the responsibility lies with me).
> >
> > I agree with the notation you mentioned in 7-3. However, in order to distinguish single-channel and multi-channel scalars, I would like to make some improvements based on your suggestion. In fact, it is the notation used in HEGNN:
> > - $x\in\mathbb{R}$ represents a single-channel scalar
> > - $\boldsymbol{x}\in\mathbb{R}^{C}$ represents a multi-channel scalar (the number of channels is $C$)
> > - $\vec{\boldsymbol{x}}\in\mathbb{R}^{3}$ represents a Cartesian vector
> > - $\vec{\boldsymbol{X}}\in\mathbb{R}^{3\times C}$ represents a Cartesian vector group, which contains $C$
> > - $\tilde{\boldsymbol{x}}^{(l)}\in\mathbb{R}^{2l+1}$ represents a $l$th-degree steerable vector
> > - $\tilde{\boldsymbol{X}}^{(l)}\in\mathbb{R}^{(2l+1)\times C}$ represents the $l$th-degree steerable vector group, which contains $C$
> > - $\tilde{\boldsymbol{x}}^{(0:L)}\in\mathbb{R}^{L^2}$ represents the $0$th to $L$th-degree steerable vector group
> > - $\tilde{\boldsymbol{X}}^{(0:L)}\in\mathbb{R}^{L^2\times C}$ represents the $0$th to $L$th-degree steerable vector group, which contains $C$ for each degree
> > - If different degrees are different, it is recommended to use e3nn's string representation directly
> >
> > It is worth noting that Cartesian vectors are also 1st-degree steerable features, but because they are used so frequently, they are given a separate symbol. In addition, this also makes it easier to discuss translation equivariance separately, because steerable features are generally set to be translation invariant.

---

> > > ### Public Comment · ~Tatsunori_Taniai1 · 2024-11-28
> > > **Thank you Reviewer hFv3 for response**
> > >
> > > Dear Reviewer hFv3,
> > >
> > > Thank you for your response and clarification on Q 5. I totally agree with you on the benefit of using HEGNN's notation. If the paper strictly follows this notation throughout, it will greatly reduce ambiguity and enhance readability. In my opinion, the paper in its current state requires mature eyes of experts, such as Reviewer hFv3, to accurately understand the content. I am not against this work, but I sincerely hope that the paper's presentation will be improved before publication to make it accessible to a broad audience, for the sake of the community.
> > >
> > > Best,
> > > Tatsunori

---

> > > > ### Comment · Reviewer_hFv3 · 2024-11-28
> > > >
> > > > Dear Dr. Taniai,
> > > > I couldn't agree more with your point of view. A good article needs a good presentation, so that it can guide the newcomers (especially beginners) in the whole field. Especially for this article, which is bound to become a must-read in equivariant models, a clear and easy-to-follow architecture explanation is very necessary. Enthusiastic people like you are very much needed to provide suggestions (especially those points that hinder understanding), which I believe will also be welcomed by the authors. Thank you again for your valuable insights.

---

> > ### Author Response · Authors · 2024-11-28
> >
> > Dear Dr. Taniai,
> >
> > Thank you for your detailed feedback on our manuscript. As authors, we are committed to making our research clear and understandable for future readers. We truly appreciate your constructive feedback and keen interest in our work.
> >
> > We have revised the manuscript to align with the notation system suggested by reviewer hFv3 (1-1, A-2, A-3, A-4). Most notably replacing $\boldsymbol{\tilde{v}}$ with $\boldsymbol{\tilde{X}} \in \mathbb{R}^{(1 + 2l)\times d_{ne}}$ for improved clarity. We have also revised our manuscript to clarify (1-3, 2, 3, A-1, A-5, A-8).
> >
> > (1-2, 1-4) The value of $S$ is determined by the total number of scalar outputs. Specifically, $\\| \\{ \mathbf{o}^{s}\_{ij} \\}\\| + \\|\\{\mathbf{o}^{d, (l)}\_{ij}\\}^{L\_{\max}}\_{l=1}\\| + \\|\\{\mathbf{o}^{t, (l)}\_{ij}\\}^{L\_{\max}}\_{l=1}\\| = 1 + L\_{\max} + L\_{\max} = (1 + 2L\_{\max})$. While it happens to match the number of components at degree $l$, this is coincidental. The intuition for decomposing $\boldsymbol{o}$ into $\boldsymbol{o}^s, \boldsymbol{o}^d, \boldsymbol{o}^t$ is to denote their usage, with all components being invariant scalars. The scalar $\boldsymbol{o}^s$ updates the scalar representations $\boldsymbol{h}$, while for each degree $l$ (where $l \in [1, L\_{\max}]$), the coefficients $\boldsymbol{o}^{d, (l)}\_{ij}$ and $\boldsymbol{o}^{t, (l)}\_{ij}$ scale the $l$-degree $\tilde{\boldsymbol{r}}^{(l)}\_{ij}$ and steerable features $\boldsymbol{\tilde{X}}\_{j}^{(l)}$ respectively. For detailed performance comparisons between separate and shared coefficients, we refer you to our discussion with reviewer hFv3, which will be included in our appendix.
> >
> > (6) The key distinction is that $\texttt{Agg}$ is not required to be permutation invariant, unlike $\bigoplus$.
> >
> > (7) Your intuition about $\boldsymbol{m}\_1$ and $\boldsymbol{m}\_2$ being invariant is correct. $\textbf{W}\_{vu}$ could theoretically differ without compromising equivariance, we opted for shared weights for simplicity, as our experiments showed no significant performance advantage with separate weights.
> >
> > We hope these clarifications help explain the technical aspects of our work. Please don't hesitate to reach out with any additional questions or suggestions - we welcome further discussion to ensure our work is as clear and useful as possible!

---

> ### Public Comment · ~Tatsunori_Taniai1 · 2024-11-28
>
> Dear Authors,
>
> Thank you for your clarifications and revisions. While checking your reply and the paper, I'd like to write a quick comment on the updates because I know the deadline for the paper revision is very close.
>
> - While it is indeed helpful that $X$ follows the notation of HEGNN, I (and probably Reviewer hFv3) suggested that all geometric vectors/matrices/tensors (not limited to $X$ but including all the others such as $m_i$, $t_{ij}$, $q_i$, $k_j$, $v_j$, $sae_{ij}$, $\alpha_{ij}$) follow the consistent notation of HEGNN. Due to limited time, I cound not check completely if the paper already satisfies this or not.
> For example, $\boldsymbol{v}_j$ in Eq 5 seems to be a matrix in $\mathbb{R}^{S \times d _{ne}}$, so I think it's better to write $\boldsymbol{V}_j$ (?). The same style should apply to $\textbf{sae} _{ij}$ (as it has the same dimentionalities with $\boldsymbol{V}_j$). Likewise, if $\boldsymbol{\alpha} _{ij}$ in Eq 6 is a scalar, then it's better to write $\alpha _{ij}$.
>
> - **[Resolved]** Also, $v$ in Eq 6 probably needs the subscript $j$ ?
>
> - **[Resolved]** Additionally, I noticed that Q3 and Q4 are not addressed. Because of this, I think $z\boldsymbol{A}$ in Eq 1 and Eq 2 as well as $tW$ in Eq 4 are kind of ill-defined.
>
> - Another question: ~~Does the "sae" in Eq 7 have the dimensionalities of $S \times d_{de}$ ? If so, I suppose the splitting in Eq 7 results in $o^s _{ij}$, $o^{d,(l)} _{ij}$, $o^{t,(l)} _{ij}$ that are all scalars (ie, $o^s _{ij}, o^{d,(l)} _{ij} ,o^{t,(l)} _{ij} \in \mathbb{R}$). Correct? If so,  these $o$ should be non-bold in Eq 7 and Eq 8, and the two $\circ$ in Eq 8 are probably unnecessary.~~ Sorry, this was my misunderstanding. I suppose $o$'s are all vectors in $\mathbb{R}^{d _{de}}$, correct? If so, how are element-wise muptiplications in Eq 8, ie, $\boldsymbol{o}^{d,(l)} _{ij} \circ \tilde{\boldsymbol{r}}^{(l)} _{ij}$ and $\boldsymbol{o}^{t,(l)} _{ij} \circ \tilde{\boldsymbol{X}}^{(l)} _{j}$, defined? I suppose $\tilde{\boldsymbol{r}}^{(l)} _{ij} \in \mathbb{R}^{1 + 2l}$ and $\tilde{\boldsymbol{X}}^{(l)} _{j} \in \mathbb{R}^{(1+2l)\times d _{ne}}$ and they have different dimensionalities with vectors $\boldsymbol{o} _{ij} \in \mathbb{R}^{d _{de}}$.
>
> - The same question applies to $\boldsymbol{m}_2 \circ \tilde{\boldsymbol{X}}^{(l)} \boldsymbol{W} _{vu}$ in Eq 13.
>
> - Other suggestions:
>   - If $\tilde{\cdot}$ is used to represent steerable features, then notation of $\tilde{\boldsymbol{EQ}}^{(l)} _i$ and $\tilde{\boldsymbol{EK}}^{(l)} _j$  in Eq 10 seems redundant, because both $\tilde{\cdot}$ and $E$ express steerable features. I think simply writting them as  $\tilde{\boldsymbol{Q}}^{(l)} _i$ and $\tilde{\boldsymbol{K}}^{(l)} _j$ suffices.
>   - I think the notation of $\textbf{sae}_{ij}$ is not preferable. I suggest simply using a single capital-bold character (assuming that $\textbf{sae} _{ij}$ is a matrix in $\mathbb{R}^{S \times d _{ne}}$).
>
> Thank you

---

> ### Public Comment · ~Tatsunori_Taniai1 · 2024-11-28
> **Thanks for the paper updates**
>
> Dear Authors,
>
> I noticed new revisions in the paper. I deeply thank the authors. Given these revisions, Sec 3 is mostly clear to me. Now I have much better understanding of the method. Although there is still slight abuse of notation (eg, summing a matrix and a vector with implicit vector-matrix reshaping), I think that is allowable.
>
> Meanwhile, there are equations that still seem ill-defined. I list them below, hoping that they are addressed in the final version:
> - Definitions of $\boldsymbol{o}^{d,(l)} _{ij} \circ \tilde{\boldsymbol{r}}^{(l)} _{ij}$ and $\boldsymbol{o}^{t,(l)} _{ij} \circ \tilde{\boldsymbol{X}}^{(l)} _{j}$ in Eq 8 are unclear, given $\boldsymbol{o}^{(l)} _{ij} \in \mathbb{R}^{d _{de}}$,  $\tilde{\boldsymbol{r}}^{(l)} _{ij} \in \mathbb{R}^{(1 + 2l)}$, and $\tilde{\boldsymbol{X}}^{(l)} _{j} \in \mathbb{R}^{(1+2l)\times d _{ne}}$.
>   - Probably, the former is $(\tilde{\boldsymbol{r}}^{(l)} _{ij})^T \boldsymbol{o}^{d,(l)} _{ij}$ and the latter multipies the same $\boldsymbol{o}^{t,(l)} _{ij}$ to all $(1+2l)$ components of $\tilde{\boldsymbol{X}}^{(l)} _{j}$ (ie, broadcast in numpy).
> - Definition of $\boldsymbol{m}_2 \circ \tilde{\boldsymbol{X}}^{(l)} \boldsymbol{W} _{vu}$ in Eq 13 is unclear.
>   - Probably, broadcasting applies to $\boldsymbol{m}_2 \in \mathbb{R}^{d _{ne}}$ again.
> - Similar issues for three $\circ$'s in Eq 4.
>
> Additionally, I would like to leave several suggestions, which will hopefully increase the clarity.
>
> - Sec 3.1:
>   - When introducing notation, I think it is important to clarify that $\tilde{\cdot}$ express steerable features that contains $L_{max}$ degrees and $(1+2l)$ components for each degree $l$. Although Sec 3.1 already explains so for $\tilde{\boldsymbol{r}}^{(l)} _{ij}$ and $\tilde{\boldsymbol{X}}^{(l)}$, it is important to clarify that this notation generally applies to other symbols.
>   - Also, it's better to clarify that **vectors in $\mathbb{R}^d$ are row vectors** in the paper.
> - Eq 1: Although I can guess, it is more reader-friendly to clarify $\boldsymbol{m}_i \in \mathbb{R}^{d _{ne}}$ and $\boldsymbol{z} \in \mathbb{R}^{|\mathcal{Z}|}$.
> - Eq 3
>   - Please clarify $\boldsymbol{t} _{ij,init} \in \mathbb{R}^{d _{ed}}$. I think $\boldsymbol{t} _{ij}$ is as important as $\boldsymbol{h}$ and $\tilde{\boldsymbol{X}}^{(l)}$. Thus, such dim information is important.
>   - $(\sigma(\cdot))$ is simplified to $\sigma(\cdot)$ (same for Eq 2). Instead, Eq 3 needs to be clarified if $W$ is applied after or before $\circ$, by inserting ( ) at proper positions. Ie, there is ambiguity whether $(( t_i + t_j ) \circ \sigma (\cdot)) W$ or $( t_i + t_j ) \circ (\sigma (\cdot) W)$.
> - Eq 5: Given $\boldsymbol{v}_j$ in vector style, it's better to define $\gamma_v: \mathbb{R}^{d _{ne}} \to \mathbb{R}^{S \cdot d _{ne}}$ instead of $\mathbb{R}^{S \times d _{ne}}$ at the line after Eq 5. Accordingly, Eq 7 should use $\gamma_s: \mathbb{R}^{d _{ne}} \to \mathbb{R}^{S \cdot d _{ne}}$. This way, the notation abuse (ie, vector + matrix) is also resolved.
> - Eq 6: If $\boldsymbol{\alpha} _{ij}$ is a scalar in $\mathbb{R}$, then I strongly recommend to write non-bold ${\alpha} _{ij}$.
> - Eq 9 and Eq 12: I suggest using "$\gets$" instead of "$=$", ie, $X \gets X+\Delta X$. The same applies to $h$ in Eq 9 and $t$ in Eq 12.
> - Overall Sec 3: At first, the high-level objectives of Sec 3.3 & 3.4 were unclear to me. Now I have a better perspective:
>   - The network holds node features (ie, invariant $\boldsymbol{h}_i$ and steerable $\tilde{\boldsymbol{X}}^{(l)} _i$) and edge features (ie, invariant $\boldsymbol{t} _{ij}$) throughout, initializes them in Sec 3.2, and repeatedly updates $\boldsymbol{h}_i$ and $\tilde{\boldsymbol{X}}^{(l)} _i$ in Sec 3.3 & 3.5 as well as $\boldsymbol{t} _{ij}$ in Sec 3.4.
>   - Sec 3.3 updates $\boldsymbol{h}_i$ and $\tilde{\boldsymbol{X}}^{(l)} _i$ using attention-based message passing in a degree-wise manner, where the information of $\tilde{\boldsymbol{X}}^{(l)}$ is not mixed across degrees $l$. Equivariance is ensured by relying on invariant features (ie, $\boldsymbol{h}_i$, $\boldsymbol{t} _{ij}$, and $\tilde{\boldsymbol{r}}^{(0)} _{ij}$) to compute coefficients for steerable $\tilde{\boldsymbol{X}}^{(l)} _i$ as well as updates for invariant $\boldsymbol{h}_i$.
>   - Sec 3.4 updates $\boldsymbol{t} _{ij}$ using the information of $\tilde{\boldsymbol{X}}^{(l)} _i$ and $\tilde{\boldsymbol{X}}^{(l)} _j$ mixed across degrees $l$, by utilizing their inner products to produce invariant updates for $\boldsymbol{t} _{ij}$.
>   - Sec 3.5 updates $\boldsymbol{h}_i$ and $\tilde{\boldsymbol{X}}^{(l)} _i$ using a node-wise & degree-wise feed-forward net.
>   - Providing such high-level views before presenting details will help readers.
>
> Since Reviewer bNe3 seems absent and his/her clarity concern remains alive, I'd like to personally endorce the paper's clarity. I think the paper (Sec 3) is much clearer now, and addressing the above points will further enhance clarity.
>
> Best,
> Tatsunori

---

> ### Author Response · Authors · 2024-12-01
>
> Dear Dr. Taniai,
>
> Thank you for your thorough review and continuing engagement with our manuscript, especially during this Thanksgiving week. We truly appreciate your dedication to improving our work during the holiday period. We are **particularly grateful for your endorsement of the paper's clarity and your suggestions that will help further improve the final version**.
>
> We will refine our notation throughout Section 3 to enhance clarity and consistency. Regarding your specific comments:
>
> - We will improve the notation based on your suggestions to make the equations more precise and consistent.
> - We will maintain our original notation using for element-wise operations, and have added explicit clarification about its broadcasting behavior in the manuscript to ensure clarity.
> - We will introduce $\tilde{\cdot}$ notation in Section 3.1 to consistently denote steerable features containing $l$-degree features with $(1+2l)$ components for each $l$ until $L_{\max}$.
> - We will clarify dimensions throughout Section 3.2, including $\boldsymbol{m}\_i \in \mathbb{R}^{d_{ne}}$, $\boldsymbol{z} \in \mathbb{R}^{|\mathcal{Z}|}$, and $\boldsymbol{t}\_{ij,\text{init}} \in \mathbb{R}^{d_{ed}}$.
> - We will update the function definitions after Equations 5 and 7 to $\gamma_v, \gamma_s: \mathbb{R}^{d_{ne}} \to \mathbb{R}^{S \cdot d\_{ne}}$ to maintain consistency with vector representations and resolve notation ambiguity. We will also improve equation clarity by adjusting parentheses placement in Eq 3: $\boldsymbol{t}\_{ij, \text{init}} = (\boldsymbol{h}\_{i, {\text{init}}} + \boldsymbol{h}\_{j, {\text{init}}}) \circ \Bigg(\sigma\Big(\mathrm{LN}\big(\varphi(\tilde{\boldsymbol{r}}^{(0)}\_{ij})\mathbf{W}\_{\text{erd}}\big)\Big)\mathbf{W}\_{\text{eru}}\Bigg).$
> - We keep $\boldsymbol{\alpha}\_{ij}$ in bold as it is a vector $\boldsymbol{\alpha}\_{ij} \in \mathbb{R}^c$ containing attention weights for each head, where $c$ is the number of attention heads.
> - We will adopt the $\gets$ symbol for update equations where appropriate and improve Figure 2's caption clarity.
> - We will incorporate your excellent high-level overview of the architecture into Section 3, which indeed helps readers better understand the components before diving into details.
>
> All these improvements will appear in the final version of our manuscript. Thank you again for your detailed feedback and commitment to enhancing the clarity of our work!

---

### Meta-Review · Area_Chair_HWcB · 2024-12-20

**Metareview:**

This paper develops a new $E(3)$ equivariant transformer architecture for molecular data modeling. At the core of the method is a Geometry Aware Tensor Attention (GATA) acting on input node and edge features and providing a good tradeoff between computational complexity and expressiveness. Initial reviewers’ concerns were regarding presentation and exposition of the paper. The authors addressed these issues during the rebuttal period to the satisfaction of most of the reviewers. All reviewers felt the experimental part is solid and demonstrates the efficacy of the suggested architecture.

**Additional Comments On Reviewer Discussion:**

No additional comments.

---

### Decision · Program_Chairs · 2025-01-22

Accept (Poster)